# STIMULUS: Achieving Fast Convergence and Low Sample Complexity in Stochastic Multi-Objective Learning

## Abstract

Recently, multi-objective optimization (MOO) problems have received increasing attention due to their wide range of applications in various fields, such as machine learning (ML), operations research, and many engineering applications. However, MOO algorithm design remains in its infancy and many existing MOO methods suffer from unsatisfactory convergence performance. To address this challenge, in this paper, we propose an algorithm called STIMULUS (stochastic path-integrated multi-graident recursive estimator), a new and robust approach for solving MOO problems. Different from the traditional methods, STIMULUS introduces a simple yet powerful recursive framework for updating stochastic gradient estimates. This methodology improves convergence performance by reducing the variance in multi-gradient estimation, leading to more stable convergence paths. In addition, we introduce an enhanced version of STIMULUS, termed STIMULUS-M, which incorporates the momentum term to further expedite convergence. One of the key contributions of this paper is the theoretical analysis for both STIMULUS and STIMULUS-M, where we establish an $\mathcal{O}(\frac{1}{T})$ convergence rate for both methods, which implies a state-of-the-art sample complexity of $O\left(n + \sqrt{n}\epsilon^{-1}\right)$ under non-convexity settings. In the case where the objectives are strongly convex, we further establish a linear convergence rate of $\mathcal{O}(\exp -\mu T)$ of the proposed methods, which suggests an even stronger $\mathcal{O}\left(n + \sqrt{n}\ln(\mu/\epsilon)\right)$ sample complexity. Moreover, to further alleviate the periodic full gradient evaluation requirement in STIMULUS and STIMULUS-M, we further propose enhanced versions with adaptive batching called STIMULUS$^+$/STIMULUS-M$^+$ and provide their theoretical analysis. Our extensive experimental results verify the efficacy of our proposed algorithms and their superiority over existing methods.

## 1 Introduction

Since its inception as a discipline, machine learning (ML) has heavily relied on optimization formulations and algorithms. While traditional ML problems generally focus on minimizing a single loss function, many emergent complex-structured multi-task ML problems require balancing *multiple* objectives that are often conflicting (e.g., multi-agent reinforcement learning (Parisi et al., 2014), multi-task fashion representation learning (Jiao et al., 2022; 2023), multi-task recommendation system (Chen et al., 2019; Zhou et al., 2023), multi-model learning in video captioning (Pasunuru & Bansal, 2017), and multi-label learning-to-rank (Mahapatra et al., 2023a;b)). Such ML applications necessitate solving *multi-objective* optimization (MOO) problems, which can be expressed as:

$$\min_{\mathbf{x} \in \mathcal{D}} \mathbf{F}(\mathbf{x}) := [f_1(\mathbf{x}), \cdots, f_S(\mathbf{x})], \tag{1}$$

where $\mathbf{x} \in \mathcal{D} \subseteq \mathbb{R}^d$ represents the model parameters, and each $f_s$ denotes the objective function of task $s \in [S]$, $f_s(\mathbf{x}) = \frac{1}{n} \sum_{j=1}^{n} f_{sj}(\mathbf{x}; \xi_{sj})$, $n$ denotes the total number of samples, $\xi_{sj}$ denotes the $j$-th sample for taks $s$. However, unlike traditional single-objective optimization, there may not exist a common $\mathbf{x}$-solution in MOO that can simultaneously minimize all objective functions. Instead, a more relevant optimality criterion in MOO is the notion of *Pareto-optimal solutions*, where no objective can be further improved without sacrificing other objectives. Moreover, in settings where the set of objective functions are non-convex, searching for Pareto-optimal solutions is intractable

in general. In such scenarios, the goal of MOO is usually weakened to finding a *Pareto-stationary solution*, where no improving direction exists for any objective without sacrificing other objectives.

To date, existing MOO algorithms in the literature can be generally categorized as gradient-free and gradient-based methods. Notably, gradient-based methods have attracted increasing attention recently due to their stronger empirical performances (see Section 2 for more detailed discussions). Specifically, following a similar token of (stochastic) gradient descent methods for single-objective optimization, (stochastic) multi-gradient descent (MGD/SMGD) algorithms have been proposed in (Fliege et al., 2019; Fernando et al., 2022; Zhou et al., 2022b; Liu & Vicente, 2021). The basic idea of MGD/SMGD is to iteratively update the x-variable following a common descent direction for all the objectives through a time-varying convex combination of (stochastic) gradients of all objective functions. Although MGD-type algorithms enjoy a fast $\mathcal{O}(1/T)$ convergence rate ($T$ denotes the number of iterations) in finding a Pareto-stationary solution, their $\mathcal{O}(n)$ per-iteration computation complexity in full multi-gradient evaluations becomes prohibitive when the dataset size $n$ is large. Further, in finding an $\epsilon$-stationary point for non-convex MOO (typical in ML), the high overall $\mathcal{O}(n\epsilon^{-1})$ sample complexity of MGD-type methods is not acceptable when $n$ is large. As a result, SMGD-type algorithms are often more favored in practice thanks to the lower per-iteration computation complexity in evaluating stochastic multi-gradients. However, due to the noisy stochastic multi-gradient evaluations, SMGD-type algorithms typically exhibit a slow $\mathcal{O}(1/\sqrt{T})$ convergence rate, which also induces a high $\mathcal{O}(\epsilon^{-2})$ sample complexity. Exacerbating the problem is the fact that, due to the complex coupling algorithmic structure between multiple objectives, SMGD-type methods are prone to divergence problems, particularly in scenarios with small batch and high variance (Liu & Vicente, 2021; Zhou et al., 2022a). In light of these major limitations of SMGD-type algorithms, a fundamental question naturally emerges:

*Is it possible to develop fast-convergent stochastic MOO algorithms in the sense of matching the convergence rate of deterministic MGD-type methods, while having a low per-iteration computation complexity as in SMGD-type algorithms, as well as achieving a low overall sample complexity?*

As in traditional single-objective optimization, a natural idea to achieve both fast convergence and low sample complexity in MOO is to employ the so-called "variance reduction" (VR) techniques to tame the noise in stochastic multi-gradients in SMGD-type methods. However, due to the complex coupling nature of MOO problems, developing VR-assisted algorithms for SMGD-type algorithms is far more challenging than their single-objective counterparts. First, since SMGD-type methods aim to identify the Pareto front (i.e., the set of all Pareto-optimal/stationary solutions), it is critical to ensure that the use of VR techniques does not introduce new bias into the already-noisy SGMD-type search process, which drives the search process toward certain regions of the Pareto front. Second, how to maintain solution diversity discovered by VR-assisted SMGD-type search process is challenging and yet important, particularly for the non-convex MOO problems. Last but the least, conducting theoretical analysis to prove the convergence performance of some proposed VR-based SMGD-type techniques also contains multiple challenges, including how to quantify multiple conflicting objectives, navigating trade-offs between them, handling the non-convexity objective functions, and managing the computational cost of evaluations. All of these analytical challenges are quite different from those in single-objective optimization theoretical analysis. To adapt variance reduction methods in SMGD-type algorithms, specialized proofs and analyses are needed to effectively tackle these challenges and facilitate efficient exploration of the Pareto optimality/stationarity.

The major contribution of this paper is that we overcome the aforementioned technical challenges and develop a suite of new VR-assisted SMGD-based MOO algorithms called STIMULUS (stochastic path-integrated multi-gradient recursive estimator) to achieve both fast convergence and low sample complexity in MOO. Our main technical results are summarized as follows:

- We propose a path-integrated recursive variance reduction technique in STIMULUS that is carefully designed for updating stochastic multi-gradient estimates. This technique not only enhances computational efficiency but also significantly reduces multi-gradient estimation variance, leading to more stable convergence trajectories and overcoming the divergence problem of SMGD. We theoretically establish a convergence rate of $\mathcal{O}(1/T)$ for STIMULUS in non-convex settings (typical in ML), which further implies a low sample complexity of $O\left(n + \sqrt{n}\epsilon^{-1}\right)$. In the special setting where the objectives are strongly convex, we show that STIMULUS has a linear convergence rate of $\mathcal{O}(\exp(-\mu T))$, which implies an even lower sample complexity of $\mathcal{O}\left(n + \sqrt{n}\ln(\mu/\epsilon)\right)$.

- To further improve the performance of STIMULUS, we develop an enhanced version called STIMULUS-M that incorporates momentum information to expedite convergence speed. Also, to relax the requirement for periodic full multi-gradient evaluations in STIMULUS and STIMULUS-M, we propose two enhanced variants called STIMULUS$^+$ and STIMULUS-M$^+$ based on adaptive batching, respectively. We provide theoretical convergence and sample complexity analyses for all these enhanced variants. These enhanced variants expand the practical utility of STIMULUS, offering efficient solutions that not only accelerate optimization processes but also alleviate computational burdens in a wide spectrum of multi-objective optimization applications.
- We conduct extensive numerical experiments on a variety of challenging MOO problems to empirically verify our theoretical results and illustrate the efficacy of the STIMULUS algorithm family. All our experimental results demonstrate the efficiency and superiority of the STIMULUS algorithm family over existing state-of-the-art MOO methods. These results also underscore the robustness, scalability, and flexibility of our STIMULUS algorithm family in complex MOO applications.

## 2 PRELIMINARIES AND RELATED WORK

To facilitate subsequent technical discussions, in this section, we first provide a primer on MOO fundamentals and formally define the notions of Pareto optimality/stationarity, $\epsilon$-stationarity in MOO, and the associated sample complexity. Then, we will give an overview of the most related work in the MOO literature, thus putting our work into comparative perspectives.

**1) Multi-objective optimization: A primer.** As introduced in Section 1, MOO aims to optimize multiple objectives in Eq. (1) simultaneously. However, since in general there may not exist an $\mathbf{x}$-solution that minimizes all objectives at the same time in MOO, the more appropriate notion of optimality in MOO is the so-called *Pareto optimality,* which is formally defined as follows:

**Definition 1** ((Weak) Pareto Optimality)**.** *Given two solutions* $\mathbf{x}$ *and* $\mathbf{y}$*,* $\mathbf{x}$ *is said to dominate* $\mathbf{y}$ *only if* $f_s(\mathbf{x}) \leq f_s(\mathbf{y}), \forall s \in [S]$ *and there exists at least one function,* $f_s$*, where* $f_s(\mathbf{x}) < f_s(\mathbf{y})$*. A solution* $\mathbf{x}$ *is Pareto optimal if no other solution dominates it. A solution* $\mathbf{x}$ *is defined as weakly Pareto optimal if there is no solution* $\mathbf{y}$ *for which* $f_s(\mathbf{x}) > f_s(\mathbf{y}), \forall s \in [S]$*.*

Finding a Pareto-optimal solution in MOO is as complex as solving single-objective non-convex optimization problems and is NP-Hard in general. Consequently, practical efforts in MOO often aim to find a solution that meets the weaker notion called Pareto-stationarity (a necessary condition for Pareto optimality), which is defined as follows Fliege & Svaiter (2000); Miettinen (2012):

**Definition 2** (Pareto Stationarity)**.** *A solution* $\mathbf{x}$ *is Pareto-stationary if no common descent direction* $\mathbf{d} \in \mathbb{R}^d$ *exists such that* $\nabla f_s(\mathbf{x})^\top \mathbf{d} < 0, \forall s \in [S]$*.*

It is worth noting that, since non-convex MOO problems are intractable in general, the goal in solving non-convex MOO problems is often to identify Pareto-stationary points that serve as the necessary condition for Pareto-optimality. Note also that in the special setting with strongly convex objective functions, Pareto-stationary solutions are Pareto-optimal.

Following directly from Pareto-stationarity in Definition 2, gradient-based MOO algorithms strive to find a common descent (i.e., improving) direction $\mathbf{d} \in \mathbb{R}^d$, such that $\nabla f_s(\mathbf{x})^\top \mathbf{d} \leq 0, \forall s \in [S]$. If such a direction does not exist at $\mathbf{x}$, then $\mathbf{x}$ is Pareto-stationary according to Definition 2. Toward this end, the MGD method (Désidéri, 2012) identifies an optimal weight $\boldsymbol{\lambda}^*$ for the multi-gradient set $\nabla \mathbf{F}(\mathbf{x}) \triangleq \{\nabla f_s(\mathbf{x}), \forall s \in [S]\}$ by solving $\boldsymbol{\lambda}^*(\mathbf{x}) \in \operatorname{argmin}_{\boldsymbol{\lambda} \in C} \|\boldsymbol{\lambda}^\top \nabla \mathbf{F}(\mathbf{x})\|^2$. Consequently, the common descent direction can be defined as $\mathbf{d} = \boldsymbol{\lambda}^\top \nabla \mathbf{F}(\mathbf{x})$. Then, MGD follows the iterative update rule $\mathbf{x} \leftarrow \mathbf{x} - \eta \mathbf{d}$ in the hope that a Pareto-stationary point can be reached, where $\eta$ signifies a learning rate. SMGD Liu & Vicente (2021) follows a similar approach, but with full multi-gradients being replaced by stochastic multi-gradients. For both MGD and SMGD, it has been shown that if $\|\boldsymbol{\lambda}^\top \nabla \mathbf{F}(\mathbf{x})\| = 0$ for some $\boldsymbol{\lambda} \in C$, where $C \triangleq \{\mathbf{y} \in [0,1]^S, \sum_{s \in [S]} y_s = 1\}$, then $\mathbf{x}$ is a Pareto stationary solution Fliege et al. (2019); Zhou et al. (2022b).

Next, to define sample complexity in MOO, we first need the following definition of $\epsilon$-stationarity:

**Definition 3** ($\epsilon$-stationarity)**.** *In MOO, a point is $\epsilon$-stationary if the common descent direction* $\mathbf{d}$ *at* $\mathbf{x}$ *satisfies the following condition:* $\mathbb{E}\| \sum_{s \in [S]} \lambda_t^s \nabla f_s(\mathbf{x}_t)\|^2 \leq \epsilon$ *in non-convex MOO problems and* $\mathbb{E}[\sum_{s \in [S]} \lambda_t^s [f_s(\mathbf{x}_t) - f_s(\mathbf{x}_*)]] < \epsilon$ *in strongly-convex MOO problems.*

Table 1: Convergence comparisons between MOO algorithms, where $n$ is the size of dataset; $\epsilon$ is the convergence error. Our proposed algorithms are marked in a shaded background.

| Algorithm | Multi-gradient | Non-convex case | | Strongly-Convex case | |
|---|---|---|---|---|---|
| | | Rate | Samp. Complex. | Rate | Samp. Complex. |
| MGD (Fliege et al., 2019) | Deterministic | $\mathcal{O}\left(\frac{1}{T}\right)$ | $\mathcal{O}\left(n\epsilon^{-1}\right)$ | $\mathcal{O}(\exp(-\mu T))$ | $\mathcal{O}\left(n\ln(\mu/\epsilon)\right)$ |
| SMGD (Yang et al., 2022) | Stochastic | $\mathcal{O}\left(\frac{1}{\sqrt{T}}\right)$ | $\mathcal{O}\left(\epsilon^{-2}\right)$ | $\mathcal{O}\left(\frac{1}{T}\right)$ | $\mathcal{O}\left(\epsilon^{-1}\right)$ |
| MoCo (Fernando et al., 2022) | Stochastic | $\mathcal{O}\left(\frac{1}{\sqrt{T}}\right)$ | $\mathcal{O}\left(\epsilon^{-2}\right)$ | $\mathcal{O}\left(\frac{1}{T}\right)$ | $\mathcal{O}\left(\epsilon^{-1}\right)$ |
| CR-MOGM (Zhou et al., 2022b) | Stochastic | $\mathcal{O}\left(\frac{1}{\sqrt{T}}\right)$ | $\mathcal{O}\left(\epsilon^{-2}\right)$ | $\mathcal{O}\left(\frac{1}{T}\right)$ | $\mathcal{O}\left(\epsilon^{-1}\right)$ |
| STIMULUS/ STIMULUS-M | Stochastic | $\mathcal{O}\left(\frac{1}{T}\right)$ | $\mathcal{O}\left(n+\sqrt{n}\epsilon^{-1}\right)$ | $\mathcal{O}(\exp(-\mu T))$ | $\mathcal{O}\left(n+\sqrt{n}\ln(\mu/\epsilon)\right)$ |
| STIMULUS$^+$ / STIMULUS-M$^+$ | Stochastic | $\mathcal{O}\left(\frac{1}{T}\right)$ | $\mathcal{O}\left(n+\sqrt{n}\epsilon^{-1}\right)$ | $\mathcal{O}(\exp(-\mu T))$ | $\mathcal{O}\left(n+\sqrt{n}\ln(\mu/\epsilon)\right)$ |

We note that the quantity $\|\mathbf{d}\|^2 = \|\boldsymbol{\lambda}^\top \nabla \mathbf{F}(\mathbf{x})\|^2$ can be used as a metric for evaluating the convergence speed of MOO algorithms in the non-convex setting (Fliege et al., 2019; Zhou et al., 2022b; Fernando et al., 2022). On the other hand, in the context of more manageable strongly convex MOO problems, the optimality gap $\sum_{s \in [S]} \lambda_s [f_s(\mathbf{x}) - f_s(\mathbf{x}^*)]$ is usually used as the convergence metric (Liu & Vicente, 2021), where $\mathbf{x}^*$ denotes the Pareto-optimal point.

With the notion of $\epsilon$-stationarity in MOO, we are now in a position to define the concept of sample complexity in MOO as follows:

**Definition 4** (Sample Complexity). *The sample complexity in MOO is defined as the total number of incremental first-order oracle (IFO) calls required by an MOO algorithm to converge to an $\epsilon$-stationary point, where one IFO call evaluates the multi-gradient $\nabla_{\mathbf{x}} f_{sj}(\mathbf{x}; \xi_{sj})$ for all tasks $s$.*

**2) Overview of MOO Algorithms:** As mentioned in Section 1, MOO algorithms can be classified into two primary categories. The first category is usually referred to as gradient-free methods. Typical gradient-free methods include evolutionary MOO algorithms and Bayesian MOO algorithms (Zhang & Li, 2007; Deb et al., 2002; Belakaria et al., 2020; Laumanns & Ocenasek, 2002). These techniques are suitable for small-scale problems but inefficient in solving high-dimensional MOO models (e.g., deep neural networks). In contrast, the second class is gradient-based MOO methods (Fliege & Svaiter, 2000; Désidéri, 2012; Fliege et al., 2019; Peitz & Dellnitz, 2018; Liu & Vicente, 2021), which have shown to be more effective in solving high-dimensional MOO problems. As discussed in Section 1, the most notable gradient-based MOO algorithms include multi-gradient descent (MGD) (Fliege et al., 2019) and stochastic multi-gradient descent (SMGD) (Liu & Vicente, 2021), which achieves $\mathcal{O}(1/T)$ and $\mathcal{O}(1\sqrt{T})$ convergence rates, respectively. Although SMGD is easier to implement in practice thanks to the use of stochastic multi-gradient, it has been shown that the noisy common descent direction in SMGD could potentially cause divergence (cf. the example in Sec. 4 in (Zhou et al., 2022b)). There also have been recent works on using momentum-based methods for bias mitigation in MOO, and these methods are further applied within the context of bi-level optimization problems. (Zhou et al., 2022b; Fernando et al., 2022). Note, however, that the $\mathcal{O}(1/\sqrt{T})$ convergence rates of (Zhou et al., 2022b; Fernando et al., 2022) remain unsatisfactory compared to the $\mathcal{O}(1/T)$ convergence rate of our STIMULUS algorithm family. For easier comparisons, we summarize the state-of-the-art gradient-based MOO algorithms and their convergence rate results under non-convex and strongly convex settings in Table 1.

## 3 THE STIMULUS ALGORITHM FAMILY

In this section, we first present the basic version of the STIMULUS algorithm in Section 3.1, which is followed by the momentum and adaptive-batching variants in Sections 3.2 and 3.3, respectively.

### 3.1 THE STIMULUS ALGORITHM

Our STIMULUS algorithm is presented in Algorithm 1, where we propose a new variance-reduced (VR) multi-gradient estimator. It can be seen from Algorithm 1 that our proposed VR approach has a double-loop structure, where the inner loop is of length $q > 0$. More specifically, different from MGD where a full multi-gradient direction $\mathbf{u}_t^s = \nabla f_s(\mathbf{x}_t), \forall s \in [S]$ is evaluated in all iterations, our

STIMULUS algorithm only evaluates a full multi-gradient every $q$ iterations (i.e., $\mathrm{mod}(t, q) = 0$). For other iterations $t$ with $\mathrm{mod}(t, q) \neq 0$, our STIMULUS algorithm uses a stochastic multi-gradient estimator $\mathbf{u}_t^s$ based on a mini-batch $\mathcal{A}$ with a recursive correction term as follows:

$$\mathbf{u}_t^s = \mathbf{u}_{t-1}^s + \frac{1}{|\mathcal{A}|} \sum_{j \in \mathcal{A}} \left( \nabla f_{sj}(\mathbf{x}_t; \xi_{sj}) - \nabla f_{sj}(\mathbf{x}_{t-1}; \xi_{sj}) \right), \text{ for all } s \in [S]. \quad (2)$$

Eq. (2) shows that the estimator is constructed iteratively based on information from $\mathbf{x}_{t-1}$ and $\mathbf{u}_{t-1}^s$, both of which are obtained from the previous update. We will show later in Section 4 that, thanks to the $q$-periodic full multi-gradients and the recursive correction terms, STIMULUS is able to achieve a convergence rate of $\mathcal{O}(1/T)$. Moreover, due to the stochastic subsampling in mini-batch $\mathcal{A}$, STIMULUS has a lower sample complexity than MGD. In STIMULUS, the update rule for parameters in $\mathbf{x}$ is written as:

$$\mathbf{x}_{t+1} = \mathbf{x}_t - \eta \mathbf{d}_t, \quad (3)$$

where $\eta$ is the learning rate. Here, $\mathbf{d}_t$ is the update direction defined as $\mathbf{d}_t = \sum_{s \in [S]} \lambda_t^s \mathbf{u}_t^s$, where the $\lambda_t^s$-values are obtained by solving the following quadratic optimization problem:

$$\min_{\lambda_t^s \geq 0} \left\| \sum_{s \in [S]} \lambda_t^s \mathbf{u}_t^s \right\|^2, \text{ s.t. } \sum_{s \in [S]} \lambda_t^s = 1. \quad (4)$$

---

**Algorithm 1** STIMULUS algorithm and its variants.

**Require:** Initial point $\mathbf{x}_0$, parameters $T$, $q$.
1: Initialize: Choose $\mathbf{x}_0$.
2: **for** $t = 0, 1, \ldots, T$ **do**
3:    **if** $\mathrm{mod}(t, q) = 0$ **then**
4:       **if** STIMULUS or STIMULUS-M **then**
5:          Compute: $\mathbf{u}_t^s = \frac{1}{n} \sum_{j=1}^n \nabla f_{sj}(\mathbf{x}_t; \xi_{sj}), \forall s \in [S]$
6:       **end if**
7:       **if** STIMULUS$^+$ or STIMULUS-M$^+$ **then**
8:          Compute: $\mathbf{u}_t^s$ as in Eq. (6).
9:       **end if**
10:   **else**
11:      Compute $\mathbf{u}_t^s$ as in Eqs. (2).
12:   **end if**
13:   Optimize:$\min_{\lambda_t^s \geq 0} \| \sum_{s \in [S]} \lambda_t^s \mathbf{u}_t^s \|^2, s.t \sum_{s \in [S]} \lambda_t^s = 1.$
14:   Compute: $\mathbf{d}_t = \sum_{s \in [S]} \lambda_t^s \mathbf{u}_t^s$.
15:   **if** STIMULUS or STIMULUS$^+$ **then**
16:      Update: $\mathbf{x}_{t+1} = \mathbf{x}_t - \eta \mathbf{d}_t$.
17:   **end if**
18:   **if** STIMULUS-M or STIMULUS-M$^+$ **then**
19:      Update: $\mathbf{x}_{t+1} = \mathbf{x}_t + \alpha(\mathbf{x}_t - \mathbf{x}_{t-1}) - \eta \mathbf{d}_t$.
20:   **end if**
21: **end for**

---

The combined iterative update in Eqs. (3) and (4) follows the same token as in the MGDA algorithm in (Mukai, 1980; Sener & Koltun, 2018; Lin et al., 2019; Fliege et al., 2019).

## 3.2 THE MOMENTUM-BASED STIMULUS-M ALGORITHM

Although it can be shown that STIMULUS achieves a theoretical $\mathcal{O}(1/T)$ convergence rate, it could be sensitive to the choice of learning rate and suffer from similar oscillation issues in practice as gradient-descent-type methods do in single-objective optimization when some objectives are ill-conditioned.

To further improve the empirical performance of STIMULUS, we now propose a momentum-assisted enhancement for STIMULUS called STIMULUS-M. The rationale behind STIMULUS-M is to take into account the past trajectories to smooth the update direction. Specifically, in addition to the same combined iterative update as in Eqs. (3) and (4), the update rule in STIMULUS-M incorporates an $\alpha$-parameterized momentum term as follows:

$$\mathbf{x}_{t+1} = \mathbf{x}_t - \eta \mathbf{d}_t + \underbrace{\alpha(\mathbf{x}_t - \mathbf{x}_{t-1})}_{\text{Momentum}}, \quad \forall s \in [S], \quad (5)$$

where $\alpha \in (0, 1)$ is the momentum coefficient.

## 3.3 THE ADAPTIVE-BATCHING-BASED STIMULUS$^+$ /STIMULUS-M$^+$ ALGORITHMS

Note that in both STIMULUS and STIMULUS-M, one still needs to evaluate a full multi-gradient every $q$ iterations, which remains computationally demanding in the large data regime. Moreover, if the objectives are in an expectation or "online" form rather than the finite-sum setting, it is infeasible to compute a full multi-gradient. To address these limitations, we propose two *adaptive-batching* enhanced versions for STIMULUS and STIMULUS-M called STIMULUS$^+$ and STIMULUS-M$^+$,

respectively. Specifically, rather than using a $q$-periodic full multi-gradient $\mathbf{u}_t^s = \nabla f_s(\mathbf{x}_t) = \frac{1}{n}\sum_{j=1}^{n}\nabla f_{sj}(\mathbf{x}_t; \xi_{sj})$, $\forall s \in [S]$, in iteration $t$ with $\mathrm{mod}(t, q) = 0$, we utilize an adaptive-batching stochastic multi-gradient as follows:

$$\mathbf{u}_t^s = \frac{1}{|\mathcal{N}_s|}\sum_{j\in\mathcal{N}_s}\nabla f_{sj}(\mathbf{x}_t; \xi_{sj}), \quad \forall s \in [S], \tag{6}$$

where $\mathcal{N}_s$ is an $\epsilon$-adaptive batch sampled from the dataset uniformly at random with size:

$$|\mathcal{N}_s| = \min\left\{ c_\gamma \sigma^2 \gamma_t^{-1}, c_\epsilon \sigma^2 \epsilon^{-1}, n \right\}. \tag{7}$$

We choose constants $c_\gamma \geq 8$, $c_\epsilon \geq \eta$ in non-convex case and $c_\gamma \geq \frac{8\mu}{\eta}$, $c_\epsilon \geq \frac{\mu}{2}$ in strongly-convex case (see detailed discussions in Section 4). The $\sigma^2$, represents variance bound of stochastic gradient norms (cf. Assumption. 2). In STIMULUS$^+$ , we choose $\gamma_{t+1} = \sum_{i=(n_k-1)q}^{t} \frac{\|\mathbf{d}_i\|^2}{q}$, while in the momentum based algorithm STIMULUS-M$^+$, we choose $\gamma_{t+1} = \sum_{i=(n_k-1)q}^{t}\|\alpha^{(t-i)}\mathbf{d}_i\|^2/q$. The term $\gamma_{t+1}$ offers further refinement to improve convergence.

## 4 PARETO STATIONARITY CONVERGENCE ANALYSIS

In this section, we will theoretically analyze the Pareto stationarity convergence performance of our STIMULUS algorithm family under both non-convex and strongly convex settings. Before presenting our Pareto stationarity convergence results for our STIMULUS algorithms, we first state two needed assumptions, which are conventional in the optimization literature:

**Assumption 1** (*L*-Lipschitz Smoothness). *There exists a constant $L > 0$ such that $\|\nabla f_s(\mathbf{x}) - \nabla f_s(\mathbf{y})\| \leq L\|\mathbf{x} - \mathbf{y}\|, \forall \mathbf{x}, \mathbf{y} \in \mathbb{R}^d, \forall s \in [S]$.*

**Assumption 2** (Bounded Variance of Stochastic Gradient Norms). *There exists a constant $\sigma > 0$ such that for all $\mathbf{x} \in \mathbb{R}^d$, $\|\nabla_\mathbf{x} f_s(\mathbf{x}; \xi) - \nabla_\mathbf{x} f_s(\mathbf{x})\|^2 \leq \sigma^2, \forall s \in S$.*

With the assumptions above, we are now in a position to discuss the Pareto stationary convergence of the STIMULUS family.

### 4.1 PARETO-STATIONARITY CONVERGENCE OF STIMULUS

**1) STIMULUS: The Non-convex Setting.** First, we show that the basic STIMULUS algorithm achieves an $\mathcal{O}(1/T)$ convergence rate for non-convex MOO problems as follows. Note that this result matches that of the deterministic MGD method.

**Theorem 1** (STIMULUS for Non-convex MOO). *Let $\eta_t = \eta \leq \min\{\frac{1}{2L}, \frac{1}{2}\}$. Under Assumption 1, if at least one objective function $f_s(\cdot)$, $s \in [S]$ is bounded from below by $f_s^{\min}$, then the sequence $\{\mathbf{x}_t\}$ output by STIMULUS satisfies: $\frac{1}{T}\sum_{t=0}^{T-1}\mathbb{E}\|\sum_{s\in[S]}\lambda_t^s\nabla f_s(\mathbf{x}_t)\|^2 = \mathcal{O}(1/T)$.*

Following from Theorem. 1, we immediately have the following sample complexity for the STIMULUS algorithm by choosing $q = |\mathcal{A}| = \lceil\sqrt{n}\rceil$:

**Corollary 1** (Sample Complexity of STIMULUS for Non-convex MOO). *By choosing $\eta \leq \min\{\frac{1}{2L}, \frac{1}{2}\}$, $q = |\mathcal{A}| = \lceil\sqrt{n}\rceil$, the overall sample complexity of STIMULUS for finding an $\epsilon$-stationary point for non-convex MOO problems is $\mathcal{O}\left(\sqrt{n}\epsilon^{-1} + n\right)$.*

Several interesting remarks regarding Theorem 1 and Corollary 1 are in order: **1)** Our proof of STIMULUS's Pareto-stationarity convergence only relies on standard assumptions commonly used in first-order optimization techniques. This is in stark contrast to prior research, where unconventional and hard-to-verify assumptions were required (e.g., an assumption on the convergence of $\mathbf{x}$-sequence is used in Fliege et al. (2019)). **2)** While both MGD and our STIMULUS methods share the same $\mathcal{O}(1/T)$ convergence rate, STIMULUS enjoys a substantially lower sample complexity than MGD. More specifically, the sample complexity of STIMULUS is reduced by a factor of $\sqrt{n}$ when compared to MGD. This becomes particularly advantageous in the "big data" regime where $n$ is large.

**2) STIMULUS: The Strongly Convex Setting.** Now, we consider the strongly convex setting, which is more tractable but still of interest in many learning problems in practice (e.g., multi-objective ridge regression). In the strongly convex setting, we have the following additional assumption:

**Assumption 3** ($\mu$-Strongly Convex Function). *Each objective $f_s(\mathbf{x})$, $s \in [S]$ is a $\mu$-strongly convex function, i.e., $f_s(\mathbf{y}) \geq f_s(\mathbf{x}) + \nabla f_s(\mathbf{x})(\mathbf{y} - \mathbf{x}) + \frac{\mu}{2}\|\mathbf{y} - \mathbf{x}\|^2, \forall \mathbf{x}, \mathbf{y}$, for some $\mu > 0$.*

For strongly convex MOO problems, the next result says that STIMULUS achieves a much stronger expected linear Pareto-optimality convergence performance:

**Theorem 2** (STIMULUS for $\mu$-Strongly Convex MOO). *Let $\eta \leq \min\{\frac{1}{2}, \frac{1}{2\mu}, \frac{1}{8L}, \frac{\mu}{64L^2}\}, q = |\mathcal{A}| = \lceil\sqrt{n}\rceil$. Under Assumptions 1–3, pick $\mathbf{x}_t$ as the final output of STIMULUS with probability $w_t = (1 - \frac{3\mu\eta}{4})^{1-t}$. Then, we have $\mathbb{E}_t[\sum_{s \in [S]} \lambda_t^s [f_s(\mathbf{x}_t) - f_s(\mathbf{x}_*)]] \leq \|\mathbf{x}_0 - \mathbf{x}_*\|^2 \mu \exp(-\frac{3\eta\mu T}{4})$.*

Further, Theorem 2 immediately implies following with logarithmic sample complexity (in terms of $\epsilon$) STIMULUS with a proper choice of learning rate and $q = |\mathcal{A}| = \lceil\sqrt{n}\rceil$.

**Corollary 2** (Sample Complexity of STIMULUS for Solving $\mu$-Strongly Convex MOO). *By choosing $\eta \leq \min\{\frac{1}{2}, \frac{1}{2\mu}, \frac{1}{8L}, \frac{\mu}{64L^2}\}, q = |\mathcal{A}| = \lceil\sqrt{n}\rceil\}$, the overall sample complexity of STIMULUS for solving strongly convex MOO is $\mathcal{O}(n + \sqrt{n}\ln(\mu/\epsilon))$.*

There are also several interesting insights from Theorem 2 and Corollary 2 regarding STIMULUS's performance for solving strongly convex MOO problems: **1)** STIMULUS achieves an expected linear convergence rate of $\mathcal{O}(\mu\exp(-\mu T))$. Interestingly, this convergence rate matches that of MGD for strongly convex MOO problems as well as gradient descent for strongly convex single-objective optimization. **2)** Another interesting feature of STIMULUS for strongly convex MOO stems from its use of randomly selected outputs $\mathbf{x}_t$ along with associated weights $w_t$ from the trajectory of $\mathbf{x}_t$, which is inspired by the similar idea for stochastic gradient descent (SGD) (Ghadimi & Lan, 2013). Note that, for implementation in practice, one does not need to store all $\mathbf{x}_t$-values. Instead, the algorithm can be implemented by using a random clock for stopping (Ghadimi & Lan, 2013).

## 4.2 PARETO STATIONARITY CONVERGENCE OF STIMULUS-M

Next, we turn our attention to the Pareto stationarity convergence of the STIMULUS-M algorithm. Again, we analyze STIMULUS-M for non-convex and strongly convex MOO problems:

**Theorem 3** (STIMULUS-M for Non-convex MOO). *Let $\eta_t = \eta \leq \min\{\frac{1}{2L}, \frac{1}{2}\}, q = |\mathcal{A}| = \lceil\sqrt{n}\rceil$. Under Assumptions 1, if at least one objective function $f_s(\cdot)$, $s \in [S]$, is bounded from below by $f_s^{\min}$, then the sequence $\{\mathbf{x}_t\}$ output by STIMULUS-M satisfies $\frac{1}{T}\sum_{t=0}^{T-1} \mathbb{E}\|\sum_{s \in [S]} \lambda_t^s \nabla f_s(\mathbf{x}_t)\|^2 = \mathcal{O}(\frac{1}{T})$.*

Similar to the basic STIMULUS algorithm, by choosing the appropriate learning rate and inner loop length parameters, we immediately have the following sample complexity result for STIMULUS-M for solving non-convex MOO problems:

**Corollary 3** (Sample Complexity of STIMULUS-M for Solving Non-convex MOO). *By choosing $\eta_t = \eta \leq \min\{\frac{1}{2L}, \frac{1}{2}\}, q = |\mathcal{A}| = \lceil\sqrt{n}\rceil$. The overall sample complexity of STIMULUS-M under non-convex objective functions is $\mathcal{O}\left(\sqrt{n}\epsilon^{-1} + n\right)$.*

The next two results state the Pareto optimality and sample complexity results for STIMULUS-M:

**Theorem 4** (STIMULUS-M for $\mu$-Strongly Convex MOO). *Let $\eta \leq \min\{\frac{1}{2}, \frac{1}{2\mu}, \frac{1}{8L}, \frac{\mu}{64L^2}\}, q = |\mathcal{A}| = \lceil\sqrt{n}\rceil$. Under Assumptions 1–3, pick $\mathbf{x}_t$ as the final output of STIMULUS-M with probability $w_t = (1 - \frac{3\mu\eta}{4})^{1-t}$. Then, we have $\mathbb{E}_t[\sum_{s \in [S]} \lambda_t^s [f_s(\mathbf{x}_t) - f_s(\mathbf{x}_*)]] \leq \|\mathbf{x}_0 - \mathbf{x}_*\|^2 \mu \exp(-\frac{3\eta\mu T}{4})$.*

**Corollary 4** (Sample Complexity of STIMULUS-M for Solving Strongly Convex MOO). *By choosing $\eta \leq \min\{\frac{1}{2}, \frac{1}{2\mu}, \frac{1}{8L}, \frac{\mu}{64L^2}\}, q = |\mathcal{A}| = \lceil\sqrt{n}\rceil$, the overall sample complexity of STIMULUS-M for solving strongly convex MOO is $\mathcal{O}(n + \sqrt{n}\ln(\mu/\epsilon))$.*

We remark that the convergence rate upper bound of STIMULUS-M is the same as that in Theorem 2, which suggests a potentially loose convergence upper bound in Theorem 4 due to the technicality and intricacies in analyzing momentum-based stochastic multi-gradient algorithms for solving non-convex MOO problems. Yet, we note that even this potentially loose convergence rate upper bound in Theorem 4 already suffices to establish a linear convergence rate for STIMULUS-M in solving strongly convex MOO problems. Moreover, we will show later in Section 5 that this momentum-assisted method significantly accelerates the empirical convergence speed performance. It is also

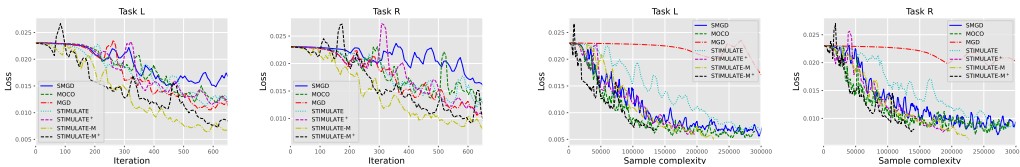

(a) Training loss convergence in terms of iterations.    (b) Training loss convergence in terms of samples.

Figure 1: Training loss convergence comparisons between different MOO algorithms.

worth noting that there are two key differences in the proofs of Theorem 3 and 4 compared to those of the momentum-based stochastic gradient algorithm for single-objective non-convex optimization: 1) our proof exploits the martingale structure of the $\mathbf{u}_t^s$. This enables us to tightly bound the mean-square error term $\mathbb{E} \|\nabla f_s(\mathbf{x}_t) - \mathbf{u}_t^s\|^2$ under the momentum scheme. In contrast, in the traditional analysis of stochastic algorithms with momentum, this error term corresponds to the variance of the stochastic estimator and is typically assumed to be bounded by a universal constant. 2) Our proof requires careful manipulation of the bounding strategy to effectively handle the accumulation of the mean-square error term $\mathbb{E} \|\nabla f_s(\mathbf{x}_k) - \mathbf{u}_t^s\|^2$ over the entire optimization trajectory in non-convex MOO.

### 4.3 PARETO STATIONARITY CONVERGENCE OF STIMULUS$^+$ /STIMULUS-M$^+$

Next, we present the Pareto stationarity convergence and the associated sample complexity results of the STIMULUS$^+$ /STIMULUS-M$^+$ algorithms for non-convex MOO as follows:

**Theorem 5** (STIMULUS$^+$ /STIMULUS-M$^+$ for Non-convex MOO). *Let $\eta \leq \min\{\frac{1}{4L}, \frac{1}{2}\}, q = |\mathcal{A}| = \lceil \sqrt{n} \rceil$. By choosing $c_\gamma$ and $c_\epsilon$ as such that $c_\gamma \geq 8$, and $c_\epsilon \geq \eta$, under Assumptions 1 and 2, if at least one function $f_s(\cdot)$, $s \in [S]$ is bounded from below by $f_s^{\min}$, then the sequence $\{\mathbf{x}_t\}$ output by STIMULUS$^+$ /STIMULUS-M$^+$ satisfies: $\frac{1}{T} \sum_{t=0}^{T-1} \mathbb{E}\| \sum_{s\in[S]} \lambda_t^s \nabla f_s(\mathbf{x}_t)\|^2 = \mathcal{O}(\frac{1}{T})$.*

**Corollary 5** (Sample Complexity of STIMULUS$^+$ /STIMULUS-M$^+$ for Non-convex MOO). *By choosing $\eta \leq \min\{\frac{1}{4L}, \frac{1}{2}\}, q = |\mathcal{A}| = \lceil \sqrt{n} \rceil$, $c_\gamma \geq 8$, and $c_\epsilon \geq \eta$. The overall sample complexity of STIMULUS$^+$ /STIMULUS-M$^+$ under non-convex objective functions is $\mathcal{O}\left(\sqrt{n}\epsilon^{-1} + n\right)$.*

**Theorem 6** (STIMULUS$^+$ /STIMULUS-M$^+$ for $\mu$-Strongly Convex case.). *Let $\eta \leq \min\{\frac{1}{2}, \frac{1}{2\mu}, \frac{1}{8L}, \frac{\mu}{64L^2}\}, c_\gamma \geq \frac{8\mu}{\eta}, c_\epsilon \geq \frac{\mu}{2}, q = |\mathcal{A}| = \lceil \sqrt{n} \rceil$. Under Assumptions 1- 3, pick $\mathbf{x}_t$ as the final output of the STIMULUS$^+$ /STIMULUS-M$^+$ algorithm with weights $w_t = (1 - \frac{3\mu\eta}{4})^{1-t}$. Then, it holds that $\mathbb{E}_t[\sum_{s\in[S]} \lambda_t^s [f_s(\mathbf{x}_t) - f_s(\mathbf{x}_*)]] \leq \|\mathbf{x}_0 - \mathbf{x}_*\|^2 \mu \exp(-\frac{3\eta\mu T}{4})$.*

**Corollary 6** (Sample Complexity of STIMULUS$^+$ /STIMULUS-M$^+$ for Strongly Convex MOO). *By choosing $\eta \leq \min\{\frac{1}{2}, \frac{1}{2\mu}, \frac{1}{8L}, \frac{\mu}{64L^2}\}, c_\gamma \geq \frac{8\mu}{\eta}, c_\epsilon \geq \frac{\mu}{2}, q = |\mathcal{A}| = \lceil \sqrt{n} \rceil$, the overall sample complexity of STIMULUS$^+$ /STIMULUS-M$^+$ for solving strongly MOO is $\mathcal{O}\left(n + \sqrt{n}\ln(\mu/\epsilon)\right)$.*

We note that, although the theoretical sample complexity bounds of STIMULUS$^+$ /STIMULUS-M$^+$ are the same as those of STIMULUS/STIMULUS-M, respectively, the fact that STIMULUS$^+$ and STIMULUS-M$^+$ do not need full multi-gradient evaluations implies that STIMULUS/STIMULUS-M use significantly fewer samples than STIMULUS/STIMULUS-M in the large dataset regime. Our experimental results in the next section will also empirically confirm this.

## 5 EXPERIMENTAL RESULTS

In this section, we conduct numerical experiments to verify the efficacy of our STIMULUS algorithm family. Due to space limitation, we only present experiments for non-convex MOO problems and relegate experimental results for strongly convex MOO problems in the appendix.

**1) Two-Objective Experiments on the MultiMNIST Dataset:** First, we test the convergence performance of our STIMULUS algorithms using the "MultiMNIST" dataset (Sabour et al., 2017), which is a multi-task learning version of the MNIST dataset (LeCun et al., 1998) from LIBSVM repository. Specifically, MultiMNIST converts the hand-written classification problem in MNIST into a two-task problem, where the two tasks are task "L" (to categorize the top-left digit) and task "R" (to classify the bottom-right digit). The goal is to classify the images of different tasks.

We compare our STIMULUS algorithms with MGD, SMGD, and MOCO. All algorithms use the same randomly generated initial point. The learning rates are chosen as $\eta = 0.3, \alpha = 0.5$, constant $c = 32$ and solution accuracy $\epsilon = 10^{-3}$. The batch-size for MOCO and SMGD is 96. The full batch size for MGD is 1024, and the inner loop batch-size $|\mathcal{N}_s|$ for STIMULUS, STIMULUS-M, STIMULUS$^+$, STIMULUS-M$^+$is 96. As shown in Fig. 1(a), SMGD exhibits the slowest convergence speed, while MOCO has a slightly faster convergence. MGD and our STIMULUS algorithms have comparable performances. The STIMULUS-M /STIMULUS-M$^+$ algorithms converge faster than MGD, STIMULUS , and STIMULUS$^+$ , primarily due to the use of momentum. Fig. 1(b) highlights differences in sample complexity. MGD suffers the highest sample complexity, while STIMULUS$^+$ and STIMULUS-M$^+$ demonstrate a more efficient utilization of samples in comparison to STIMULUS and STIMULUS-M. These results are consistent with our theoretical analyses as outlined in Theorems 1, 3, and 5.

**2) Eight-Objective Experiments on the River Flow Dataset:**
We further test our algorithms on an 8-task problem with the river flow dataset (Nie et al., 2017), which is for flow prediction at eight locations in the Mississippi river network. In this experiment, we set $\eta = 0.001, \alpha = 0.1$, the batch size for MOCO and SMGD is 8, the full batch size for MGD is 128, and the inner loop batch size $|\mathcal{N}_s|$ for STIMULUS, STIMULUS-M, STIMULUS$^+$ , STIMULUS-M$^+$is eight. To better visualize different tasks, we plot the normalized loss in a radar chart as shown in Fig. 2, where we can see that our STIMULUS algorithms achieve a much smaller footprint, which is desirable. Furthermore, we compare the sample complexity results of all algorithms in Table 3 (relegated to the

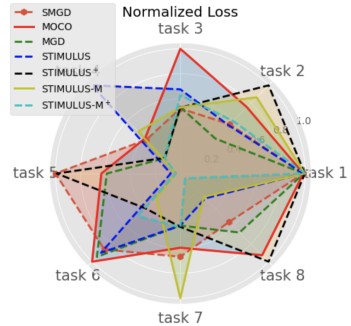

Figure 2: Training loss convergence comparison (8-objective).

appendix due to space limitation), which reveals a significant reduction in sample utilization by STIMULUS$^+$ /STIMULUS-M$^+$ compared to MGD, while achieving better loss compared to SGMD and MOCO (cf. Fig. 2).

**3) 40-Objective Experiments with the CelebA Dataset:**
Lastly, we conduct large-scale 40-objective experiments with the CelebA dataset (Liu et al., 2015), which contains 200K facial images annotated with 40 attributes. Each attribute corresponds to a binary classification task, resulting in a 40-objective problem. We use a ResNet-18 He et al. (2016) model without the final layer for each attribute, and we attach a linear layer to each attribute for classification. In this experiment, we set $\eta = 0.0005, \alpha = 0.01$, the full batch size for MGD is 1024, and the batch size for SMGD and MOCO and the inner loop batch size $|\mathcal{N}_s|$ for STIMULUS, STIMULUS-M,

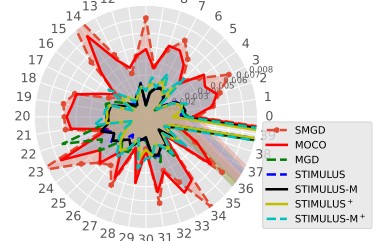

Figure 3: Training loss convergence comparison (40-task).

STIMULUS$^+$ , STIMULUS-M$^+$is 32. As shown in Fig. 3, MGD, STIMULUS, STIMULUS-M, STIMULUS$^+$ , and STIMULUS-M$^+$significantly outperform SMGD and MOCO in terms of training loss. Also, we note that STIMULUS$^+$ and STIMULUS-M$^+$ consume fewer sample (approximately 11,000) samples compared to STIMULUS and STIMULUS-M , which consume approximately 13,120 samples, and MGD, which consumes approximately 102,400 samples. These results are consistent with our theoretical results in Theorems 1, 3, and 5.

## 6 CONCLUSION

In this paper, we proposed STIMULUS, a new variance-reduction-based stochastic multi-gradient-based algorithm to achieve fast convergence and low sample complexity multi-objective optimization (MOO). We theoretically analyzed the Pareto stationarity convergence and sample complexity of our proposed STIMULUS algorithms under non-convex and strongly convex settings. To further enhance the empirical Pareto stationarity convergence of STIMULUS, we further proposed an algorithms called STIMULUS-M , which incorporates momentum information to expedite convergence. To alleviate the periodic full multi-gradient evaluation requirement in STIMULUS and STIMULUS-M, we further proposed enhanced versions for both algorithms with adaptive batching called STIMULUS$^+$ /STIMULUS-M$^+$. We provided theoretical performance analysis for all enhanced algorithms. Collectively, our proposed STIMULUS algorithm family advances the state of the art of MOO algorithm design and analysis.

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
