## A    PROOF OF CONVERGENCE OF VR-MOO

Table 2: List of key notation.

| Notation | Definition |
| --- | --- |
| $n$ | Total number of samples per task |
| $s$ | Objective/task index |
| $S$ | Total number of Objectives/tasks |
| $t$ | Iteration number index |
| $T$ | Total number of iterations |
| $\mathbf{x} \in \mathbb{R}^d$ | Model parameters in Problem (1) |
| $\mathbf{x}_* \in \mathbb{R}^d$ | A pareto optimal solution of Problem (1) |
| $\eta$ | The learning rate |
| $\alpha$ | The momentum constant |

For clarity of notation, we drop $*$ for $\lambda$, that is, we use $\lambda_t^s$ to represent the solution of quadratic problem for task $s$ in the $t$-th round.

**Lemma 1.** *Let Assumption 1 hold. The gradient estimator $\mathbf{u}_t^s$ satisfies for all $(n_t - 1)q + 1 \leq t \leq n_t q - 1$:*

$$\mathbb{E}_t \|\nabla f_s(\mathbf{x}_t) - \mathbf{u}_t^s\|^2 \leq \frac{L^2}{|\mathcal{A}|} \sum_{i=(n_t-1)q}^{t} \mathbb{E}\|\mathbf{x}_{i+1} - \mathbf{x}_i\|^2 + \mathbb{E}_t \|\nabla f_s(\mathbf{x}_{(n_t-1)q}) - \mathbf{u}_{(n_t-1)q}^s\|^2. \quad (8)$$

**Proof of Lemma. 1.**

*Proof.* From Lemma 1 in Fang et al. (2018), we have

$$\mathbb{E}_t \|\nabla f_s(\mathbf{x}_t) - \mathbf{u}_t^s\|^2 \overset{(a)}{=} \mathbb{E}_t \|\nabla f_s(\mathbf{x}_{t-1}) - \mathbf{u}_{t-1}^s\|^2$$
$$+ \mathbb{E}_t \|\frac{1}{|\mathcal{A}|} \sum_{j \in \mathcal{A}} \left( \nabla f_{sj}(\mathbf{x}_t; \xi_{sj}) - \nabla f_{sj}(\mathbf{x}_{t-1}; \xi_{sj}) + \nabla f_s(\mathbf{x}_{t-1}) - \nabla f_s(\mathbf{x}_t) \right) \|^2$$
$$\overset{(b)}{\leq} \mathbb{E}_t \|\nabla f_s(\mathbf{x}_{(n_t-1)q}) - \mathbf{u}_{(n_t-1)q}^s\|^2 + L^2 \sum_{i=(n_t-1)q}^{t} \frac{1}{|\mathcal{A}|} \mathbb{E}\|\mathbf{x}_{i+1} - \mathbf{x}_i\|^2. \quad (9)$$

$(a)$ stems from Proposition 1 in Fang et al. (2018), where the expectation of the gradient difference is broken down. $(b)$ leverages Eq. (2.3) from Fang et al. (2018), applying a bound based on the Lipschitz continuity of the gradient.

Telescoping over from $(n_t - 1)q + 1$ to $t$, where $t \leq n_t q - 1$, we obtain that

$$\mathbb{E}_t \|\nabla f_s(\mathbf{x}_t) - \mathbf{u}_t^s\|^2 \leq \mathbb{E}_t \|\nabla f_s(\mathbf{x}_{(n_t-1)q}) - \mathbf{u}_{(n_t-1)q}^s\|^2 + L^2 \sum_{i=(n_t-1)q}^{t} \frac{1}{|\mathcal{A}|} \mathbb{E}\|\mathbf{x}_{i+1} - \mathbf{x}_i\|^2 \quad (10)$$

Then, we have

$$\mathbb{E}_t \|\nabla f_s(\mathbf{x}_t) - \mathbf{u}_t^s\|^2 \leq \frac{L^2}{|\mathcal{A}|} \sum_{i=(n_t-1)q}^{t} \mathbb{E}\|\mathbf{x}_{i+1} - \mathbf{x}_i\|^2 + \mathbb{E}_t \|\nabla f_s(\mathbf{x}_{(n_t-1)q}) - \mathbf{u}_{(n_t-1)q}^s\|^2. \quad (11)$$

$\square$

**Lemma 2.** *For general L-smooth functions $\{f_s, s \in [S]\}$, choose the learning rate $\eta$ s.t. $\eta \leq \frac{1}{2}$, the update $d_t$ of the algorithm satisfies:*

$$f_s(\mathbf{x}_{t+1}) \leq f_s(\mathbf{x}_t) + \frac{\eta}{2} \|\nabla f_s(\mathbf{x}_t) - \mathbf{u}_t^s\|^2 - \frac{\eta}{4} \|\mathbf{d}_t\|^2. \quad (12)$$

**Proof of Lemma. 2.**

*Proof.*

$$f_s(\mathbf{x}_{t+1}) \overset{(a)}{\leq} f_s(\mathbf{x}_t) + \langle \nabla f_s(\mathbf{x}_t), -\eta \mathbf{d}_t \rangle + \frac{1}{2} L \|\eta \mathbf{d}_t\|^2$$

$$= f_s(\mathbf{x}_t) - \eta \langle \nabla f_s(\mathbf{x}_t) - \mathbf{u}_t^s, \mathbf{d}_t \rangle - \eta \langle \mathbf{u}_t^s, \mathbf{d}_t \rangle + \frac{1}{2} L \|\eta \mathbf{d}_t\|^2$$

$$\overset{(b)}{\leq} f_s(\mathbf{x}_t) - \eta \langle \nabla f_s(\mathbf{x}_t) - \mathbf{u}_t^s, \mathbf{d}_t \rangle - \eta \|\mathbf{d}_t\|^2 + \frac{1}{2} L \|\eta \mathbf{d}_t\|^2$$

$$\overset{(c)}{\leq} f_s(\mathbf{x}_t) + \frac{\eta}{2} \|\nabla f_s(\mathbf{x}_t) - \mathbf{u}_t^s\|^2 + \frac{1}{2} \eta \|\mathbf{d}_t\|^2 - \eta \|\mathbf{d}_t\|^2 + \frac{1}{2} L \eta^2 \|\mathbf{d}_t\|^2$$

$$= f_s(\mathbf{x}_t) + \frac{\eta}{2} \|\nabla f_s(\mathbf{x}_t) - \mathbf{u}_t^s\|^2 - \eta \left( \frac{1}{2} - \frac{1}{2} L \eta \right) \|\mathbf{d}_t\|^2. \tag{13}$$

$(a)$ follows from the objective function $f_s$ is $L$-smooth. $(b)$ follows from $\langle \mathbf{u}_t^s, \mathbf{d}_t \rangle \geq \|\mathbf{d}_t\|^2$ since $\mathbf{d}_t$ is a general solution in the convex hull of the family of vectors $\{\mathbf{u}_t^s, s \in [S]\}$ (see Lemma 2.1 Désidéri (2012)). $(c)$ follows from the triangle inequality.

By setting $\left( \frac{1}{2} - \frac{1}{2}\eta \right) \geq \frac{1}{4}$, that is, $\eta \leq \frac{1}{2}$, we have

$$f_s(\mathbf{x}_{t+1}) \leq f_s(\mathbf{x}_t) + \frac{\eta}{2} \|\nabla f_s(\mathbf{x}_t) - \mathbf{u}_t^s\|^2 - \frac{\eta}{4} \|\mathbf{d}_t\|^2. \tag{14}$$

$\square$

**Proof of Theorem. 1**

*Proof.* Taking expectation on both sides of the inequality in Lemma. 2, we have

$$\mathbb{E}[f_s(\mathbf{x}_{t+1})] \overset{(a)}{\leq} \mathbb{E}[f_s(\mathbf{x}_t)] + \frac{\eta}{2} \mathbb{E}\|\nabla f_s(\mathbf{x}_t) - \mathbf{u}_t^s\|^2 - \frac{\eta}{4} \mathbb{E}\|\mathbf{d}_t\|^2$$

$$\overset{(b)}{\leq} \mathbb{E}[f_s(\mathbf{x}_t)] - \frac{\eta}{4} \mathbb{E}\|\mathbf{d}_t\|^2 + \mathbb{E}\frac{\eta}{2}[\frac{L^2}{|\mathcal{A}|} \sum_{i=(n_t-1)q}^{t} \mathbb{E}\|\mathbf{x}_{i+1} - \mathbf{x}_i\|^2 + \mathbb{E}\|\nabla f_s(\mathbf{x}_{(n_t-1)q}) - \mathbf{u}_{(n_t-1)q}^s\|^2]$$

$$\overset{(c)}{=} \mathbb{E}[f_s(\mathbf{x}_t)] - \frac{\eta}{4} \mathbb{E}\|\mathbf{d}_t\|^2 + \frac{\eta}{2}[\frac{L^2}{|\mathcal{A}|} \sum_{i=(n_t-1)q}^{t} \eta^2 \mathbb{E}\|\mathbf{d}_i\|^2]. \tag{15}$$

$(a)$ follows from Lemma. 2. $(b)$ follows from the Lemma. 1. $(c)$ follows from the update rule of $\mathbf{x}$ as shown in Eq. (6) and $\mathbb{E}\|\nabla f_s(\mathbf{x}_{(n_t-1)q}) - \mathbf{u}_{(n_t-1)q}^s\|^2 = 0$ as shown in Line 5 in our Algorithm. 1.

Next, telescoping the above inequality over $t$ from $(n_t - 1)q$ to $t$ where $t \leq n_t q - 1$ and noting that for $(n_t - 1)q \leq j \leq n_t q - 1, n_j = n_t$, we obtain

$$\mathbb{E}[f_s(\mathbf{x}_{t+1})]$$

$$\leq \mathbb{E}[f_s(\mathbf{x}_{(n_t-1)q})] - \frac{\eta}{4} \sum_{j=(n_t-1)q}^{t} \mathbb{E}\|\mathbf{d}_j\|^2 + \frac{\eta}{2}[\frac{L^2}{|\mathcal{A}|} \sum_{j=(n_t-1)q}^{t} \sum_{i=(n_t-1)q}^{j} \eta^2 \mathbb{E}\|\mathbf{d}_i\|^2]$$

$$\overset{(a)}{\leq} \mathbb{E}[f_s(\mathbf{x}_{(n_t-1)q})] - \frac{\eta}{4} \sum_{j=(n_t-1)q}^{t} \mathbb{E}\|\mathbf{d}_j\|^2 + \frac{\eta}{2}[\frac{L^2}{|\mathcal{A}|} \sum_{j=(n_t-1)q}^{t} \sum_{i=(n_t-1)q}^{t} \eta^2 \mathbb{E}\|\mathbf{d}_i\|^2]$$

$$\overset{(b)}{\leq} \mathbb{E}[f_s(\mathbf{x}_{(n_t-1)q})] - \frac{\eta}{4} \sum_{j=(n_t-1)q}^{t} \mathbb{E}\|\mathbf{d}_j\|^2 + \frac{\eta^3 q}{2}[\frac{L^2}{|\mathcal{A}|} \sum_{j=(n_t-1)q}^{t} \mathbb{E}\|\mathbf{d}_j\|^2]$$

$$= \mathbb{E}[f_s(\mathbf{x}_{(n_t-1)q})] - [\frac{\eta}{4} - \frac{\eta^3 q}{2} \frac{L^2}{|\mathcal{A}|}] \sum_{j=(n_t-1)q}^{t} \mathbb{E}\|\mathbf{d}_j\|^2. \tag{16}$$

where $(a)$ extends the summation of the third term from $j$ to $t$, $(b)$ follows from the fact that $t \leq n_t q - 1$.

We continue the proof by further driving

$$\mathbb{E}[f_s(\mathbf{x}_T)] - \mathbb{E}[f_s(\mathbf{x}_0)]$$
$$= (\mathbb{E}[f_s(\mathbf{x}_q)] - \mathbb{E}[f_s(\mathbf{x}_0)]) + (\mathbb{E}[f_s(\mathbf{x}_{2q})] - \mathbb{E}[f_s(\mathbf{x}_q)]) + \cdot + (\mathbb{E}[f_s(\mathbf{x}_T)] - \mathbb{E}[f_s(\mathbf{x}_{(n_T-1)q})])$$
$$\leq -[\frac{\eta}{4} - \frac{\eta^3 q}{2} \frac{L^2}{|\mathcal{A}|}] \sum_{t=0}^{T-1} \mathbb{E}\|\mathbf{d}_t\|^2 \tag{17}$$

Note that $\mathbb{E}[f_s(\mathbf{x}_{T+1})] \geq f_s^* \triangleq \inf_{\mathbf{x} \in \mathbb{R}^d} f_s(\mathbf{x})$. Hence, we have

$$[\frac{\eta}{4} - \frac{\eta^3 q}{2} \frac{L^2}{|\mathcal{A}|}] \sum_{t=0}^{T-1} \mathbb{E}\|\mathbf{d}_t\|^2 \leq [[f_s(\mathbf{x}_0)] - [f_s(\mathbf{x}_T)]] \leq [[f_s(\mathbf{x}_0)] - f_s^*]. \tag{18}$$

Based on the parameter setting $q = |\mathcal{A}| = \lceil \sqrt{n} \rceil$, we have

$$[\frac{\eta}{4} - \frac{\eta^3 L^2}{2}] \sum_{t=0}^{T-1} \|\mathbf{d}_t\|^2 \leq [[f_s(\mathbf{x}_0)] - f_s^*]. \tag{19}$$

Thus, we have

$$\frac{1}{T} \sum_{t=0}^{T-1} \mathbb{E}\|\mathbf{d}_t\|^2 \leq \frac{[[f_s(\mathbf{x}_0)] - f_s^*]}{[\frac{\eta}{4} - \frac{\eta^3 L^2}{2}]T}. \tag{20}$$

Since $\frac{1}{T} \sum_{t=0}^{T-1} \mathbb{E}\|d_t\|^2$ is just common descent directions. According to Definition. 3 shown in the paper, the quantity to our interest is $\|\sum_{s \in [S]} \lambda_t^s \nabla f(\mathbf{x})\|^2$.

$$\frac{1}{T} \sum_{t=0}^{T-1} \mathbb{E}\| \sum_{s \in [S]} \lambda_t^s \nabla f_s(\mathbf{x}_t)\|^2$$

$$\overset{(a)}{\leq} \frac{1}{T} \sum_{t=0}^{T-1} 2\mathbb{E}\| \sum_{s \in [S]} \lambda_t^s \nabla f_s(\mathbf{x}_t) - \sum_{s \in [S]} \lambda_t^s \mathbf{u}_t^s \|^2 + \frac{1}{T} \sum_{t=0}^{T-1} 2\mathbb{E}\| \sum_{s \in [S]} \lambda_t^s \mathbf{u}_t^s \|^2$$

$$\overset{(b)}{=} \frac{1}{T} \sum_{t=0}^{T-1} 2\mathbb{E}\| \sum_{s \in [S]} \lambda_t^s (\nabla f_s(\mathbf{x}_t) - \mathbf{u}_t^s)\|^2 + \frac{1}{T} \sum_{t=0}^{T-1} 2\mathbb{E}\|\mathbf{d}_t\|^2$$

$$\overset{(c)}{\leq} \frac{1}{T} \sum_{t=0}^{T-1} 2S \sum_{s \in [S]} (\lambda_t^s)^2 \mathbb{E}\|(\nabla f_s(\mathbf{x}_t) - \mathbf{u}_t^s)\|^2 + \frac{1}{T} \sum_{t=0}^{T-1} 2\mathbb{E}\|\mathbf{d}_t\|^2$$

$$\overset{(d)}{\leq} \frac{1}{T} \sum_{t=0}^{T-1} 2S \sum_{s \in [S]} (\lambda_t^s)^2 [\mathbb{E}_t \|\nabla f_s(\mathbf{x}_{(n_t-1)q}) - \mathbf{u}_{(n_t-1)q}^s\|^2 + L^2 \sum_{i=(n_t-1)q}^{t} \frac{1}{|\mathcal{A}|} \mathbb{E}\|\mathbf{x}_{i+1} - \mathbf{x}_i\|^2]$$

$$+ \frac{1}{T} \sum_{t=0}^{T-1} 2\mathbb{E}\|\mathbf{d}_t\|^2$$

$$= \frac{1}{T} \sum_{t=0}^{T-1} 2S \sum_{s \in [S]} (\lambda_t^s)^2 [\mathbb{E}_t \|\nabla f_s(\mathbf{x}_{(n_t-1)q}) - \mathbf{u}_{(n_t-1)q}^s\|^2]$$

$$+ 2SL^2 \frac{1}{T} \sum_{t=0}^{T-1} \sum_{i=(n_t-1)q}^{t} \frac{1}{|\mathcal{A}|} \mathbb{E}\|\mathbf{x}_{i+1} - \mathbf{x}_i\|^2 + \frac{1}{T} \sum_{t=0}^{T-1} 2\mathbb{E}\|\mathbf{d}_t\|^2$$

$$\overset{(e)}{\leq} \frac{1}{T} \sum_{t=0}^{T-1} 2S \sum_{s \in [S]} (\lambda_t^s)^2 [\mathbb{E}_t \|\nabla f_s(\mathbf{x}_{(n_t-1)q}) - \mathbf{u}_{(n_t-1)q}^s\|^2]$$

$$+ 2SL^2 \frac{1}{T} \sum_{t=0}^{T-1} \sum_{i=(n_t-1)q}^{n_t q - 1} \frac{1}{|\mathcal{A}|} \mathbb{E}\|\mathbf{x}_{t+1} - \mathbf{x}_t\|^2 + \frac{1}{T} \sum_{t=0}^{T-1} 2\mathbb{E}\|\mathbf{d}_t\|^2$$

$$= 2SL^2 \frac{1}{T} \sum_{t=0}^{T-1} \frac{q}{|\mathcal{A}|} \mathbb{E}\|\mathbf{x}_{t+1} - \mathbf{x}_t\|^2 + \frac{1}{T} \sum_{t=0}^{T-1} 2\mathbb{E}\|\mathbf{d}_t\|^2$$

$$\overset{(f)}{=} 2SL^2 \eta^2 \frac{1}{T} \sum_{t=0}^{T-1} \mathbb{E}\|\mathbf{d}_t\|^2 + \frac{1}{T} \sum_{t=0}^{T-1} 2\mathbb{E}\|\mathbf{d}_t\|^2$$

$$= (2SL^2 \eta^2 + 2) \frac{1}{T} \sum_{t=0}^{T-1} \mathbb{E}\|\mathbf{d}_t\|^2 \tag{21}$$

where $(a)$ and $(c)$ hold from the triangle inequality. (b) is because the definition $\mathbf{d}_t = \sum_{s \in [S]} \lambda_t^s \mathbf{u}_t^s$ as shown in Line 14 in Algorithm. 1. $(d)$ follows from the Lemma. 1. (e) is because $t \leq n_t q - 1$. $(f)$ is because we have $q = |\mathcal{A}| = \lceil \sqrt{n} \rceil$.

Then, we can conclude that

$$\frac{1}{T} \sum_{t=0}^{T-1} \mathbb{E}\| \sum_{s \in [S]} \lambda_t^s \nabla f_s(\mathbf{x}_t)\|^2 \overset{(a)}{\leq} (2SL^2 \eta^2 + 2) \frac{[[f_s(\mathbf{x}_0)] - f_s^*]}{[\frac{\eta}{4} - \frac{\eta^3 L^2}{2}]T}, \tag{22}$$

where $(a)$ follows from Eqs. (21) and Eqs. 20.

Let $\eta \leq \frac{1}{2L}$, we have

$$\frac{1}{T} \sum_{t=0}^{T-1} \mathbb{E}\| \sum_{s \in [S]} \lambda_t^s \nabla f_s(\mathbf{x}_t)\|^2$$

$$\leq \frac{(2SL^2 \eta^2 \frac{1}{T} + 2)[[f_s(\mathbf{x}_0)] - f_s^*]}{[\frac{\eta}{8}]T} = \frac{(2SL^2 \eta^2 + 2)\frac{8}{\eta}[[f_s(\mathbf{x}_0)] - f_s^*]}{T} = \mathcal{O}(\frac{1}{T}). \tag{23}$$

Lastly, to show the sample complexity, the number of samples with $mod(t,q) = 0$ can be calculated as: $\lceil \frac{T}{q} \rceil \cdot M$. Also, the number of samples with $mod(t,q) \neq 0$ can be calculated as $T \cdot |\mathcal{A}|$. Hence, the total sample complexity can be calculated as: $\lceil \frac{T}{q} \rceil n + T \cdot |\mathcal{A}| \leq \frac{T+q}{q} n + T\sqrt{n} = T\sqrt{n} + n + T\sqrt{n} = O(n + \sqrt{n}\epsilon^{-1})$. Thus, the overall sample complexity is $\mathcal{O}(n + \sqrt{n}\epsilon^{-1})$. This completes the proof.

$\square$

## A.1 Proof of Theorem. 2

*Proof.*

$$f_s(\mathbf{x}_{t+1})$$

$$\leq f_s(\mathbf{x}_t) + \langle \nabla f_s(\mathbf{x}_t), -\eta \mathbf{d}_t \rangle + \frac{1}{2}L\|\eta \mathbf{d}_t\|^2$$

$$\overset{(a)}{\leq} f_s(\mathbf{x}_*) + \langle \nabla f_s(\mathbf{x}_t), \mathbf{x}_t - \mathbf{x}_* \rangle - \frac{\mu}{2}\|\mathbf{x}_t - \mathbf{x}_*\|^2 + \langle \nabla f_s(\mathbf{x}_t), -\eta \mathbf{d}_t \rangle + \frac{1}{2}L\|\eta \mathbf{d}_t\|^2$$

$$= f_s(\mathbf{x}_*) + \langle \nabla f_s(\mathbf{x}_t), \mathbf{x}_t - \mathbf{x}_* - \eta \mathbf{d}_t \rangle - \frac{\mu}{2}\|\mathbf{x}_t - \mathbf{x}_*\|^2 + \frac{1}{2}L\|\eta \mathbf{d}_t\|^2$$

$$\overset{(b)}{\leq} f_s(\mathbf{x}_*) + \langle \nabla f_s(\mathbf{x}_t) - \mathbf{u}_t^s, \mathbf{x}_t - \mathbf{x}_* - \eta \mathbf{d}_t \rangle + \langle \mathbf{u}_t^s, \mathbf{x}_t - \mathbf{x}_* - \eta \mathbf{d}_t \rangle$$

$$- \frac{\mu}{2}\|\mathbf{x}_t - \mathbf{x}_*\|^2 + \frac{1}{2}L\|\eta \mathbf{d}_t\|^2$$

$$
\overset{(c)}{\leq} f_s(\mathbf{x}_*) + \frac{1}{2\delta}\|\nabla f_s(\mathbf{x}_t) - \mathbf{u}_t^s\|^2 + \frac{\delta}{2}\|\mathbf{x}_t - \mathbf{x}_* - \eta\mathbf{d}_t\|^2 + \langle \mathbf{u}_t^s, \mathbf{x}_t - \mathbf{x}_* - \eta\mathbf{d}_t \rangle
$$
$$
- \frac{\mu}{2}\|\mathbf{x}_t - \mathbf{x}_*\|^2 + \frac{1}{2}L\|\eta\mathbf{d}_t\|^2
$$
$$
\overset{(d)}{\leq} f_s(\mathbf{x}_*) + \frac{1}{2\delta}\|\nabla f_s(\mathbf{x}_t) - \mathbf{u}_t^s\|^2 + \delta\|\mathbf{x}_t - \mathbf{x}_*\|^2 + \delta\|\eta\mathbf{d}_t\|^2
$$
$$
+ \langle \mathbf{u}_t^s, \mathbf{x}_t - \mathbf{x}_* - \eta\mathbf{d}_t \rangle - \frac{\mu}{2}\|\mathbf{x}_t - \mathbf{x}_*\|^2 + \frac{1}{2}L\|\eta\mathbf{d}_t\|^2, \tag{24}
$$

the first inequality is due to $L$-smoothness, the second inequality follows from $\mu$-strongly convex. The last two inequality follows from the triangle inequality.

According to Definition. 3 shown in the paper, the quantity to our interest is $\sum_{s\in[S]} \lambda_t^s [f_s(\mathbf{x}_{t+1}) - f_s(\mathbf{x}_*)]$, then we have

$$
\sum_{s\in[S]} \lambda_t^s [f_s(\mathbf{x}_{t+1}) - f_s(\mathbf{x}_*)]
$$
$$
\overset{(a)}{\leq} \frac{1}{2\delta}\sum_{s\in[S]} \lambda_t^s\|\nabla f_s(\mathbf{x}_t) - \mathbf{u}_t^s\|^2 + \delta\|\mathbf{x}_t - \mathbf{x}_*\|^2 + \delta\|\eta\mathbf{d}_t\|^2
$$
$$
+ \left\langle \sum_{s\in[S]} \lambda_t^s\mathbf{u}_t^s, \mathbf{x}_t - \mathbf{x}_* \right\rangle - \frac{\mu}{2}\|\mathbf{x}_t - \mathbf{x}_*\|^2 + \left\langle \sum_{s\in[S]} \lambda_t^s\mathbf{u}_t^s, -\eta\mathbf{d}_t \right\rangle + \frac{1}{2}L\|\eta\mathbf{d}_t\|^2
$$
$$
= \frac{1}{2\delta}\sum_{s\in[S]} \lambda_t^s\|\nabla f_s(\mathbf{x}_t) - \mathbf{u}_t^s\|^2 + \delta\|\mathbf{x}_t - \mathbf{x}_*\|^2 + \delta\|\eta\mathbf{d}_t\|^2
$$
$$
+ \left\langle \sum_{s\in[S]} \lambda_t^s\mathbf{u}_t^s, \mathbf{x}_t - \mathbf{x}_* - \eta\mathbf{d}_t \right\rangle - \frac{\mu}{2}\|\mathbf{x}_t - \mathbf{x}_*\|^2 + \frac{1}{2}L\|\eta\mathbf{d}_t\|^2
$$
$$
\overset{(b)}{\leq} \frac{1}{2\delta}\sum_{s\in[S]} \lambda_t^s\|\nabla f_s(\mathbf{x}_t) - \mathbf{u}_t^s\|^2 + \delta\|\mathbf{x}_t - \mathbf{x}_*\|^2 + \delta\|\eta\mathbf{d}_t\|^2
$$
$$
+ \langle \mathbf{d}_t, \mathbf{x}_t - \mathbf{x}_* - \eta\mathbf{d}_t \rangle - \frac{\mu}{2}\|\mathbf{x}_t - \mathbf{x}_*\|^2 + \frac{1}{2}L\|\eta\mathbf{d}_t\|^2
$$
$$
= \langle \mathbf{d}_t, \mathbf{x}_t - \mathbf{x}_* \rangle - \eta\|\mathbf{d}_t\|^2 - \frac{\mu}{2}\|\mathbf{x}_t - \mathbf{x}_*\|^2 + \frac{1}{2}L\eta^2\|\mathbf{d}_t\|^2
$$
$$
+ \frac{1}{2\delta}\sum_{s\in[S]} \lambda_t^s\|\nabla f_s(\mathbf{x}_t) - \mathbf{u}_t^s\|^2 + \delta\|\mathbf{x}_t - \mathbf{x}_*\|^2 + \delta\|\eta\mathbf{d}_t\|^2
$$
$$
\overset{(c)}{\leq} \frac{1}{2\eta}\left(\|\mathbf{x}_t - \mathbf{x}_*\|^2 - \|\mathbf{x}_{t+1} - \mathbf{x}_*\|^2\right) - \frac{1}{2}\eta\|\mathbf{d}_t\|^2 - \frac{\mu}{2}\|\mathbf{x}_t - \mathbf{x}_*\|^2 + \frac{1}{2}L\eta^2\|\mathbf{d}_t\|^2
$$
$$
+ \frac{4}{\mu}\sum_{s\in[S]} \lambda_t^s\|\nabla f_s(\mathbf{x}_t) - \mathbf{u}_t^s\|^2 + \frac{\mu}{8}\|\mathbf{x}_t - \mathbf{x}_*\|^2 + \frac{\mu}{8}\|\eta\mathbf{d}_t\|^2
$$
$$
\overset{(d)}{\leq} \frac{1}{2\eta}\left((1 - \frac{3\mu\eta}{4})\|\mathbf{x}_t - \mathbf{x}_*\|^2 - \|\mathbf{x}_{t+1} - \mathbf{x}_*\|^2\right) - (\frac{1}{2}\eta - \frac{\mu}{8}\eta^2 - \frac{1}{2}L\eta^2)\|\mathbf{d}_t\|^2
$$
$$
+ \frac{4}{\mu}\sum_{s\in[S]} \lambda_t^s\|\nabla f_s(\mathbf{x}_t) - \mathbf{u}_t^s\|^2
$$
$$
\overset{(e)}{\leq} \frac{1}{2\eta}\left((1 - \frac{3\mu\eta}{4})\|\mathbf{x}_t - \mathbf{x}_*\|^2 - \|\mathbf{x}_{t+1} - \mathbf{x}_*\|^2\right) - (\frac{1}{2}\eta - \frac{\mu}{8}\eta^2 - \frac{1}{2}L\eta^2)\|\mathbf{d}_t\|^2
$$
$$
+ \frac{4}{\mu}(\frac{L^2}{|\mathcal{A}|}\sum_{i=(n_t-1)q}^{t}\|\mathbf{x}_{i+1} - \mathbf{x}_i\|^2 + \sum_{s\in[S]}\lambda_t^s\|\nabla f_s(\mathbf{x}_{(n_t-1)q}) - \mathbf{u}_{(n_t-1)q}^s\|^2)
$$

$$= \frac{1}{2\eta}\left((1 - \frac{3\mu\eta}{4})\|\mathbf{x}_t - \mathbf{x}_*\|^2 - \|\mathbf{x}_{t+1} - \mathbf{x}_*\|^2\right) - (\frac{1}{2}\eta - \frac{\mu}{8}\eta^2 - \frac{1}{2}L\eta^2)\|\mathbf{d}_t\|^2$$

$$+ \frac{4}{\mu}(\frac{L^2}{|\mathcal{A}|}\sum_{i=(n_t-1)q}^{t}\|\mathbf{x}_{i+1} - \mathbf{x}_i\|^2). \tag{25}$$

where $(a)$ follows from Eqs. (24). (b) is because the definition $\mathbf{d}_t = \sum_{s\in[S]}\lambda_t^s\mathbf{u}_t^s$ as shown in Line 14 in Algorithm. 1. $(c)$ is because $\|\mathbf{x}_t - \mathbf{x}_*\|^2 - \|\mathbf{x}_{t+1} - \mathbf{x}_*\|^2 = -\eta^2\|\mathbf{d}_t\|^2 + 2\langle\eta\mathbf{d}_t, \mathbf{x}_t - \mathbf{x}_*\rangle$, and we choose $\delta = \frac{\mu}{8}$ in $(d)$. $(e)$ follows from Lemma. 1. Next, telescoping the above inequality over $t$ from $(n_t - 1)q$ to $t$ where $t \leq n_t q - 1$ and noting that for $(n_t - 1)q \leq j \leq n_t q - 1, n_j = n_t$, we obtain

$$\sum_{i=(n_t-1)q}^{t}\sum_{s\in[S]}\lambda_i^s[f_s(\mathbf{x}_{i+1}) - f_s(\mathbf{x}_*)]$$

$$\overset{(a)}{\leq} \frac{1}{2\eta}\left((1 - \frac{3\mu\eta}{4})\sum_{i=(n_t-1)q}^{t}\|\mathbf{x}_i - \mathbf{x}_*\|^2 - \sum_{i=(n_t-1)q}^{t}\|\mathbf{x}_{i+1} - \mathbf{x}_*\|^2\right)$$

$$- (\frac{1}{2}\eta - \frac{\mu}{8}\eta^2 - \frac{1}{2}L\eta^2)\sum_{i=(n_t-1)q}^{t}\|\mathbf{d}_i\|^2 + \frac{4}{\mu}(\frac{L^2}{|\mathcal{A}|}\sum_{j=(n_t-1)q}^{t}\sum_{i=(n_j-1)q}^{j}\|\mathbf{x}_{i+1} - \mathbf{x}_i\|^2)$$

$$\overset{(b)}{\leq} \frac{1}{2\eta}\left((1 - \frac{3\mu\eta}{4})\sum_{i=(n_t-1)q}^{t}\|\mathbf{x}_i - \mathbf{x}_*\|^2 - \sum_{i=(n_t-1)q}^{t}\|\mathbf{x}_{i+1} - \mathbf{x}_*\|^2\right)$$

$$- (\frac{1}{2}\eta - \frac{\mu}{8}\eta^2 - \frac{1}{2}L\eta^2)\sum_{i=(n_t-1)q}^{t}\|\mathbf{d}_i\|^2 + \frac{4}{\mu}(\frac{L^2}{|\mathcal{A}|}\sum_{j=(n_t-1)q}^{t}\sum_{i=(n_t-1)q}^{t}\|\mathbf{x}_{i+1} - \mathbf{x}_i\|^2)$$

$$= \frac{1}{2\eta}\left((1 - \frac{3\mu\eta}{4})\sum_{i=(n_t-1)q}^{t}\|\mathbf{x}_i - \mathbf{x}_*\|^2 - \sum_{i=(n_t-1)q}^{t}\|\mathbf{x}_{i+1} - \mathbf{x}_*\|^2\right)$$

$$- (\frac{1}{2}\eta - \frac{\mu}{8}\eta^2 - \frac{1}{2}L\eta^2 - \frac{4}{\mu}\frac{L^2q\eta^2}{|\mathcal{A}|})\sum_{i=(n_t-1)q}^{t}\|\mathbf{d}_i\|^2, \tag{26}$$

where $(a)$ is from Eqs. (25). $(b)$ relaxes $j$ to $t$, since $j \leq t$. We continue the proof by further driving

$$\sum_{i=0}^{T}\sum_{s\in[S]}\lambda_i^s[f_s(\mathbf{x}_{i+1}) - f_s(\mathbf{x}_*)]$$

$$= \sum_{i=0}^{q}\sum_{s\in[S]}\lambda_i^s[f_s(\mathbf{x}_{i+1}) - f_s(\mathbf{x}_*)] + \sum_{i=q}^{2q}\sum_{s\in[S]}\lambda_i^s[f_s(\mathbf{x}_{i+1}) - f_s(\mathbf{x}_*)] +$$

$$\cdot + \sum_{i=(n_T-1)q}^{T}\sum_{s\in[S]}\lambda_i^s[f_s(\mathbf{x}_{i+1}) - f_s(\mathbf{x}_*)]$$

$$\leq \frac{1}{2\eta}\left((1 - \frac{3\mu\eta}{4})\sum_{i=0}^{T}\|\mathbf{x}_i - \mathbf{x}_*\|^2 - \sum_{i=0}^{T}\|\mathbf{x}_{i+1} - \mathbf{x}_*\|^2\right)$$

$$- (\frac{1}{2}\eta - \frac{\mu}{8}\eta^2 - \frac{1}{2}L\eta^2 - \frac{4}{\mu}\frac{L^2q\eta^2}{|\mathcal{A}|})\sum_{i=0}^{T}\|\mathbf{d}_i\|^2, \tag{27}$$

where the last inequality is from Eq. (16) and Eq. (26). Next, we have

$$\sum_{i=0}^{T} \sum_{s \in [S]} \lambda_i^s \left[ f_s(\mathbf{x}_i) - f_s(\mathbf{x}_*) \right]$$

$$= \sum_{i=0}^{T} \sum_{s \in [S]} \lambda_t^s \left[ f_s(\mathbf{x}_{i+1}) - f_s(\mathbf{x}_*) - f_s(\mathbf{x}_{i+1}) + f_s(\mathbf{x}_i) \right]$$

$$\leq \sum_{i=0}^{T} \sum_{s \in [S]} \lambda_t^s \left[ f_s(\mathbf{x}_{i+1}) - f_s(\mathbf{x}_*) \right] - \sum_{i=0}^{T} \sum_{s \in [S]} \lambda_t^s |f_s(\mathbf{x}_{i+1}) - f_s(\mathbf{x}_i)|$$

$$\leq \frac{1}{2\eta} \left( (1 - \frac{3\mu\eta}{4}) \sum_{i=0}^{T} \|\mathbf{x}_i - \mathbf{x}_*\|^2 - \sum_{i=0}^{T} \|\mathbf{x}_{i+1} - \mathbf{x}_*\|^2 \right)$$

$$- (\frac{1}{2}\eta - \frac{\mu}{8}\eta^2 - \frac{1}{2}L\eta^2 - \frac{4}{\mu}\frac{L^2 q\eta^2}{|\mathcal{A}|} - [\frac{\eta}{4} - \frac{\eta^3 q}{2}\frac{L^2}{|\mathcal{A}|}]) \sum_{i=0}^{T} \|\mathbf{d}_i\|^2 \quad (28)$$

Let $|\mathcal{A}| = q = \lceil \sqrt{n} \rceil$ and $\eta \leq \min\{\frac{1}{2\mu}, \frac{1}{8L}, \frac{\mu}{64L^2}\}$, we have $(\frac{1}{2}\eta - \frac{\mu}{8}\eta^2 - \frac{1}{2}L\eta^2 - \frac{4}{\mu}\frac{L^2 q\eta^2}{|\mathcal{A}|} - [\frac{\eta}{4} - \frac{\eta^3 q}{2}\frac{L^2}{|\mathcal{A}|}]) > \frac{\eta}{16} > 0$

Thus, we have

$$\sum_{i=0}^{T} \sum_{s \in [S]} \lambda_i^s \left[ f_s(\mathbf{x}_i) - f_s(\mathbf{x}_*) \right] \leq \frac{1}{2\eta} \left( (1 - \frac{3\mu\eta}{4}) \sum_{i=0}^{T} \|\mathbf{x}_i - \mathbf{x}_*\|^2 - \sum_{i=0}^{T} \|\mathbf{x}_{i+1} - \mathbf{x}_*\|^2 \right). \quad (29)$$

Then, we have

$$\mathbb{E}_t[\sum_{s \in [S]} \lambda_i^s \left[ f_s(\mathbf{x}_t) - f_s(\mathbf{x}_*) \right]] \leq \frac{1}{2\eta} \left( (1 - \frac{3\mu\eta}{4}) \mathbb{E}_t \|\mathbf{x}_t - \mathbf{x}_*\|^2 - \mathbb{E}_t \|\mathbf{x}_{t+1} - \mathbf{x}_*\|^2 \right). \quad (30)$$

Averaging using weight $w_t = (1 - \frac{3\mu\eta}{4})^{1-t}$ and using such weight to pick output $\mathbf{x}$, by using Lemma 1 in Karimireddy et al. (2020) with $\eta \geq \frac{1}{uR}$, we have

$$\mathbb{E}_t[\sum_{s \in [S]} \lambda_i^s \left[ f_s(\mathbf{x}_t) - f_s(\mathbf{x}_*) \right]] \leq \|\mathbf{x}_0 - \mathbf{x}_*\|^2 \mu \exp(-\frac{3\eta\mu T}{4}) \quad (31)$$

$$= \mathcal{O}(\mu \exp(-\mu T)). \quad (32)$$

Then we have the convergence rate $\mathbb{E}_t[\sum_{s \in [S]} \lambda_i^s \left[ f_s(\mathbf{x}_t) - f_s(\mathbf{x}_*) \right]] = \mathcal{O}(\mu \exp(-\mu T))$.

Lastly, the total sample complexity can be calculated as: $\lceil \frac{T}{q} \rceil n + T \cdot |\mathcal{A}| \leq \frac{T+q}{q} n + T\sqrt{n} = T\sqrt{n} + n + T\sqrt{n} = O(n + \sqrt{n} \ln(\mu/\epsilon))$. Thus, the overall sample complexity is $\mathcal{O}(n + \sqrt{n} \ln(\mu/\epsilon))$. This completes the proof.

$\square$

## B  PROOF OF CONVERGENCE OF STIMULUS-M

**Lemma 3.** *For general L-smooth functions $\{f_s, s \in [S]\}$, choose the learning rate $\eta$ s.t. $\eta \leq \frac{1}{2}$, the update $d_t$ of the VR-MOO-M algorithm satisfies:*

$$f_s(\mathbf{x}_{t+1}) \leq f_s(\mathbf{x}_t) + \frac{\eta}{2} \sum_{i=(n_t-1)q}^{t} \alpha^{(t-i)} \|\nabla f_s(\mathbf{x}_i) - \mathbf{u}_i^s\|^2 - \frac{1}{2}\eta \sum_{i=(n_t-1)q}^{t} \alpha^{(t-i)} \|\mathbf{d}_i\|^2$$

$$+ \frac{1}{2}L\|\mathbf{x}_{t+1} - \mathbf{x}_t\|^2. \quad (33)$$

**Proof of Lemma. 3.**

*Proof.*

$$f_s(\mathbf{x}_{t+1}) \le f_s(\mathbf{x}_t) + \langle \nabla f_s(\mathbf{x}_t), \mathbf{x}_{t+1} - \mathbf{x}_t \rangle + \frac{1}{2}L\|\mathbf{x}_{t+1} - \mathbf{x}_t\|^2$$

$$\overset{(a)}{\le} f_s(\mathbf{x}_t) + \langle \nabla f_s(\mathbf{x}_t), \alpha(\mathbf{x}_{t+1} - \mathbf{x}_t) \rangle + \langle \nabla f_s(\mathbf{x}_t), -\eta \mathbf{d}_t \rangle + \frac{1}{2}L\|\eta \mathbf{d}_t\|^2$$

$$\overset{(b)}{=} f_s(\mathbf{x}_t) + \sum_{i=0}^{t} \alpha^{(t-i)} \langle \nabla f_s(\mathbf{x}_i), -\eta \mathbf{d}_i \rangle + \frac{1}{2}L\|\eta \mathbf{d}_t\|^2$$

$$= f_s(\mathbf{x}_t) - \eta \sum_{i=0}^{t} \alpha^{(t-i)} \langle \nabla f_s(\mathbf{x}_i) - \mathbf{u}_i^s, \mathbf{d}_i \rangle - \eta \sum_{i=0}^{t} \alpha^{(t-i)} \langle \mathbf{u}_i^s, \mathbf{d}_i \rangle + \frac{1}{2}L\|\mathbf{x}_{t+1} - \mathbf{x}_t\|^2$$

$$\overset{(c)}{\le} f_s(\mathbf{x}_t) - \eta \sum_{i=0}^{t} \alpha^{(t-i)} \langle \nabla f_s(\mathbf{x}_i) - \mathbf{u}_i^s, \mathbf{d}_i \rangle - \eta \sum_{i=0}^{t} \alpha^{(t-i)} \|\mathbf{d}_i\|^2 + \frac{1}{2}L\|\mathbf{x}_{t+1} - \mathbf{x}_t\|^2$$

$$\overset{(d)}{\le} f_s(\mathbf{x}_t) + \frac{\eta}{2} \sum_{i=0}^{t} \alpha^{(t-i)} \|\nabla f_s(\mathbf{x}_i) - \mathbf{u}_i^s\|^2 + \frac{1}{2}\eta \sum_{i=0}^{t} \alpha^{(t-i)} \|\mathbf{d}_i\|^2$$

$$- \eta \sum_{i=0}^{t} \alpha^{(t-i)} \|\mathbf{d}_i\|^2 + \frac{1}{2}L\|\mathbf{x}_{t+1} - \mathbf{x}_t\|^2$$

$$= f_s(\mathbf{x}_t) + \frac{\eta}{2} \sum_{i=0}^{t} \alpha^{(t-i)} \|\nabla f_s(\mathbf{x}_i) - \mathbf{u}_i^s\|^2 - \frac{1}{2}\eta \sum_{i=0}^{t} \alpha^{(t-i)} \|\mathbf{d}_i\|^2 + \frac{1}{2}L\|\mathbf{x}_{t+1} - \mathbf{x}_t\|^2. \quad (34)$$

(a) follows from the objective function $f_s$ is $L$-smooth. $(b)$ follows from the update rule of $\mathbf{x}_t$ shown in Line 19 in Algorithm. 1. (c) follows from $\langle \mathbf{u}_t^s, \mathbf{d}_t \rangle \ge \|\mathbf{d}_t\|^2$ since $\mathbf{d}_t$ is a general solution in the convex hull of the family of vectors $\{\mathbf{u}_t^s, s \in [S]\}$ (see Lemma 2.1 Désidéri (2012)). (d) follows from the triangle inequality.

□

**Proof of Theorem. 3**

*Proof.* Taking expectation on both sides of the inequality in Lemma. 3, we have

$$\mathbb{E}[f_s(\mathbf{x}_{t+1})]$$

$$\overset{(a)}{\le} \mathbb{E}[f_s(\mathbf{x}_t)] + \frac{\eta}{2} \sum_{i=0}^{t} \alpha^{(t-i)} \mathbb{E}\|\nabla f_s(\mathbf{x}_i) - \mathbf{u}_i^s\|^2 - \frac{1}{2}\eta \sum_{i=0}^{t} \alpha^{(t-i)} \mathbb{E}\|\mathbf{d}_i\|^2 + \frac{1}{2}L\mathbb{E}\|\mathbf{x}_{t+1} - \mathbf{x}_t\|^2$$

$$\overset{(b)}{\le} \mathbb{E}[f_s(\mathbf{x}_t)] - \frac{1}{2}\eta \sum_{i=0}^{t} \alpha^{(t-i)} \mathbb{E}\|\mathbf{d}_i\|^2 + \frac{1}{2}L\mathbb{E}\|\mathbf{x}_{t+1} - \mathbf{x}_t\|^2$$

$$+ \frac{\eta}{2} \sum_{j=0}^{t} \alpha^{(t-j)} \Big[ \frac{L^2}{|\mathcal{A}|} \sum_{i=(n_t-1)q}^{j} \mathbb{E}\|\mathbf{x}_{i+1} - \mathbf{x}_i\|^2 + \mathbb{E}\|\nabla f_s(\mathbf{x}_{(n_t-1)q}) - \mathbf{u}_{(n_t-1)q}^s\|^2 \Big]$$

$$= \mathbb{E}[f_s(\mathbf{x}_t)] - \frac{1}{2}\eta \sum_{i=0}^{t} \alpha^{(t-i)} \mathbb{E}\|\mathbf{d}_i\|^2 + \frac{1}{2}L\mathbb{E}\|\mathbf{x}_{t+1} - \mathbf{x}_t\|^2$$

$$+ \frac{\eta}{2} \sum_{j=0}^{t} \alpha^{(t-j)} \Big[ \frac{L^2}{|\mathcal{A}|} \sum_{i=(n_t-1)q}^{j} \mathbb{E}\|\mathbf{x}_{i+1} - \mathbf{x}_i\|^2 \Big], \quad (35)$$

where $(a)$ follows from Eqs. 34. $(b)$ follows from the Lemma. 1. $(c)$ follows from $\mathbb{E}\|\nabla f_s(\mathbf{x}_{(n_t-1)q}) - \mathbf{u}^s_{(n_t-1)q}\|^2 = 0$ as shown in Line 5 in our Algorithm. 1.

Next, telescoping the above inequality over $t$ from $(n_t - 1)q$ to $t$ where $t \le n_t q - 1$ and noting that for $(n_t - 1)q \le j \le n_t q - 1, n_j = n_t$ and let $\eta \le \frac{1}{4L}$, we obtain

$$
\mathbb{E}[f_s(\mathbf{x}_{t+1})]
$$

$$
\overset{(a)}{\le} \mathbb{E}[f_s(\mathbf{x}_{(n_t-1)q})] - \frac{\eta}{2} \sum_{j=(n_t-1)q}^{t} \sum_{i=0}^{j} \alpha^{(j-i)} \mathbb{E}\|\mathbf{d}_i\|^2 + \frac{1}{2}L \sum_{i=(n_t-1)q}^{t} \mathbb{E}\|\mathbf{x}_{i+1} - \mathbf{x}_i\|^2
$$

$$
+ \frac{\eta}{2} \sum_{j=(n_t-1)q}^{t} \sum_{i=0}^{j} \alpha^{(j-i)} [\frac{L^2}{|\mathcal{A}|} \sum_{r=(n_t-1)q}^{i} \mathbb{E}\|\mathbf{x}_{r+1} - \mathbf{x}_r\|^2]
$$

$$
\overset{(b)}{\le} \mathbb{E}[f_s(\mathbf{x}_{(n_t-1)q})] - \frac{\eta}{2} \sum_{j=(n_t-1)q}^{t} \sum_{i=0}^{j} \alpha^{(j-i)} \mathbb{E}\|\mathbf{d}_i\|^2 + \frac{1}{2}L \sum_{i=(n_t-1)q}^{t} \mathbb{E}\|\mathbf{x}_{i+1} - \mathbf{x}_i\|^2
$$

$$
+ \frac{\eta}{2} \sum_{j=(n_t-1)q}^{t} \sum_{i=0}^{j} \alpha^{(j-i)} [\frac{L^2}{|\mathcal{A}|} \sum_{r=(n_t-1)q}^{n_t q-1} \mathbb{E}\|\mathbf{x}_{r+1} - \mathbf{x}_r\|^2]
$$

$$
\le \mathbb{E}[f_s(\mathbf{x}_{(n_t-1)q})] - \frac{\eta}{2} \sum_{j=(n_t-1)q}^{t} \sum_{i=0}^{j} \alpha^{(j-i)} \mathbb{E}\|\mathbf{d}_i\|^2 + \frac{1}{2}L \sum_{i=(n_t-1)q}^{t} \mathbb{E}\|\mathbf{x}_{i+1} - \mathbf{x}_i\|^2
$$

$$
+ \frac{\eta}{2} \sum_{j=(n_t-1)q}^{t} \sum_{i=0}^{j} \alpha^{(j-i)} [\frac{L^2}{|\mathcal{A}|} q \mathbb{E}\|\mathbf{x}_{j+1} - \mathbf{x}_j\|^2]
$$

$$
\overset{(c)}{=} \mathbb{E}[f_s(\mathbf{x}_{(n_t-1)q})] - \frac{\eta}{2} \sum_{j=(n_t-1)q}^{t} \sum_{i=0}^{j} \alpha^{(j-i)} \mathbb{E}\|\mathbf{d}_i\|^2 + \frac{1}{2}L \sum_{i=(n_t-1)q}^{t} \mathbb{E}\|\mathbf{x}_{i+1} - \mathbf{x}_i\|^2
$$

$$
+ \frac{\eta}{2} \sum_{j=(n_t-1)q}^{t} \sum_{i=0}^{j} \alpha^{(j-i)} [L^2 \mathbb{E}\|\mathbf{x}_{j+1} - \mathbf{x}_j\|^2]
$$

$$
\overset{(d)}{=} \mathbb{E}[f_s(\mathbf{x}_{(n_t-1)q})] - \frac{\eta}{2} \sum_{j=(n_t-1)q}^{t} \sum_{i=0}^{j} \alpha^{(j-i)} \mathbb{E}\|\mathbf{d}_i\|^2 + \frac{1}{2}L \sum_{i=(n_t-1)q}^{t} \mathbb{E}\|\mathbf{x}_{i+1} - \mathbf{x}_i\|^2
$$

$$
+ \frac{\eta}{2} \sum_{j=(n_t-1)q}^{t} \sum_{i=0}^{j} \alpha^{(j-i)} [L^2 \mathbb{E}\|\eta \sum_{r=0}^{j} \alpha^{(j-r)} \mathbf{d}_r\|^2]
$$

$$
\overset{(e)}{\le} \mathbb{E}[f_s(\mathbf{x}_{(n_t-1)q})] - \frac{\eta}{2} \sum_{j=(n_t-1)q}^{t} \sum_{i=0}^{j} \alpha^{(j-i)} \mathbb{E}\|\mathbf{d}_i\|^2 + \frac{1}{2}L \sum_{i=(n_t-1)q}^{t} \mathbb{E}\|\mathbf{x}_{i+1} - \mathbf{x}_i\|^2
$$

$$
+ \frac{\eta}{2} \sum_{j=(n_t-1)q}^{t} \sum_{i=0}^{j} \alpha^{2(j-i)} [L^2 \eta^2 \mathbb{E}\|\mathbf{d}_i\|^2]
$$

$$
\overset{(f)}{\le} \mathbb{E}[f_s(\mathbf{x}_{(n_t-1)q})] - \frac{\eta}{4} \sum_{j=(n_t-1)q}^{t} \sum_{i=0}^{j} \alpha^{(j-i)} \mathbb{E}\|\mathbf{d}_i\|^2 + \frac{1}{2}L \sum_{j=(n_t-1)q}^{t} \mathbb{E}\|\eta \sum_{i=0}^{j} \alpha^{(j-i)} \mathbf{d}_j\|^2
$$

$$
\overset{(g)}{\le} \mathbb{E}[f_s(\mathbf{x}_{(n_t-1)q})] - \frac{\eta}{8} \sum_{j=(n_t-1)q}^{t} \mathbb{E}\|\mathbf{d}_j\|^2, \tag{36}
$$

where $(a)$ holds from Eqs. (35). $(b)$ is extend $i$ to $t$ since $i \le n_t q - 1$. $(c)$ is because $q = |\mathcal{A}| = \lceil \sqrt{n} \rceil$. $(d)$ follows from the update rule of $\mathbf{x}_t$ shown in Line 19 in Algorithm. 1. $(e)$ follows from the triangle

inequality. $(f)$ and $(g)$ hold from $\eta \leq \frac{1}{2L}$ and $0 < \alpha < 1$. We continue the proof by further driving

$$
\begin{aligned}
&[f_s(\mathbf{x}_T)] - [f_s(\mathbf{x}_0)] \\
&= ([f_s(\mathbf{x}_q)] - [f_s(\mathbf{x}_0)]) + ([f_s(\mathbf{x}_{2q})] - [f_s(\mathbf{x}_q)]) + \cdot + ([f_s(\mathbf{x}_T)] - [f_s(\mathbf{x}_{(n_T-1)q})]) \\
&\leq -[\frac{\eta}{8}] \sum_{t=0}^{T-1} \|\mathbf{d}_t\|^2
\end{aligned}
\tag{37}
$$

Note that $[f_s(\mathbf{x}_{T+1})] \geq f_s^* \triangleq \inf_{\mathbf{x} \in \mathbb{R}^d} f_s(\mathbf{x})$. Hence, we have

$$
[\frac{\eta}{8}] \sum_{t=0}^{T-1} \|\mathbf{d}_t\|^2 \leq [[f_s(\mathbf{x}_0)] - [f_s(\mathbf{x}_T)]] \leq [[f_s(\mathbf{x}_0)] - f_s^*].
\tag{38}
$$

Based on the parameter setting $q = |\mathcal{A}| = \sqrt{n}$, we have

$$
[\frac{\eta}{8}] \sum_{t=0}^{T-1} \|\mathbf{d}_t\|^2 \leq [[f_s(\mathbf{x}_0)] - f_s^*].
\tag{39}
$$

Since $\frac{1}{T} \sum_{t=0}^{T-1} \mathbb{E}\|d_t\|^2$ is just common descent directions. According to Definition. 3 shown in the paper, the quantity to our interest is $\|\sum_{s \in [S]} \lambda_t^s \nabla f(\mathbf{x})\|^2$.

$$
\frac{1}{T} \sum_{t=0}^{T-1} \mathbb{E}\| \sum_{s \in [S]} \lambda_t^s \nabla f_s(\mathbf{x}_t)\|^2 \overset{(a)}{\leq} (2SL^2\eta^2 \frac{1}{T} + 2) \frac{1}{T} \sum_{t=0}^{T-1} \mathbb{E}\|\mathbf{d}_t\|^2
\tag{40}
$$

where $(a)$ follows from Eqs. (21).

Then, we can conclude that

$$
\frac{1}{T} \sum_{t=0}^{T-1} \mathbb{E}\| \sum_{s \in [S]} \lambda_t^s \nabla f_s(\mathbf{x}_t)\|^2 \overset{(a)}{\leq} (2SL^2\eta^2 + 2) \frac{[\mathbb{E}[f_s(\mathbf{x}_0)] - f_s^*]}{\frac{\eta}{8}T},
\tag{41}
$$

where $(a)$ follows from Eqs. (21) and Eqs. 20.

Thus, we have

$$
\frac{1}{T} \sum_{t=0}^{T-1} \mathbb{E}\| \sum_{s \in [S]} \lambda_t^s \nabla f_s(\mathbf{x}_t)\|^2 = \mathcal{O}(\frac{1}{T}).
\tag{42}
$$

The total sample complexity can be calculated as: $\lceil \frac{T}{q} \rceil n + T \cdot |\mathcal{A}| \leq \frac{T+q}{q} n + T\sqrt{n} = T\sqrt{n} + n + T\sqrt{n} = O(n + \sqrt{n}\epsilon^{-1})$. Thus, the overall sample complexity is $\mathcal{O}(n + \sqrt{n}\epsilon^{-1})$. This completes the proof.

$\square$

### B.1 PROOF OD THEOREM. 4

*Proof.*

$$
\begin{aligned}
&f_s(\mathbf{x}_{t+1}) \\
&\overset{(a)}{\leq} f_s(\mathbf{x}_t) + \left\langle \nabla f_s(\mathbf{x}_t), -\eta \sum_{t=0}^{T} \alpha^{(t-i)} \mathbf{d}_i \right\rangle + \frac{1}{2}L\|\eta \sum_{t=0}^{T} \alpha^{(t-i)} \mathbf{d}_i\|^2 \\
&\overset{(b)}{\leq} f_s(\mathbf{x}_*) + \langle \nabla f_s(\mathbf{x}_t), \mathbf{x}_t - \mathbf{x}_* \rangle - \frac{\mu}{2}\|\mathbf{x}_t - \mathbf{x}_*\|^2 + \left\langle \nabla f_s(\mathbf{x}_t), -\eta \sum_{t=0}^{T} \alpha^{(t-i)} \mathbf{d}_i \right\rangle
\end{aligned}
$$

$$+ \frac{1}{2}L\|\eta\sum_{t=0}^{T}\alpha^{(t-i)}\mathbf{d}_i\|^2$$

$$= f_s(\mathbf{x}_*) + \left\langle \nabla f_s(\mathbf{x}_t), \mathbf{x}_t - \mathbf{x}_* - \eta\sum_{t=0}^{T}\alpha^{(t-i)}\mathbf{d}_i \right\rangle - \frac{\mu}{2}\|\mathbf{x}_t - \mathbf{x}_*\|^2 + \frac{1}{2}L\|\eta\sum_{t=0}^{T}\alpha^{(t-i)}\mathbf{d}_i\|^2$$

$$= f_s(\mathbf{x}_*) + \left\langle \nabla f_s(\mathbf{x}_t) - \mathbf{u}_t^s, \mathbf{x}_t - \mathbf{x}_* - \eta\sum_{t=0}^{T}\alpha^{(t-i)}\mathbf{d}_i \right\rangle + \left\langle \mathbf{u}_t^s, \mathbf{x}_t - \mathbf{x}_* - \eta\sum_{t=0}^{T}\alpha^{(t-i)}\mathbf{d}_i \right\rangle$$

$$- \frac{\mu}{2}\|\mathbf{x}_t - \mathbf{x}_*\|^2 + \frac{1}{2}L\|\eta\sum_{t=0}^{T}\alpha^{(t-i)}\mathbf{d}_i\|^2$$

$$\overset{(c)}{\leq} f_s(\mathbf{x}_*) + \frac{1}{2\delta}\|\nabla f_s(\mathbf{x}_t) - \mathbf{u}_t^s\|^2 + \frac{\delta}{2}\|\mathbf{x}_t - \mathbf{x}_* - \eta\sum_{t=0}^{T}\alpha^{(t-i)}\mathbf{d}_i\|^2$$

$$+ \left\langle \mathbf{u}_t^s, \mathbf{x}_t - \mathbf{x}_* - \eta\sum_{t=0}^{T}\alpha^{(t-i)}\mathbf{d}_i \right\rangle - \frac{\mu}{2}\|\mathbf{x}_t - \mathbf{x}_*\|^2 + \frac{1}{2}L\|\eta\sum_{t=0}^{T}\alpha^{(t-i)}\mathbf{d}_i\|^2$$

$$\overset{(d)}{\leq} f_s(\mathbf{x}_*) + \frac{1}{2\delta}\|\nabla f_s(\mathbf{x}_t) - \mathbf{u}_t^s\|^2 + \delta\|\mathbf{x}_t - \mathbf{x}_*\|^2 + \delta\|\eta\sum_{t=0}^{T}\alpha^{(t-i)}\mathbf{d}_i\|^2$$

$$+ \left\langle \mathbf{u}_t^s, \mathbf{x}_t - \mathbf{x}_* - \eta\sum_{t=0}^{T}\alpha^{(t-i)}\mathbf{d}_i \right\rangle - \frac{\mu}{2}\|\mathbf{x}_t - \mathbf{x}_*\|^2 + \frac{1}{2}L\|\eta\sum_{t=0}^{T}\alpha^{(t-i)}\mathbf{d}_i\|^2, \tag{43}$$

where $(a)$ is due to $L$-smoothness, $(b)$ follows from $\mu$-strongly convex. $(c)$ and $(d)$ follow from the Young's inequality.

Next, we have

$$\sum_{s\in[S]} \lambda_t^s \left[ f_s(\mathbf{x}_{t+1}) - f_s(\mathbf{x}_*) \right]$$

$$\overset{(a)}{\leq} \frac{1}{2\delta} \sum_{s\in[S]} \lambda_t^s \|\nabla f_s(\mathbf{x}_t) - \mathbf{u}_t^s\|^2 + \delta\|\mathbf{x}_t - \mathbf{x}_*\|^2 + \delta\|\eta\sum_{t=0}^{T}\alpha^{(t-i)}\mathbf{d}_i\|^2$$

$$+ \left\langle \sum_{s\in[S]} \lambda_t^s \mathbf{u}_t^s, \mathbf{x}_t - \mathbf{x}_* \right\rangle - \frac{\mu}{2}\|\mathbf{x}_t - \mathbf{x}_*\|^2 + \left\langle \sum_{s\in[S]} \lambda_t^s \mathbf{u}_t^s, -\eta\sum_{t=0}^{T}\alpha^{(t-i)}\mathbf{d}_i \right\rangle$$

$$+ \frac{1}{2}L\|\eta\sum_{t=0}^{T}\alpha^{(t-i)}\mathbf{d}_i\|^2$$

$$= \frac{1}{2\delta} \sum_{s\in[S]} \lambda_t^s \|\nabla f_s(\mathbf{x}_t) - \mathbf{u}_t^s\|^2 + \delta\|\mathbf{x}_t - \mathbf{x}_*\|^2 + \delta\|\eta\sum_{t=0}^{T}\alpha^{(t-i)}\mathbf{d}_i\|^2$$

$$+ \left\langle \sum_{s\in[S]} \lambda_t^s \mathbf{u}_t^s, \mathbf{x}_t - \mathbf{x}_* - \eta\sum_{t=0}^{T}\alpha^{(t-i)}\mathbf{d}_i \right\rangle - \frac{\mu}{2}\|\mathbf{x}_t - \mathbf{x}_*\|^2 + \frac{1}{2}L\|\eta\sum_{t=0}^{T}\alpha^{(t-i)}\mathbf{d}_i\|^2$$

$$\overset{(b)}{=} \frac{1}{2\delta} \sum_{s\in[S]} \lambda_t^s \|\nabla f_s(\mathbf{x}_t) - \mathbf{u}_t^s\|^2 + \delta\|\mathbf{x}_t - \mathbf{x}_*\|^2 + \delta\|\eta\sum_{t=0}^{T}\alpha^{(t-i)}\mathbf{d}_i\|^2$$

$$+ \left\langle \mathbf{d}_t, \mathbf{x}_t - \mathbf{x}_* - \eta\sum_{t=0}^{T}\alpha^{(t-i)}\mathbf{d}_i \right\rangle - \frac{\mu}{2}\|\mathbf{x}_t - \mathbf{x}_*\|^2 + \frac{1}{2}L\|\eta\sum_{t=0}^{T}\alpha^{(t-i)}\mathbf{d}_i\|^2$$

$$\overset{(c)}{\leq} \frac{1}{2\eta} \left( \|\mathbf{x}_t - \mathbf{x}_*\|^2 - \|\mathbf{x}_{t+1} - \mathbf{x}_*\|^2 \right) - \frac{1}{2}\eta\|\sum_{t=0}^{T}\alpha^{(t-i)}\mathbf{d}_i\|^2 - \frac{\mu}{2}\|\mathbf{x}_t - \mathbf{x}_*\|^2$$

$$+ \frac{1}{2}L\|\eta\sum_{t=0}^{T}\alpha^{(t-i)}\mathbf{d}_i\|^2 + \frac{4}{\mu}\sum_{s\in[S]}\lambda_t^s\|\nabla f_s(\mathbf{x}_t) - \mathbf{u}_t^s\|^2 + \frac{\mu}{8}\|\mathbf{x}_t - \mathbf{x}_*\|^2 + \frac{\mu}{8}\|\eta\sum_{t=0}^{T}\alpha^{(t-i)}\mathbf{d}_i\|^2$$

$$= \frac{1}{2\eta}\left((1 - \frac{3\mu\eta}{4})\|\mathbf{x}_t - \mathbf{x}_*\|^2 - \|\mathbf{x}_{t+1} - \mathbf{x}_*\|^2\right) - (\frac{1}{2}\eta - \frac{\mu}{8}\eta^2 - \frac{1}{2}L\eta^2)\|\sum_{t=0}^{T}\alpha^{(t-i)}\mathbf{d}_i\|^2$$

$$+ \frac{4}{\mu}\sum_{s\in[S]}\lambda_t^s\|\nabla f_s(\mathbf{x}_t) - \mathbf{u}_t^s\|^2$$

$$\overset{(e)}{\leq} \frac{1}{2\eta}\left((1 - \frac{3\mu\eta}{4})\|\mathbf{x}_t - \mathbf{x}_*\|^2 - \|\mathbf{x}_{t+1} - \mathbf{x}_*\|^2\right) - (\frac{1}{2}\eta - \frac{\mu}{8}\eta^2 - \frac{1}{2}L\eta^2)\|\sum_{t=0}^{T}\alpha^{(t-i)}\mathbf{d}_i\|^2$$

$$+ \frac{4}{\mu}(\frac{L^2}{|\mathcal{A}|}\sum_{i=(n_t-1)q}^{t}\|\mathbf{x}_{i+1} - \mathbf{x}_i\|^2 + \sum_{s\in[S]}\lambda_t^s\|\nabla f_s(\mathbf{x}_{(n_t-1)q}) - \mathbf{u}_{(n_t-1)q}^s\|^2)$$

$$\overset{(f)}{=} \frac{1}{2\eta}\left((1 - \frac{3\mu\eta}{4})\|\mathbf{x}_t - \mathbf{x}_*\|^2 - \|\mathbf{x}_{t+1} - \mathbf{x}_*\|^2\right) - (\frac{1}{2}\eta - \frac{\mu}{8}\eta^2 - \frac{1}{2}L\eta^2)\|\sum_{t=0}^{T}\alpha^{(t-i)}\mathbf{d}_i\|^2$$

$$+ \frac{4}{\mu}(\frac{L^2}{|\mathcal{A}|}\sum_{i=(n_t-1)q}^{t}\|\mathbf{x}_{i+1} - \mathbf{x}_i\|^2). \tag{44}$$

where $(a)$ follows from Eqs. (43). (b) is because the definition $\mathbf{d}_t = \sum_{s\in[S]}\lambda_t^s\mathbf{u}_t^s$ as shown in Line 14 in Algorithm. 1. $(c)$ is because $\|\mathbf{x}_t - \mathbf{x}_*\|^2 - \|\mathbf{x}_{t+1} - \mathbf{x}_*\|^2 = -\eta^2\|\sum_{t=0}^{T}\alpha^{(t-i)}\mathbf{d}_i\|^2 + 2\left\langle\eta\sum_{t=0}^{T}\alpha^{(t-i)}\mathbf{d}_i, \mathbf{x}_t - \mathbf{x}_*\right\rangle$, and we choose $\delta = \frac{\mu}{8}$. $(e)$ and $(f)$ follow from $\sum_{s\in[S]}\lambda_t^s = 1$ and $\|\nabla f_s(\mathbf{x}_{(n_t-1)q}) - \mathbf{u}_{(n_t-1)q}^s\|^2 = 0$.

Next, telescoping the above inequality over $t$ from $(n_t - 1)q$ to $t$ where $t \leq n_tq - 1$ and noting that for $(n_t - 1)q \leq j \leq n_tq - 1, n_j = n_t$, we obtain

$$\sum_{i=(n_t-1)q}^{t}\sum_{s\in[S]}\lambda_t^s\left[f_s(\mathbf{x}_{i+1}) - f_s(\mathbf{x}_*)\right]$$

$$\overset{(a)}{=} \frac{1}{2\eta}\left((1 - \frac{3\mu\eta}{4})\sum_{i=(n_t-1)q}^{t}\|\mathbf{x}_i - \mathbf{x}_*\|^2 - \sum_{i=(n_t-1)q}^{t}\|\mathbf{x}_{i+1} - \mathbf{x}_*\|^2\right)$$

$$- (\frac{1}{2}\eta - \frac{\mu}{8}\eta^2 - \frac{1}{2}L\eta^2)\sum_{i=(n_t-1)q}^{t}\|\sum_{t=0}^{T}\alpha^{(t-i)}\mathbf{d}_i\|^2 + \frac{4}{\mu}(\frac{L^2}{|\mathcal{A}|}\sum_{j=(n_t-1)q}^{t}\sum_{i=(n_j-1)q}^{j}\|\mathbf{x}_{i+1} - \mathbf{x}_i\|^2)$$

$$\overset{(b)}{\leq} \frac{1}{2\eta}\left((1 - \frac{3\mu\eta}{4})\sum_{i=(n_t-1)q}^{t}\|\mathbf{x}_i - \mathbf{x}_*\|^2 - \sum_{i=(n_t-1)q}^{t}\|\mathbf{x}_{i+1} - \mathbf{x}_*\|^2\right)$$

$$- (\frac{1}{2}\eta - \frac{\mu}{8}\eta^2 - \frac{1}{2}L\eta^2)\sum_{i=(n_t-1)q}^{t}\|\sum_{t=0}^{T}\alpha^{(t-i)}\mathbf{d}_i\|^2 + \frac{4}{\mu}(\frac{L^2}{|\mathcal{A}|}\sum_{j=(n_t-1)q}^{t}\sum_{i=(n_t-1)q}^{t}\|\mathbf{x}_{i+1} - \mathbf{x}_i\|^2)$$

$$\overset{(c)}{=} \frac{1}{2\eta}\left((1 - \frac{3\mu\eta}{4})\sum_{i=(n_t-1)q}^{t}\|\mathbf{x}_i - \mathbf{x}_*\|^2 - \sum_{i=(n_t-1)q}^{t}\|\mathbf{x}_{i+1} - \mathbf{x}_*\|^2\right)$$

$$- (\frac{1}{2}\eta - \frac{\mu}{8}\eta^2 - \frac{1}{2}L\eta^2 - \frac{4}{\mu}\frac{L^2q\eta^2}{|\mathcal{A}|})\sum_{i=(n_t-1)q}^{t}\|\sum_{t=0}^{T}\alpha^{(t-i)}\mathbf{d}_i\|^2), \tag{45}$$

where $(a)$ follows from Eqs. (44), $(b)$ extend $j$ to $t$. $(c)$ follows from the update rule of $\mathbf{x}_{t+1}$ shown in Eqs. (5).

We continue the proof by further driving

$$\sum_{t=0}^{T} \sum_{s \in [S]} \lambda_t^s \left[ f_s(\mathbf{x}_{i+1}) - f_s(\mathbf{x}_*) \right]$$

$$= \sum_{i=0}^{q} \sum_{s \in [S]} \lambda_t^s \left[ f_s(\mathbf{x}_{i+1}) - f_s(\mathbf{x}_*) \right] + \sum_{i=q}^{2q} \sum_{s \in [S]} \lambda_t^s \left[ f_s(\mathbf{x}_{i+1}) - f_s(\mathbf{x}_*) \right] +$$

$$\sum_{i=(n_T-1)q}^{T} \sum_{s \in [S]} \lambda_t^s \left[ f_s(\mathbf{x}_{i+1}) - f_s(\mathbf{x}_*) \right]$$

$$\overset{(a)}{\leq} \frac{1}{2\eta} \left( (1 - \frac{3\mu\eta}{4}) \sum_{i=0}^{T} \|\mathbf{x}_i - \mathbf{x}_*\|^2 - \sum_{t=0}^{T} \|\mathbf{x}_{i+1} - \mathbf{x}_*\|^2 \right)$$

$$- (\frac{1}{2}\eta - \frac{\mu}{8}\eta^2 - \frac{1}{2}L\eta^2 - \frac{4}{\mu} \frac{L^2 q \eta^2}{|\mathcal{A}|}) \sum_{t=0}^{T} \| \sum_{t=0}^{T} \alpha^{(t-i)} \mathbf{d}_i \|^2), \tag{46}$$

where $(a)$ follows from Eqs. (45). Next, we have

$$\sum_{t=0}^{T} \sum_{s \in [S]} \lambda_t^s \left[ f_s(\mathbf{x}_i) - f_s(\mathbf{x}_*) \right]$$

$$= \sum_{t=0}^{T} \sum_{s \in [S]} \lambda_t^s \left[ f_s(\mathbf{x}_{i+1}) - f_s(\mathbf{x}_*) - f_s(\mathbf{x}_{i+1}) + f_s(\mathbf{x}_i) \right]$$

$$= \sum_{t=0}^{T} \sum_{s \in [S]} \lambda_t^s \left[ f_s(\mathbf{x}_{i+1}) - f_s(\mathbf{x}_*) \right] - \sum_{t=0}^{T} \sum_{s \in [S]} \lambda_t^s |f_s(\mathbf{x}_{i+1}) - f_s(\mathbf{x}_i)|$$

$$\overset{(a)}{leq} \frac{1}{2\eta} \left( (1 - \frac{3\mu\eta}{4}) \sum_{i=0}^{T} \|\mathbf{x}_i - \mathbf{x}_*\|^2 - \sum_{t=0}^{T} \|\mathbf{x}_{i+1} - \mathbf{x}_*\|^2 \right)$$

$$- (\frac{1}{2}\eta - \frac{\mu}{8}\eta^2 - \frac{1}{2}L\eta^2 - \frac{4}{\mu} \frac{L^2 q \eta^2}{|\mathcal{A}|} - [\frac{\eta}{4} - \frac{\eta^3 q}{2} \frac{L^2}{|\mathcal{A}|}]) \sum_{t=0}^{T} \| \sum_{t=0}^{T} \alpha^{(t-i)} \mathbf{d}_i \|^2, \tag{47}$$

where $(a)$ follows from Eqs. (46). Let $|\mathcal{A}| = q = \lceil \sqrt{n} \rceil$ and $\eta \leq \min\{\frac{1}{2\mu}, \frac{1}{8L}, \frac{\mu}{64L^2}\}$, we have $(\frac{1}{2}\eta - \frac{\mu}{8}\eta^2 - \frac{1}{2}L\eta^2 - \frac{4}{\mu} \frac{L^2 q \eta^2}{|\mathcal{A}|} - [\frac{\eta}{4} - \frac{\eta^3 q}{2} \frac{L^2}{|\mathcal{A}|}]) > \frac{\eta}{16} > 0$

Thus, we have

$$\sum_{t=0}^{T} \sum_{s \in [S]} \lambda_t^s \left[ f_s(\mathbf{x}_i) - f_s(\mathbf{x}_*) \right] \leq \frac{1}{2\eta} \left( (1 - \frac{3\mu\eta}{4}) \sum_{i=0}^{T} \|\mathbf{x}_i - \mathbf{x}_*\|^2 - \sum_{t=0}^{T} \|\mathbf{x}_{i+1} - \mathbf{x}_*\|^2 \right). \tag{48}$$

Then, we have

$$\mathbb{E}[\sum_{s \in [S]} \lambda_t^s \left[ f_s(\mathbf{x}_t) - f_s(\mathbf{x}_*) \right]] \leq \frac{1}{2\eta} \left( (1 - \frac{3\mu\eta}{4}) \mathbb{E}\|\mathbf{x}_t - \mathbf{x}_*\|^2 - \mathbb{E}\|\mathbf{x}_{t+1} - \mathbf{x}_*\|^2 \right). \tag{49}$$

Averaging using weight $w_t = (1 - \frac{3\mu\eta}{4})^{1-t}$ and using such weight to pick output $\mathbf{x}$, by using Lemma 1 in Karimireddy et al. (2020) with $\eta \geq \frac{1}{uR}$, we have

$$\mathbb{E}[\sum_{s \in [S]} \lambda_t^s \left[ f_s(\mathbf{x}_t) - f_s(\mathbf{x}_*) \right]] \leq \|\mathbf{x}_0 - \mathbf{x}_*\|^2 \mu \exp(-\frac{3\eta\mu T}{4}) \tag{50}$$

$$= \mathcal{O}(\mu \exp(-\mu T)). \tag{51}$$

Then we have the convergence rate $\mathbb{E}[\sum_{s\in[S]} \lambda_t^s [f_s(\mathbf{x}_t) - f_s(\mathbf{x}_*)]] = \mathcal{O}(\mu \exp(-\mu T))$. the total sample complexity can be calculated as: $\lceil \frac{T}{q} \rceil n + T \cdot |\mathcal{A}| \leq \frac{T+q}{q} n + T\sqrt{n} = T\sqrt{n} + n + T\sqrt{n} = O(n + \sqrt{n} \ln(\mu/\epsilon))$. Thus, the overall sample complexity is $\mathcal{O}(n + \sqrt{n} \ln(\mu/\epsilon))$. This completes the proof.

$\square$

## C  PROOF OF CONVERGENCE OF STIMULUS$^+$

**Proof of Theorem. 5 [Part 1]**

*Proof.* Recall that $\mathcal{N}_s = \min\{c_\gamma \sigma^2 (\gamma_t)^{-1}, c_\epsilon \sigma^2 \epsilon^{-1}, n\}$. Then we have

$$\frac{I_{(\mathcal{N}_s < n)}}{\mathcal{N}_s} \leq \frac{1}{\min\{c_\epsilon \sigma^2(\epsilon)^{-1}, c_\gamma \sigma^2 (\gamma_t)^{-1}\}}$$
$$= \max\{\frac{\gamma_t}{c_\gamma \sigma^2}, \frac{\epsilon}{c_\epsilon \sigma^2}\} \leq \frac{\gamma_t}{c_\gamma \sigma^2} + \frac{\epsilon}{c_\epsilon \sigma^2}. \tag{52}$$

From Lemma. 2, we have

$$[f_s(\mathbf{x}_{t+1})] \overset{(a)}{\leq} [f_s(\mathbf{x}_t)] + \frac{\eta}{2}\|\nabla f_s(\mathbf{x}_t) - \mathbf{u}_t^s\|^2 - \frac{\eta}{4}\|\mathbf{d}_t\|^2$$

$$\overset{(b)}{\leq} [f_s(\mathbf{x}_t)] - \frac{\eta}{4}\|\mathbf{d}_t\|^2$$
$$+ \frac{\eta}{2}[\frac{L^2}{|\mathcal{A}|} \sum_{i=(n_t-1)q}^{t} \|\mathbf{x}_{i+1} - \mathbf{x}_i\|^2 + \|\nabla f_s(\mathbf{x}_{(n_t-1)q}) - \mathbf{u}_{(n_t-1)q}^s\|^2]$$

$$\overset{(c)}{\leq} [f_s(\mathbf{x}_t)] - \frac{\eta}{4}\|\mathbf{d}_t\|^2 + \frac{\eta}{2}[\frac{L^2}{|\mathcal{A}|} \sum_{i=(n_t-1)q}^{t} \eta^2\|\mathbf{d}_i\|^2 + \frac{I_{(\mathcal{N}_s < n)}}{\mathcal{N}_s}\sigma^2], \tag{53}$$

where $(a)$ follows from Lemma. 2. $(b)$ follows from Lemma. 1. (c) follows from the update rule shown in Eqs. (6).

Next, telescoping the above inequality over $t$ from $(n_t - 1)q$ to $t$ where $t \leq n_t q - 1$ and noting that for $(n_t - 1)q \leq j \leq n_t q - 1, n_j = n_t$, and aking expectation on both sides of the inequality in Eqs. (53),we obtain

$$\mathbb{E}[f_s(\mathbf{x}_{t+1})]$$
$$\overset{(a)}{\leq} \mathbb{E}[f_s(\mathbf{x}_{(n_t-1)q})] - \frac{\eta}{4} \sum_{j=(n_t-1)q}^{t} \mathbb{E}\|\mathbf{d}_j\|^2$$
$$+ \frac{\eta}{2}[\frac{L^2}{|\mathcal{A}|} \sum_{j=(n_t-1)q}^{t} \sum_{i=(n_t-1)q}^{j} \eta^2 \mathbb{E}\|\mathbf{d}_i\|^2 + \sum_{i=(n_t-1)q}^{t} \frac{I_{(\mathcal{N}_s<n)}}{\mathcal{N}_s}\sigma^2]$$
$$\overset{(b)}{\leq} \mathbb{E}[f_s(\mathbf{x}_{(n_t-1)q})] - \frac{\eta}{4} \sum_{j=(n_t-1)q}^{t} \mathbb{E}\|\mathbf{d}_j\|^2$$
$$+ \frac{\eta}{2} \sum_{i=(n_t-1)q}^{t} [\frac{L^2}{|\mathcal{A}|} \sum_{j=(n_t-1)q}^{t} \sum_{i=(n_t-1)q}^{t} \eta^2 \mathbb{E}\|\mathbf{d}_i\|^2] + \frac{\eta}{2} \sum_{i=(n_t-1)q}^{t} \frac{I_{(\mathcal{N}_s<n)}}{\mathcal{N}_s}\sigma^2$$

$$= \mathbb{E}[f_s(\mathbf{x}_{(n_t-1)q})] - \frac{\eta}{4} \sum_{j=(n_t-1)q}^{t} \mathbb{E}\|\mathbf{d}_j\|^2$$

$$+ \frac{\eta^3 q}{2} [\frac{L^2}{|\mathcal{A}|} \sum_{j=(n_t-1)q}^{t} \mathbb{E}\|\mathbf{d}_j\|^2] + \frac{\eta}{2} \sum_{i=(n_t-1)q}^{t} \frac{I_{(\mathcal{N}_s < n)}}{\mathcal{N}_s} \sigma^2$$

$$\overset{(c)}{=} \mathbb{E}[f_s(\mathbf{x}_{(n_t-1)q})] - [\frac{\eta}{4} - \frac{\eta^3 q}{2} \frac{L^2}{|\mathcal{A}|}] \sum_{j=(n_t-1)q}^{t} \mathbb{E}\|\mathbf{d}_j\|^2 + \frac{\eta}{2} \sum_{i=(n_t-1)q}^{t} (\frac{\gamma_i}{c_\gamma} + \frac{\epsilon}{c_\epsilon}), \quad (54)$$

where $(a)$ follows from Eqs. (53), $(b)$ extends $j$ to $t$. $(c)$ follows from Eqs. (52)

Recall that $\gamma_t = \frac{1}{q} \sum_{i=(n_t-1)q}^{t} \|\mathbf{d}_t\|^2$. Then, we have We continue the proof by further driving

$$\mathbb{E}[f_s(\mathbf{x}_T) - f_s(\mathbf{x}_0)]$$
$$= \mathbb{E}[([f_s(\mathbf{x}_q)] - [f_s(\mathbf{x}_0)]) + ([f_s(\mathbf{x}_{2q})] - [f_s(\mathbf{x}_q)]) + \cdots + ([f_s(\mathbf{x}_T)] - [f_s(\mathbf{x}_{(n_T-1)q})])]$$
$$\overset{(a)}{\leq} -[\frac{\eta}{4} - \frac{\eta^3 q}{2} \frac{L^2}{|\mathcal{A}|}] \sum_{t=0}^{T-1} \mathbb{E}\|\mathbf{d}_t\|^2 + \frac{\eta}{2} \sum_{t=0}^{T-1} (\frac{\mathbb{E}[\gamma_i]}{c_\gamma} + \frac{\epsilon}{c_\epsilon})$$
$$\overset{(b)}{\leq} -[\frac{\eta}{4} - \frac{\eta^3 q}{2} \frac{L^2}{|\mathcal{A}|} - \frac{\eta}{2c_\gamma}] \sum_{t=0}^{T-1} \mathbb{E}\|\mathbf{d}_t\|^2 + \frac{\eta}{2} T \frac{\epsilon}{c_\epsilon}, \quad (55)$$

where $(a)$ is from Eqs. (54). $(b)$ follows from $\gamma_t = \frac{1}{q} \sum_{i=(n_t-1)q}^{t} \|\mathbf{d}_t\|^2$.

Note that $[f_s(\mathbf{x}_{T+1})] \geq f_s^* \triangleq \inf_{\mathbf{x} \in \mathbb{R}^d} f_s(\mathbf{x})$. Let $c_\gamma > 4$. Hence, we have

$$[\frac{\eta}{8} - \frac{\eta^3 q}{2} \frac{L^2}{|\mathcal{A}|} - \frac{\eta}{2c_\gamma}] \sum_{t=0}^{T-1} \mathbb{E}\|\mathbf{d}_t\|^2 \leq \mathbb{E}[[f_s(\mathbf{x}_0)] - [f_s(\mathbf{x}_T)]] \leq \mathbb{E}[[f_s(\mathbf{x}_0)] - f_s^*] + \frac{\eta}{2} T \frac{\epsilon}{c_\epsilon}. \quad (56)$$

Based on the parameter setting $q = |\mathcal{A}| = \lceil \sqrt{n} \rceil$, we have

$$[\frac{\eta}{8} - \frac{\eta^3 L^2}{2} - \frac{\eta}{2c_\gamma}] \sum_{t=0}^{T-1} \mathbb{E}\|\mathbf{d}_t\|^2 \leq \mathbb{E}[[f_s(\mathbf{x}_0)] - f_s^*] + \frac{\eta}{2} T \frac{\epsilon}{c_\epsilon}. \quad (57)$$

Thus, we have

$$\frac{1}{T} \sum_{t=0}^{T-1} \mathbb{E}\|\mathbf{d}_t\|^2 \leq \frac{\mathbb{E}[[f_s(\mathbf{x}_0)] - f_s^*]}{[\frac{\eta}{8} - \frac{\eta^3 L^2}{2} - \frac{\eta}{2c_\gamma}]T} + \frac{\eta}{2} \frac{\epsilon}{c_\epsilon}. \quad (58)$$

Let $\eta \leq \frac{1}{4L}, c_\gamma \geq 8, c_\epsilon \geq \eta$, we have

Since $\frac{1}{T} \sum_{t=0}^{T-1} \mathbb{E}\|d_t\|^2$ is just common descent directions. According to Definition. 3 shown in the paper, the quantity to our interest is $\|\sum_{s \in [S]} \lambda_t^s \nabla f(\mathbf{x})\|^2$.

$$\frac{1}{T} \sum_{t=0}^{T-1} \mathbb{E}\|\sum_{s \in [S]} \lambda_t^s \nabla f_s(\mathbf{x}_t)\|^2 \overset{(a)}{\leq} (2SL^2\eta^2 + 2) \frac{1}{T} \sum_{t=0}^{T-1} \mathbb{E}\|\mathbf{d}_t\|^2 \quad (59)$$

where $(a)$ follows from Eqs. (21).

Then, we can conclude that

$$\frac{1}{T} \sum_{t=0}^{T-1} \mathbb{E}\|\sum_{s \in [S]} \lambda_t^s \nabla f_s(\mathbf{x}_t)\|^2 \overset{(a)}{\leq} (2SL^2\eta^2 + 2)(\frac{\mathbb{E}[[f_s(\mathbf{x}_0)] - f_s^*]}{[\frac{\eta}{8} - \frac{\eta^3 L^2}{2} - \frac{\eta}{2c_\gamma}]T} + \frac{\eta}{2} \frac{\epsilon}{c_\epsilon}), \quad (60)$$

where $(a)$ follows from Eqs. (21) and Eqs. 20.

Thus, we have

$$\frac{1}{T}\sum_{t=0}^{T-1}\mathbb{E}\|\sum_{s\in[S]}\lambda_t^s\nabla f_s(\mathbf{x}_t)\|^2 = \mathcal{O}(\frac{1}{T}). \tag{61}$$

The total sample complexity can be calculated as: $\lceil\frac{T}{q}\rceil n + T \cdot |\mathcal{A}| \leq \frac{T+q}{q}n + T\sqrt{n} = T\sqrt{n} + n + T\sqrt{n} = O(n + \sqrt{n}\epsilon^{-1})$. Thus, the overall sample complexity is $\mathcal{O}(n + \sqrt{n}\epsilon^{-1})$. This completes the proof.

$\square$

## C.1 Proof of Theorem. 6 [Part 1]

*Proof.*

$$f_s(\mathbf{x}_{t+1})$$
$$\overset{(a)}{\leq} f_s(\mathbf{x}_t) + \langle\nabla f_s(\mathbf{x}_t), -\eta\mathbf{d}_t\rangle + \frac{1}{2}L\|\eta\mathbf{d}_t\|^2$$
$$\overset{(b)}{\leq} f_s(\mathbf{x}_*) + \langle\nabla f_s(\mathbf{x}_t), \mathbf{x}_t - \mathbf{x}_*\rangle - \frac{\mu}{2}\|\mathbf{x}_t - \mathbf{x}_*\|^2 + \langle\nabla f_s(\mathbf{x}_t), -\eta\mathbf{d}_t\rangle + \frac{1}{2}L\|\eta\mathbf{d}_t\|^2$$
$$= f_s(\mathbf{x}_*) + \langle\nabla f_s(\mathbf{x}_t), \mathbf{x}_t - \mathbf{x}_* - \eta\mathbf{d}_t\rangle - \frac{\mu}{2}\|\mathbf{x}_t - \mathbf{x}_*\|^2 + \frac{1}{2}L\|\eta\mathbf{d}_t\|^2$$
$$= f_s(\mathbf{x}_*) + \langle\nabla f_s(\mathbf{x}_t) - \mathbf{u}_t^s, \mathbf{x}_t - \mathbf{x}_* - \eta\mathbf{d}_t\rangle + \langle\mathbf{u}_t^s, \mathbf{x}_t - \mathbf{x}_* - \eta\mathbf{d}_t\rangle$$
$$\quad - \frac{\mu}{2}\|\mathbf{x}_t - \mathbf{x}_*\|^2 + \frac{1}{2}L\|\eta\mathbf{d}_t\|^2$$
$$\overset{(c)}{\leq} f_s(\mathbf{x}_*) + \frac{1}{2\delta}\|\nabla f_s(\mathbf{x}_t) - \mathbf{u}_t^s\|^2 + \frac{\delta}{2}\|\mathbf{x}_t - \mathbf{x}_* - \eta\mathbf{d}_t\|^2 + \langle\mathbf{u}_t^s, \mathbf{x}_t - \mathbf{x}_* - \eta\mathbf{d}_t\rangle$$
$$\quad - \frac{\mu}{2}\|\mathbf{x}_t - \mathbf{x}_*\|^2 + \frac{1}{2}L\|\eta\mathbf{d}_t\|^2$$
$$\overset{(d)}{\leq} f_s(\mathbf{x}_*) + \frac{1}{2\delta}\|\nabla f_s(\mathbf{x}_t) - \mathbf{u}_t^s\|^2 + \delta\|\mathbf{x}_t - \mathbf{x}_*\|^2 + \delta\|\eta\mathbf{d}_t\|^2$$
$$\quad + \langle\mathbf{u}_t^s, \mathbf{x}_t - \mathbf{x}_* - \eta\mathbf{d}_t\rangle - \frac{\mu}{2}\|\mathbf{x}_t - \mathbf{x}_*\|^2 + \frac{1}{2}L\|\eta\mathbf{d}_t\|^2, \tag{62}$$

where $(a)$ follows from $L$-smoothness, $(b)$ follows from $\mu$-strongly convexity. $(c)$ follows from Young's inequality, and $(d)$ follows from triangle inequality.

Then, we have

$$\sum_{s\in[S]}\lambda_t^s\left[f_s(\mathbf{x}_{t+1}) - f_s(\mathbf{x}_*)\right] \tag{63}$$

$$\overset{(a)}{\leq} \frac{1}{2\delta}\|\nabla f_s(\mathbf{x}_t) - \mathbf{u}_t^s\|^2 + \delta\|\mathbf{x}_t - \mathbf{x}_*\|^2 + \delta\|\eta\mathbf{d}_t\|^2$$
$$+ \left\langle\sum_{s\in[S]}\lambda_t^s\mathbf{u}_t^s, \mathbf{x}_t - \mathbf{x}_*\right\rangle - \frac{\mu}{2}\|\mathbf{x}_t - \mathbf{x}_*\|^2 + \left\langle\sum_{s\in[S]}\lambda_t^s\mathbf{u}_t^s, -\eta\mathbf{d}_t\right\rangle + \frac{1}{2}L\|\eta\mathbf{d}_t\|^2 \tag{64}$$

$$= \frac{1}{2\delta}\|\nabla f_s(\mathbf{x}_t) - \mathbf{u}_t^s\|^2 + \delta\|\mathbf{x}_t - \mathbf{x}_*\|^2 + \delta\|\eta\mathbf{d}_t\|^2$$
$$+ \left\langle\sum_{s\in[S]}\lambda_t^s\mathbf{u}_t^s, \mathbf{x}_t - \mathbf{x}_* - \eta\mathbf{d}_t\right\rangle - \frac{\mu}{2}\|\mathbf{x}_t - \mathbf{x}_*\|^2 + \frac{1}{2}L\|\eta\mathbf{d}_t\|^2 \tag{65}$$

$$\overset{(b)}{\leq} \frac{1}{2\delta}\|\nabla f_s(\mathbf{x}_t) - \mathbf{u}_t^s\|^2 + \delta\|\mathbf{x}_t - \mathbf{x}_*\|^2 + \delta\|\eta\mathbf{d}_t\|^2$$

$$+ \langle \mathbf{d}_t, \mathbf{x}_t - \mathbf{x}_* - \eta \mathbf{d}_t \rangle - \frac{\mu}{2} \|\mathbf{x}_t - \mathbf{x}_*\|^2 + \frac{1}{2} L \|\eta \mathbf{d}_t\|^2 \tag{66}$$

$$= \langle \mathbf{d}_t, \mathbf{x}_t - \mathbf{x}_* \rangle - \eta \|\mathbf{d}_t\|^2 - \frac{\mu}{2} \|\mathbf{x}_t - \mathbf{x}_*\|^2 + \frac{1}{2} L \eta^2 \|\mathbf{d}_t\|^2$$

$$+ \frac{1}{2\delta} \|\nabla f_s(\mathbf{x}_t) - \mathbf{u}_t^s\|^2 + \delta \|\mathbf{x}_t - \mathbf{x}_*\|^2 + \delta \|\eta \mathbf{d}_t\|^2$$

$$\overset{(c)}{=} \frac{1}{2\eta} \left( \|\mathbf{x}_t - \mathbf{x}_*\|^2 - \|\mathbf{x}_{t+1} - \mathbf{x}_*\|^2 \right) - \frac{1}{2} \eta \|\mathbf{d}_t\|^2 - \frac{\mu}{2} \|\mathbf{x}_t - \mathbf{x}_*\|^2 + \frac{1}{2} L \eta^2 \|\mathbf{d}_t\|^2$$

$$+ \frac{4}{\mu} \|\nabla f_s(\mathbf{x}_t) - \mathbf{u}_t^s\|^2 + \frac{\mu}{8} \|\mathbf{x}_t - \mathbf{x}_*\|^2 + \frac{\mu}{8} \|\eta \mathbf{d}_t\|^2 \tag{67}$$

$$= \frac{1}{2\eta} \left( (1 - \frac{3\mu\eta}{4}) \|\mathbf{x}_t - \mathbf{x}_*\|^2 - \|\mathbf{x}_{t+1} - \mathbf{x}_*\|^2 \right) - (\frac{1}{2}\eta - \frac{\mu}{8}\eta^2 - \frac{1}{2}L\eta^2) \|\mathbf{d}_t\|^2$$

$$+ \frac{4}{\mu} \|\nabla f_s(\mathbf{x}_t) - \mathbf{u}_t^s\|^2 \tag{68}$$

$$\overset{(d)}{\leq} \frac{1}{2\eta} \left( (1 - \frac{3\mu\eta}{4}) \|\mathbf{x}_t - \mathbf{x}_*\|^2 - \|\mathbf{x}_{t+1} - \mathbf{x}_*\|^2 \right) - (\frac{1}{2}\eta - \frac{\mu}{8}\eta^2 - \frac{1}{2}L\eta^2) \|\mathbf{d}_t\|^2$$

$$+ \frac{4}{\mu} (\frac{L^2}{|\mathcal{A}|} \sum_{i=(n_t-1)q}^{t} \|\mathbf{x}_{i+1} - \mathbf{x}_i\|^2 + \|\nabla f_s(\mathbf{x}_{(n_t-1)q}) - \mathbf{u}_{(n_t-1)q}^s\|^2) \tag{69}$$

$$\overset{(f)}{\leq} \frac{1}{2\eta} \left( (1 - \frac{3\mu\eta}{4}) \|\mathbf{x}_t - \mathbf{x}_*\|^2 - \|\mathbf{x}_{t+1} - \mathbf{x}_*\|^2 \right) - (\frac{1}{2}\eta - \frac{\mu}{8}\eta^2 - \frac{1}{2}L\eta^2) \|\mathbf{d}_t\|^2$$

$$+ \frac{4}{\mu} (\frac{L^2}{|\mathcal{A}|} \sum_{i=(n_t-1)q}^{t} \|\mathbf{x}_{i+1} - \mathbf{x}_i\|^2) + \frac{\mu}{4} \frac{I_{(\mathcal{N}_s < n)}}{\mathcal{N}_s} \sigma^2. \tag{70}$$

where $(a)$ follows from Eqs.(62). (b) follows from the definition $\mathbf{d}_t = \sum_{s \in [S]} \lambda_t^s \mathbf{u}_t^s$ as shown in Line 14 in Algorithm. 1. $(c)$ is because $\|\mathbf{x}_t - \mathbf{x}_*\|^2 - \|\mathbf{x}_{t+1} - \mathbf{x}_*\|^2 = -\eta^2 \|\mathbf{d}_t\|^2 + 2 \langle \eta \mathbf{d}_t, \mathbf{x}_t - \mathbf{x}_* \rangle$. $(d)$ is from Lemma. 1 and we choose $\delta = \frac{\mu}{8}$. $(e)$ is from Eqs. (52).

Next, telescoping the above inequality over $t$ from $(n_t - 1) q$ to $t$ where $t \leq n_t q - 1$ and noting that for $(n_t - 1) q \leq j \leq n_t q - 1, n_j = n_t$, we obtain

$$\sum_{i=(n_t-1)q}^{t} \sum_{s \in [S]} \lambda_t^s [f_s(\mathbf{x}_{i+1}) - f_s(\mathbf{x}_*)]$$

$$\overset{(a)}{\leq} \frac{1}{2\eta} \left( (1 - \frac{3\mu\eta}{4}) \sum_{i=(n_t-1)q}^{t} \|\mathbf{x}_i - \mathbf{x}_*\|^2 - \sum_{i=(n_t-1)q}^{t} \|\mathbf{x}_{i+1} - \mathbf{x}_*\|^2 \right)$$

$$- (\frac{1}{2}\eta - \frac{\mu}{8}\eta^2 - \frac{1}{2}L\eta^2) \sum_{i=(n_t-1)q}^{t} \|\mathbf{d}_i\|^2 + \frac{4}{\mu} (\frac{L^2}{|\mathcal{A}|} \sum_{j=(n_t-1)q}^{t} \sum_{i=(n_j-1)q}^{j} \|\mathbf{x}_{i+1} - \mathbf{x}_i\|^2)$$

$$+ \frac{\mu S}{4} \sum_{i=(n_t-1)q}^{t} \frac{I_{(\mathcal{N}_s < n)}}{\mathcal{N}_s} \sigma^2$$

$$\overset{(b)}{\leq} \frac{1}{2\eta} \left( (1 - \frac{3\mu\eta}{4}) \sum_{i=(n_t-1)q}^{t} \|\mathbf{x}_i - \mathbf{x}_*\|^2 - \sum_{i=(n_t-1)q}^{t} \|\mathbf{x}_{i+1} - \mathbf{x}_*\|^2 \right)$$

$$- (\frac{1}{2}\eta - \frac{\mu}{8}\eta^2 - \frac{1}{2}L\eta^2) \sum_{i=(n_t-1)q}^{t} \|\mathbf{d}_i\|^2 + \frac{4}{\mu} (\frac{L^2}{|\mathcal{A}|} \sum_{j=(n_t-1)q}^{t} \sum_{i=(n_t-1)q}^{t} \|\mathbf{x}_{i+1} - \mathbf{x}_i\|^2)$$

$$+ \frac{\mu}{4} \sum_{i=(n_t-1)q}^{t} \frac{I_{(\mathcal{N}_s < n)}}{\mathcal{N}_s} \sigma^2$$

$$\overset{(c)}{\leq} \frac{1}{2\eta} \left( (1 - \frac{3\mu\eta}{4}) \sum_{i=(n_t-1)q}^{t} \|\mathbf{x}_i - \mathbf{x}_*\|^2 - \sum_{i=(n_t-1)q}^{t} \|\mathbf{x}_{i+1} - \mathbf{x}_*\|^2 \right)$$

$$- (\frac{1}{2}\eta - \frac{\mu}{8}\eta^2 - \frac{1}{2}L\eta^2 - \frac{4}{\mu}\frac{L^2 q\eta^2}{|\mathcal{A}|}) \sum_{i=(n_t-1)q}^{t} \|\mathbf{d}_i\|^2)$$

$$+ \frac{\mu}{4} \sum_{i=(n_t-1)q}^{t} (\frac{[\gamma_i]}{c_\gamma} + \frac{\epsilon}{c_\epsilon}), \tag{71}$$

where $(a)$ follows from Eqs. (63) and the fact that $\lambda_t^s \leq 1 \forall s \in [S]$. $(b)$ extends $j$ to $t$. $(c)$ is because $t - (n_t - 1)q \geq q$. We continue the proof by further driving

$$\sum_{t=0}^{T} \sum_{s \in [S]} \lambda_t^s [f_s(\mathbf{x}_{i+1}) - f_s(\mathbf{x}_*)]$$

$$= \sum_{i=0}^{q} \sum_{s \in [S]} \lambda_t^s [f_s(\mathbf{x}_{i+1}) - f_s(\mathbf{x}_*)] + \sum_{i=q}^{2q} \sum_{s \in [S]} \lambda_t^s [f_s(\mathbf{x}_{i+1}) - f_s(\mathbf{x}_*)] +$$

$$\cdot + \sum_{i=(n_T-1)q}^{T} \sum_{s \in [S]} \lambda_t^s [f_s(\mathbf{x}_{i+1}) - f_s(\mathbf{x}_*)]$$

$$\overset{(a)}{\leq} \frac{1}{2\eta} \left( (1 - \frac{3\mu\eta}{4}) \sum_{i=0}^{T} \|\mathbf{x}_i - \mathbf{x}_*\|^2 - \sum_{t=0}^{T} \|\mathbf{x}_{i+1} - \mathbf{x}_*\|^2 \right)$$

$$- (\frac{1}{2}\eta - \frac{\mu}{8}\eta^2 - \frac{1}{2}L\eta^2 - \frac{4}{\mu}\frac{L^2 q\eta^2}{|\mathcal{A}|} + \frac{\mu}{4c_\gamma}) \sum_{t=0}^{T} \|\mathbf{d}_i\|^2 + \frac{\mu}{4}T\frac{\epsilon}{c_\epsilon}, \tag{72}$$

where $(a)$ follows from Eqs. (71) and $\gamma_t = \frac{1}{q} \sum_{i=(n_t-1)q}^{t} \|\mathbf{d}_t\|^2$.

Next, we have

$$\sum_{t=0}^{T} \sum_{s \in [S]} \lambda_t^s [f_s(\mathbf{x}_i) - f_s(\mathbf{x}_*)]$$

$$= \sum_{t=0}^{T} \sum_{s \in [S]} \lambda_t^s [f_s(\mathbf{x}_{i+1}) - f_s(\mathbf{x}_*) - f_s(\mathbf{x}_{i+1}) + f_s(\mathbf{x}_i)]$$

$$= \sum_{t=0}^{T} \sum_{s \in [S]} \lambda_t^s [f_s(\mathbf{x}_{i+1}) - f_s(\mathbf{x}_*)] + \sum_{t=0}^{T} \sum_{s \in [S]} \lambda_t^s |f_s(\mathbf{x}_{i+1}) - f_s(\mathbf{x}_i)|$$

$$\overset{(a)}{\leq} \frac{1}{2\eta} \left( (1 - \frac{3\mu\eta}{4}) \sum_{i=0}^{T} \|\mathbf{x}_i - \mathbf{x}_*\|^2 - \sum_{t=0}^{T} \|\mathbf{x}_{i+1} - \mathbf{x}_*\|^2 \right)$$

$$- (\frac{1}{2}\eta - \frac{\mu}{8}\eta^2 - \frac{1}{2}L\eta^2 - \frac{4}{\mu}\frac{L^2 q\eta^2}{|\mathcal{A}|} - [\frac{\eta}{4} - \frac{\eta^3 q}{2}\frac{L^2}{|\mathcal{A}|}] - \frac{\mu}{4c_\gamma}) \sum_{t=0}^{T} \|\mathbf{d}_i\|^2 + \frac{\mu}{4}T\frac{\epsilon}{c_\epsilon}, \tag{73}$$

where $(a)$ follows from Eqs. (72).

Let $|\mathcal{A}| = q = \lceil \sqrt{n} \rceil$ and $\eta \leq \min\{\frac{1}{2\mu}, \frac{1}{8L}, \frac{\mu}{64L^2}\}, c_\gamma \geq \frac{8\mu}{\eta}$, we have $(\frac{1}{2}\eta - \frac{\mu}{8}\eta^2 - \frac{1}{2}L\eta^2 - \frac{4}{\mu}\frac{L^2 q\eta^2}{|\mathcal{A}|} - [\frac{\eta}{4} - \frac{\eta^3 q}{2}\frac{L^2}{|\mathcal{A}|}] - \frac{\mu}{4c_\gamma}) > \frac{\eta}{32} > 0$

Thus, we have

$$\sum_{t=0}^{T} \sum_{s \in [S]} \lambda_t^s [f_s(\mathbf{x}_i) - f_s(\mathbf{x}_*)] \leq \frac{1}{2\eta} \left( (1 - \frac{3\mu\eta}{4}) \sum_{i=0}^{T} \|\mathbf{x}_i - \mathbf{x}_*\|^2 - \sum_{t=0}^{T} \|\mathbf{x}_{i+1} - \mathbf{x}_*\|^2 \right). \tag{74}$$

Then, we have

$$\mathbb{E}[\sum_{s\in[S]} \lambda_t^s [f_s(\mathbf{x}_t) - f_s(\mathbf{x}_*)]] \le \frac{1}{2\eta} \left( (1 - \frac{3\mu\eta}{4})\mathbb{E}\|\mathbf{x}_t - \mathbf{x}_*\|^2 - \mathbb{E}\|\mathbf{x}_{t+1} - \mathbf{x}_*\|^2 \right) + \frac{\mu}{4}T\frac{\epsilon}{c_\epsilon}.$$
(75)

Averaging using weight $w_t = (1 - \frac{3\mu\eta}{4})^{1-t}$ and using such weight to pick output $\mathbf{x}$. By using Lemma 1 in Karimireddy et al. (2020) with $\eta \ge \frac{1}{uR}, c_\epsilon > \frac{\mu}{2}$, we have

$$\mathbb{E}[\sum_{s\in[S]} \lambda_t^s [f_s(\mathbf{x}_t) - f_s(\mathbf{x}_*)]] \le \|\mathbf{x}_0 - \mathbf{x}_*\|^2\mu \exp(-\frac{3\eta\mu T}{4}) + \frac{\mu}{4}T\frac{\epsilon}{c_\epsilon}$$
(76)

$$= \mathcal{O}(\mu \exp(-\mu T)).$$
(77)

Then we have the convergence rate $\mathbb{E}[\sum_{s\in[S]} \lambda_t^s [f_s(\mathbf{x}_t) - f_s(\mathbf{x}_*)]] = \mathcal{O}(\mu \exp(-\mu T))$.

The total sample complexity can be calculated as: $\lceil \frac{T}{q} \rceil n + T \cdot |\mathcal{A}| \le \frac{T+q}{q}n + T\sqrt{n} = T\sqrt{n} + n + T\sqrt{n} = O(n + \sqrt{n}\ln(\mu/\epsilon))$. Thus, the overall sample complexity is $\mathcal{O}(n + \sqrt{n}\ln(\mu/\epsilon))$. This completes the proof. □

## D  Proof of convergence of STIMULUS-M⁺

**Proof of Theorem. 5 [Part 2]**

*Proof.* From Lemma. 3, we have

$$[f_s(\mathbf{x}_{t+1})]$$

$$\stackrel{(a)}{\le} [f_s(\mathbf{x}_t)] + \frac{\eta}{2} \sum_{i=0}^{t} \alpha^{(t-i)}\|\nabla f_s(\mathbf{x}_i) - \mathbf{u}_i^s\|^2 - \frac{1}{2}\eta \sum_{i=0}^{t} \alpha^{(t-i)}\|\mathbf{d}_i\|^2 + \frac{1}{2}L\|\mathbf{x}_{t+1} - \mathbf{x}_t\|^2$$

$$\stackrel{(b)}{\le} [f_s(\mathbf{x}_t)] - \frac{1}{2}\eta \sum_{i=0}^{t} \alpha^{(t-i)}\|\mathbf{d}_i\|^2 + \frac{1}{2}L\|\mathbf{x}_{t+1} - \mathbf{x}_t\|^2$$

$$+ \frac{\eta}{2} \sum_{j=0}^{t} \alpha^{(t-j)}[\frac{L^2}{|\mathcal{A}|} \sum_{i=(n_t-1)q}^{j} \|\mathbf{x}_{i+1} - \mathbf{x}_i\|^2 + \|\nabla f_s(\mathbf{x}_{(n_t-1)q}) - \mathbf{u}_{(n_t-1)q}^s\|^2]$$

$$\stackrel{(c)}{\le} [f_s(\mathbf{x}_t)] - \frac{1}{2}\eta \sum_{i=0}^{t} \alpha^{(t-i)}\|\mathbf{d}_i\|^2 + \frac{1}{2}L\|\mathbf{x}_{t+1} - \mathbf{x}_t\|^2 + \frac{\eta}{2} \sum_{j=0}^{t} \alpha^{(t-j)}[\frac{L^2}{|\mathcal{A}|} \sum_{i=(n_t-1)q}^{j} \|\mathbf{x}_{i+1} - \mathbf{x}_i\|^2]$$

$$+ \frac{\eta}{2} \sum_{i=0}^{t} \alpha^{(t-i)}(\frac{\gamma_i}{c_\gamma} + \frac{\epsilon}{c_\epsilon}),$$
(78)

where $(a)$ follows from Lemma 3. $(b)$ follows from Lemma. 1. $(c)$ follows from Eqs. (52).

Next, telescoping the above inequality over $t$ from $(n_t - 1)q$ to $t$ where $t \le n_tq - 1$ and noting that for $(n_t - 1)q \le j \le n_tq - 1, n_j = n_t$ and let $\eta \le \frac{1}{4L}$, we obtain

$$[f_s(\mathbf{x}_{t+1})]$$

$$\stackrel{(a)}{\le} [f_s(\mathbf{x}_{(n_t-1)q})] - \frac{\eta}{2} \sum_{j=(n_t-1)q}^{t} \sum_{i=0}^{j} \alpha^{(j-i)}\|\mathbf{d}_i\|^2 + \frac{1}{2}L \sum_{i=(n_t-1)q}^{t} \|\mathbf{x}_{i+1} - \mathbf{x}_i\|^2$$

$$+ \frac{\eta}{2} \sum_{j=(n_t-1)q}^{t} \sum_{i=0}^{j} \alpha^{(j-i)} [\frac{L^2}{|\mathcal{A}|} \sum_{r=(n_t-1)q}^{i} \|\mathbf{x}_{r+1} - \mathbf{x}_r\|^2]$$

$$+ \frac{\eta}{2} \sum_{j=(n_t-1)q}^{t} \sum_{i=0}^{j} \alpha^{(j-i)} (\frac{\lfloor \gamma_i \rfloor}{c_\gamma} + \frac{\epsilon}{c_\epsilon})$$

$$\overset{(b)}{\leq} [f_s(\mathbf{x}_{(n_t-1)q})] - \frac{\eta}{2} \sum_{j=(n_t-1)q}^{t} \sum_{i=0}^{j} \alpha^{(j-i)} \|\mathbf{d}_i\|^2 + \frac{1}{2} L \sum_{i=(n_t-1)q}^{t} \|\mathbf{x}_{i+1} - \mathbf{x}_i\|^2$$

$$+ \frac{\eta}{2} \sum_{j=(n_t-1)q}^{t} \sum_{i=0}^{j} \alpha^{(j-i)} [\frac{L^2}{|\mathcal{A}|} q \|\mathbf{x}_{j+1} - \mathbf{x}_j\|^2]$$

$$+ \frac{\eta}{2} \sum_{j=(n_t-1)q}^{t} \sum_{i=0}^{j} \alpha^{(j-i)} (\frac{\lfloor \gamma_i \rfloor}{c_\gamma} + \frac{\epsilon}{c_\epsilon})$$

$$\overset{(c)}{=} [f_s(\mathbf{x}_{(n_t-1)q})] - \frac{\eta}{2} \sum_{j=(n_t-1)q}^{t} \sum_{i=0}^{j} \alpha^{(j-i)} \|\mathbf{d}_i\|^2 + \frac{1}{2} L \sum_{i=(n_t-1)q}^{t} \|\mathbf{x}_{i+1} - \mathbf{x}_i\|^2$$

$$+ \frac{\eta}{2} \sum_{j=(n_t-1)q}^{t} \sum_{i=0}^{j} \alpha^{(j-i)} [L^2 \|\mathbf{x}_{j+1} - \mathbf{x}_j\|^2]$$

$$+ \frac{\eta}{2} \sum_{j=(n_t-1)q}^{t} \sum_{i=0}^{j} \alpha^{(j-i)} (\frac{\lfloor \gamma_i \rfloor}{c_\gamma} + \frac{\epsilon}{c_\epsilon})$$

$$\overset{(d)}{\leq} [f_s(\mathbf{x}_{(n_t-1)q})] - \frac{\eta}{2} \sum_{j=(n_t-1)q}^{t} \sum_{i=0}^{j} \alpha^{(j-i)} \|\mathbf{d}_i\|^2 + \frac{1}{2} L \sum_{i=(n_t-1)q}^{t} \|\mathbf{x}_{i+1} - \mathbf{x}_i\|^2$$

$$+ \frac{\eta}{2} \sum_{j=(n_t-1)q}^{t} \sum_{i=0}^{j} \alpha^{(j-i)} [L^2 \|\eta \sum_{r=0}^{j} \alpha^{(j-r)} \mathbf{d}_r\|^2]$$

$$+ \frac{\eta}{2} \sum_{j=(n_t-1)q}^{t} \sum_{i=0}^{j} \alpha^{(j-i)} (\frac{\lfloor \gamma_i \rfloor}{c_\gamma} + \frac{\epsilon}{c_\epsilon})$$

$$= [f_s(\mathbf{x}_{(n_t-1)q})] - \frac{\eta}{2} \sum_{j=(n_t-1)q}^{t} \sum_{i=0}^{j} \alpha^{(j-i)} \|\mathbf{d}_i\|^2 + \frac{1}{2} L \sum_{i=(n_t-1)q}^{t} \|\mathbf{x}_{i+1} - \mathbf{x}_i\|^2$$

$$+ \frac{\eta}{2} \sum_{j=(n_t-1)q}^{t} \sum_{i=0}^{j} \alpha^{3(j-i)} [L^2 \eta^2 \|\mathbf{d}_i\|^2]$$

$$+ \frac{\eta}{2} \sum_{j=(n_t-1)q}^{t} \sum_{i=0}^{j} \alpha^{(j-i)} (\frac{\lfloor \gamma_i \rfloor}{c_\gamma} + \frac{\epsilon}{c_\epsilon})$$

$$\overset{(e)}{\leq} [f_s(\mathbf{x}_{(n_t-1)q})] - \frac{\eta}{2} \sum_{j=(n_t-1)q}^{t} \sum_{i=0}^{j} \alpha^{(j-i)} \|\mathbf{d}_i\|^2 + \frac{1}{2} L \sum_{i=(n_t-1)q}^{t} \|\mathbf{x}_{i+1} - \mathbf{x}_i\|^2$$

$$+ \frac{\eta}{2} \sum_{j=(n_t-1)q}^{t} \sum_{i=0}^{j} \alpha^{(j-i)} [L^2 \eta^2 \|\mathbf{d}_i\|^2]$$

$$+ \frac{\eta}{2} \sum_{j=(n_t-1)q}^{t} \sum_{i=0}^{j} \alpha^{(j-i)} (\frac{\lfloor \gamma_i \rfloor}{c_\gamma} + \frac{\epsilon}{c_\epsilon})$$

$$
\overset{(f)}{\leq} [f_s(\mathbf{x}_{(n_t-1)q})] - \frac{\eta}{4} \sum_{j=(n_t-1)q}^{t} \sum_{i=0}^{j} \alpha^{(j-i)} \|\mathbf{d}_i\|^2 + \frac{1}{2}L \sum_{i=(n_t-1)q}^{t} \|\mathbf{x}_{i+1} - \mathbf{x}_i\|^2
$$

$$
+ \frac{\eta}{2} \sum_{j=(n_t-1)q}^{t} \sum_{i=0}^{j} \alpha^{(j-i)} \left( \frac{[\gamma_i]}{c_\gamma} + \frac{\epsilon}{c_\epsilon} \right)
$$

$$
\overset{(g)}{\leq} [f_s(\mathbf{x}_{(n_t-1)q})] - \frac{\eta}{4} \sum_{j=(n_t-1)q}^{t} \sum_{i=0}^{j} \alpha^{(j-i)} \|\mathbf{d}_i\|^2 + \frac{1}{2}L \sum_{j=(n_t-1)q}^{t} \|\eta \sum_{i=0}^{j} \alpha^{(j-i)} \mathbf{d}_j\|^2
$$

$$
+ \frac{\eta}{2} \sum_{j=(n_t-1)q}^{t} \sum_{i=0}^{j} \alpha^{(j-i)} \left( \frac{[\gamma_i]}{c_\gamma} + \frac{\epsilon}{c_\epsilon} \right)
$$

$$
\overset{(h)}{\leq} [f_s(\mathbf{x}_{(n_t-1)q})] - \frac{\eta}{8} \sum_{j=(n_t-1)q}^{t} \sum_{i=0}^{j} \alpha^{(j-i)} \|\mathbf{d}_i\|^2 + \frac{\eta}{2} \sum_{j=(n_t-1)q}^{t} \sum_{i=0}^{j} \alpha^{(j-i)} \left( \frac{[\gamma_i]}{c_\gamma} + \frac{\epsilon}{c_\epsilon} \right), \quad (79)
$$

where $(a)$ follows from Eqs. (78). $(b)$ follows from $i \leq n_t q$. $(c)$ follows from $q = |\mathcal{A}| = \lceil \sqrt{n} \rceil$. $(d)$ and $(g)$ follow from the update rule of $\mathbf{x}_t$ shown in Line 19 in Algorithm. 1. $(e)$ follows from $0 < \alpha < 1$, then we have $\alpha^2(j-i) < \alpha^{(j-i)}$. $(f)$ and $(h)$ follow from $\eta \leq \frac{1}{4L}$ Recall that $\gamma_t = \frac{1}{q} \sum_{i=(n_t-1)q}^{t} \|\mathbf{d}_t\|^2$. Then, we have

$$
\mathbb{E}[f_s(\mathbf{x}_T)] - [f_s(\mathbf{x}_0)]
$$
$$
= \mathbb{E}([f_s(\mathbf{x}_q)] - [f_s(\mathbf{x}_0)]) + ([f_s(\mathbf{x}_{2q})] - [f_s(\mathbf{x}_q)]) + \cdot + ([f_s(\mathbf{x}_T)] - [f_s(\mathbf{x}_{(n_T-1)q})])
$$
$$
\overset{(a)}{\leq} -[\frac{\eta}{8}] \sum_{t=0}^{T-1} \sum_{i=0}^{j} \alpha^{(j-i)} \mathbb{E}\|\mathbf{d}_t\|^2 + \frac{\eta}{2c_\gamma} \sum_{t=0}^{T-1} \sum_{i=0}^{j} \alpha^{(j-i)} \mathbb{E}\|\mathbf{d}_t\|^2 + \frac{\eta}{2}Tq\frac{\epsilon}{c_\epsilon}
$$
$$
\overset{(b)}{\leq} -[\frac{\eta}{16}] \sum_{t=0}^{T-1} \sum_{i=0}^{j} \alpha^{(j-i)} \mathbb{E}\|\mathbf{d}_t\|^2 + \frac{\eta}{2}Tq\frac{\epsilon}{c_\epsilon}
$$
$$
\overset{(c)}{\leq} -[\frac{\eta}{16}] \sum_{t=0}^{T-1} \mathbb{E}\|\mathbf{d}_t\|^2 + \frac{\eta}{2}Tq\frac{\epsilon}{c_\epsilon}, \quad (80)
$$

where $(a)$ follows from $c_\gamma \geq 8$, $(c)$ follows from $0 < \alpha < 1$.

Note that $[f_s(\mathbf{x}_{T+1})] \geq f_s^* \triangleq \inf_{\mathbf{x} \in \mathbb{R}^d} f_s(\mathbf{x})$. Hence, we have

$$
[\frac{\eta}{16}] \sum_{t=0}^{T-1} \|\mathbf{d}_t\|^2 \leq [[f_s(\mathbf{x}_0)] - [f_s(\mathbf{x}_T)]] \leq [[f_s(\mathbf{x}_0)] - f_s^*]. \quad (81)
$$

Based on the parameter setting $q^2 = |\mathcal{A}| = \sqrt{n}$, we have

$$
[\frac{\eta}{16}] \sum_{t=0}^{T-1} \|\mathbf{d}_t\|^2 \leq [[f_s(\mathbf{x}_0)] - f_s^*]. \quad (82)
$$

Thus, we have

$$
\frac{1}{T} \sum_{t=0}^{T-1} \|\mathbf{d}_t\|^2 \leq \frac{[[f_s(\mathbf{x}_0)] - f_s^*]}{[\frac{\eta}{16}]T}. \quad (83)
$$

Since $\frac{1}{T} \sum_{t=0}^{T-1} \mathbb{E}\|d_t\|^2$ is just common descent directions. According to Definition. 3 shown in the paper, the quantity to our interest is $\| \sum_{s \in [S]} \lambda_t^s \nabla f(\mathbf{x})\|^2$.

$$
\frac{1}{T} \sum_{t=0}^{T-1} \mathbb{E}\| \sum_{s \in [S]} \lambda_t^s \nabla f_s(\mathbf{x}_t)\|^2 \overset{(a)}{\leq} (2SL^2\eta^2 + 2)\frac{1}{T} \sum_{t=0}^{T-1} \mathbb{E}\|\mathbf{d}_t\|^2 \quad (84)
$$

where $(a)$ follows from Eqs. (21).

Then, we can conclude that

$$\frac{1}{T}\sum_{t=0}^{T-1}\mathbb{E}\|\sum_{s\in[S]}\lambda_t^s\nabla f_s(\mathbf{x}_t)\|^2 = \mathcal{O}(\frac{1}{T}).$$ (85)

The total sample complexity can be calculated as: $\lceil\frac{T}{q}\rceil n + T\cdot|\mathcal{A}| \leq \frac{T+q}{q}n + T\sqrt{n} = T\sqrt{n} + n + T\sqrt{n} = O(n + \sqrt{n}\epsilon^{-1})$. Thus, the overall sample complexity is $\mathcal{O}(n + \sqrt{n}\epsilon^{-1})$. This completes the proof.

$\square$

## D.1 Proof of Theorem. 6 [Part 2]

*Proof.*

$f_s(\mathbf{x}_{t+1})$

$\overset{(a)}{\leq} f_s(\mathbf{x}_t) + \left\langle \nabla f_s(\mathbf{x}_t), -\eta\sum_{t=0}^{T}\alpha^{(t-i)}\mathbf{d}_i \right\rangle + \frac{1}{2}L\|\eta\sum_{t=0}^{T}\alpha^{(t-i)}\mathbf{d}_i\|^2$

$\overset{(b)}{\leq} f_s(\mathbf{x}_*) + \langle \nabla f_s(\mathbf{x}_t), \mathbf{x}_t - \mathbf{x}_* \rangle - \frac{\mu}{2}\|\mathbf{x}_t - \mathbf{x}_*\|^2 + \left\langle \nabla f_s(\mathbf{x}_t), -\eta\sum_{t=0}^{T}\alpha^{(t-i)}\mathbf{d}_i \right\rangle$

$\quad + \frac{1}{2}L\|\eta\sum_{t=0}^{T}\alpha^{(t-i)}\mathbf{d}_i\|^2$

$= f_s(\mathbf{x}_*) + \left\langle \nabla f_s(\mathbf{x}_t), \mathbf{x}_t - \mathbf{x}_* - \eta\sum_{t=0}^{T}\alpha^{(t-i)}\mathbf{d}_i \right\rangle - \frac{\mu}{2}\|\mathbf{x}_t - \mathbf{x}_*\|^2 + \frac{1}{2}L\|\eta\sum_{t=0}^{T}\alpha^{(t-i)}\mathbf{d}_i\|^2$

$= f_s(\mathbf{x}_*) + \left\langle \nabla f_s(\mathbf{x}_t) - \mathbf{u}_t^s, \mathbf{x}_t - \mathbf{x}_* - \eta\sum_{t=0}^{T}\alpha^{(t-i)}\mathbf{d}_i \right\rangle + \left\langle \mathbf{u}_t^s, \mathbf{x}_t - \mathbf{x}_* - \eta\sum_{t=0}^{T}\alpha^{(t-i)}\mathbf{d}_i \right\rangle$

$\quad - \frac{\mu}{2}\|\mathbf{x}_t - \mathbf{x}_*\|^2 + \frac{1}{2}L\|\eta\sum_{t=0}^{T}\alpha^{(t-i)}\mathbf{d}_i\|^2$

$\overset{(c)}{\leq} f_s(\mathbf{x}_*) + \frac{1}{2\delta}\|\nabla f_s(\mathbf{x}_t) - \mathbf{u}_t^s\|^2 + \frac{\delta}{2}\|\mathbf{x}_t - \mathbf{x}_* - \eta\sum_{t=0}^{T}\alpha^{(t-i)}\mathbf{d}_i\|^2$

$\quad + \left\langle \mathbf{u}_t^s, \mathbf{x}_t - \mathbf{x}_* - \eta\sum_{t=0}^{T}\alpha^{(t-i)}\mathbf{d}_i \right\rangle$

$\quad - \frac{\mu}{2}\|\mathbf{x}_t - \mathbf{x}_*\|^2 + \frac{1}{2}L\|\eta\sum_{t=0}^{T}\alpha^{(t-i)}\mathbf{d}_i\|^2$

$\overset{(d)}{\leq} f_s(\mathbf{x}_*) + \frac{1}{2\delta}\|\nabla f_s(\mathbf{x}_t) - \mathbf{u}_t^s\|^2 + \delta\|\mathbf{x}_t - \mathbf{x}_*\|^2 + \delta\|\eta\sum_{t=0}^{T}\alpha^{(t-i)}\mathbf{d}_i\|^2$

$\quad + \left\langle \mathbf{u}_t^s, \mathbf{x}_t - \mathbf{x}_* - \eta\sum_{t=0}^{T}\alpha^{(t-i)}\mathbf{d}_i \right\rangle - \frac{\mu}{2}\|\mathbf{x}_t - \mathbf{x}_*\|^2 + \frac{1}{2}L\|\eta\sum_{t=0}^{T}\alpha^{(t-i)}\mathbf{d}_i\|^2,$ (86)

where $(a)$ follows from $L$-smoothness assumption, $(b)$ follows from $\mu$-strongly convex. $(c)$ and $(d)$ follow from the triangle inequality.

$$\sum_{s\in[S]}\lambda_t^s\left[f_s(\mathbf{x}_{t+1}) - f_s(\mathbf{x}_*)\right]$$ (87)

$$
\overset{(a)}{\leq} \frac{1}{2\delta} \sum_{s\in[S]} \lambda_t^s \|\nabla f_s(\mathbf{x}_t) - \mathbf{u}_t^s\|^2 + \delta\|\mathbf{x}_t - \mathbf{x}_*\|^2 + \delta\|\eta\sum_{t=0}^{T}\alpha^{(t-i)}\mathbf{d}_i\|^2
$$

$$
+ \left\langle \sum_{s\in[S]}\lambda_t^s\mathbf{u}_t^s, \mathbf{x}_t - \mathbf{x}_* \right\rangle - \frac{\mu}{2}\|\mathbf{x}_t - \mathbf{x}_*\|^2 + \left\langle \sum_{s\in[S]}\lambda_t^s\mathbf{u}_t^s, -\eta\sum_{t=0}^{T}\alpha^{(t-i)}\mathbf{d}_i \right\rangle
$$

$$
+ \frac{1}{2}L\|\eta\sum_{t=0}^{T}\alpha^{(t-i)}\mathbf{d}_i\|^2
$$

$$
= \frac{1}{2\delta}\sum_{s\in[S]}\lambda_t^s\|\nabla f_s(\mathbf{x}_t) - \mathbf{u}_t^s\|^2 + \delta\|\mathbf{x}_t - \mathbf{x}_*\|^2 + \delta\|\eta\sum_{t=0}^{T}\alpha^{(t-i)}\mathbf{d}_i\|^2
$$

$$
+ \left\langle \sum_{s\in[S]}\lambda_t^s\mathbf{u}_t^s, \mathbf{x}_t - \mathbf{x}_* - \eta\sum_{t=0}^{T}\alpha^{(t-i)}\mathbf{d}_i \right\rangle - \frac{\mu}{2}\|\mathbf{x}_t - \mathbf{x}_*\|^2
$$

$$
+ \frac{1}{2}L\|\eta\sum_{t=0}^{T}\alpha^{(t-i)}\mathbf{d}_i\|^2
$$

$$
= \frac{1}{2\delta}\sum_{s\in[S]}\lambda_t^s\|\nabla f_s(\mathbf{x}_t) - \mathbf{u}_t^s\|^2 + \delta\|\mathbf{x}_t - \mathbf{x}_*\|^2 + \delta\|\eta\sum_{t=0}^{T}\alpha^{(t-i)}\mathbf{d}_i\|^2
$$

$$
+ \left\langle \mathbf{d}_t, \mathbf{x}_t - \mathbf{x}_* - \eta\sum_{t=0}^{T}\alpha^{(t-i)}\mathbf{d}_i \right\rangle - \frac{\mu}{2}\|\mathbf{x}_t - \mathbf{x}_*\|^2 + \frac{1}{2}L\|\eta\sum_{t=0}^{T}\alpha^{(t-i)}\mathbf{d}_i\|^2
$$

$$
\overset{(b)}{\leq} \frac{1}{2\eta}\left(\|\mathbf{x}_t - \mathbf{x}_*\|^2 - \|\mathbf{x}_{t+1} - \mathbf{x}_*\|^2\right) - \frac{1}{2}\eta\|\sum_{t=0}^{T}\alpha^{(t-i)}\mathbf{d}_i\|^2 - \frac{\mu}{2}\|\mathbf{x}_t - \mathbf{x}_*\|^2
$$

$$
+ \frac{1}{2}L\|\eta\sum_{t=0}^{T}\alpha^{(t-i)}\mathbf{d}_i\|^2
$$

$$
+ \frac{4}{\mu}\sum_{s\in[S]}\lambda_t^s\|\nabla f_s(\mathbf{x}_t) - \mathbf{u}_t^s\|^2 + \frac{\mu}{8}\|\mathbf{x}_t - \mathbf{x}_*\|^2 + \frac{\mu}{8}\|\eta\sum_{t=0}^{T}\alpha^{(t-i)}\mathbf{d}_i\|^2
$$

$$
= \frac{1}{2\eta}\left((1 - \frac{3\mu\eta}{4})\|\mathbf{x}_t - \mathbf{x}_*\|^2 - \|\mathbf{x}_{t+1} - \mathbf{x}_*\|^2\right) - (\frac{1}{2}\eta - \frac{\mu}{8}\eta^2 - \frac{1}{2}L\eta^2)\|\sum_{t=0}^{T}\alpha^{(t-i)}\mathbf{d}_i\|^2
$$

$$
+ \frac{4}{\mu}\sum_{s\in[S]}\lambda_t^s\|\nabla f_s(\mathbf{x}_t) - \mathbf{u}_t^s\|^2
$$

$$
\overset{(c)}{\leq} \frac{1}{2\eta}\left((1 - \frac{3\mu\eta}{4})\|\mathbf{x}_t - \mathbf{x}_*\|^2 - \|\mathbf{x}_{t+1} - \mathbf{x}_*\|^2\right) - (\frac{1}{2}\eta - \frac{\mu}{8}\eta^2 - \frac{1}{2}L\eta^2)\|\sum_{t=0}^{T}\alpha^{(t-i)}\mathbf{d}_i\|^2
$$

$$
+ \frac{4}{\mu}(\frac{L^2}{|\mathcal{A}|}\sum_{i=(n_t-1)q}^{t}\|\mathbf{x}_{i+1} - \mathbf{x}_i\|^2 + \sum_{s\in[S]}\lambda_t^s\|\nabla f_s(\mathbf{x}_{(n_t-1)q}) - \mathbf{u}_{(n_t-1)q}^s\|^2)
$$

$$
\overset{(d)}{\leq} \frac{1}{2\eta}\left((1 - \frac{3\mu\eta}{4})\|\mathbf{x}_t - \mathbf{x}_*\|^2 - \|\mathbf{x}_{t+1} - \mathbf{x}_*\|^2\right) - (\frac{1}{2}\eta - \frac{\mu}{8}\eta^2 - \frac{1}{2}L\eta^2)\|\sum_{t=0}^{T}\alpha^{(t-i)}\mathbf{d}_i\|^2
$$

$$
+ \frac{4}{\mu}(\frac{L^2}{|\mathcal{A}|}\sum_{i=(n_t-1)q}^{t}\|\mathbf{x}_{i+1} - \mathbf{x}_i\|^2) + \frac{\mu S}{4}\frac{I_{(\mathcal{N}_s<n)}}{\mathcal{N}_s}\sigma^2. \tag{88}
$$

where $(a)$ follows from Eqs. (86), (b) follows from $\|\mathbf{x}_t - \mathbf{x}_*\|^2 - \|\mathbf{x}_{t+1} - \mathbf{x}_*\|^2 = -\eta^2\|\mathbf{d}_t\|^2 + 2\langle\eta\mathbf{d}_t, \mathbf{x}_t - \mathbf{x}_*\rangle$ and we choose $\delta = \frac{\mu}{8}$. $(c)$ is from Lemma. 1. $(d)$ is from Eqs. (52). $(d)$ follows from $0 < \lambda_t^s < 1, \forall s \in [S]$

Next, telescoping the above inequality over $t$ from $(n_t - 1)\, q$ to $t$ where $t \le n_t q - 1$ and noting that for $(n_t - 1)\, q \le j \le n_t q - 1, n_j = n_t$, we obtain

$$\sum_{i=(n_t-1)q} \sum_{s\in[S]} \lambda_t^s \left[ f_s(\mathbf{x}_{i+1}) - f_s(\mathbf{x}_*) \right]$$

$$\overset{(a)}{\le} \frac{1}{2\eta} \left( (1 - \frac{3\mu\eta}{4}) \sum_{i=(n_t-1)q}^{t} \|\mathbf{x}_i - \mathbf{x}_*\|^2 - \sum_{i=(n_t-1)q}^{t} \|\mathbf{x}_{i+1} - \mathbf{x}_*\|^2 \right)$$

$$- (\frac{1}{2}\eta - \frac{\mu}{8}\eta^2 - \frac{1}{2}L\eta^2) \sum_{i=(n_t-1)q}^{t} \| \sum_{i=0}^{t} \alpha^{(t-i)}\mathbf{d}_i\|^2$$

$$+ \frac{4}{\mu} (\frac{L^2}{|\mathcal{A}|} \sum_{j=(n_t-1)q}^{t} \sum_{i=(n_j-1)q}^{j} \|\mathbf{x}_{i+1} - \mathbf{x}_i\|^2) + \frac{\mu}{4c_\gamma} \sum_{i=(n_t-1)q}^{t} \|\alpha^{(t-i)}\mathbf{d}_i\|^2$$

$$+ \frac{\mu}{4c_\gamma} \sum_{t=(n_t-1)q}^{t} \|\alpha^{(t-i)}\mathbf{d}_i\|^2 + + \frac{\mu}{4} \sum_{i=(n_t-1)q}^{t} \frac{\epsilon}{c_\epsilon}$$

$$\overset{(b)}{\le} \frac{1}{2\eta} \left( (1 - \frac{3\mu\eta}{4}) \sum_{i=(n_t-1)q}^{t} \|\mathbf{x}_i - \mathbf{x}_*\|^2 - \sum_{i=(n_t-1)q}^{t} \|\mathbf{x}_{i+1} - \mathbf{x}_*\|^2 \right)$$

$$- (\frac{1}{2}\eta - \frac{\mu}{8}\eta^2 - \frac{1}{2}L\eta^2) \sum_{i=(n_t-1)q}^{t} \| \sum_{t=0}^{T} \alpha^{(t-i)}\mathbf{d}_i\|^2$$

$$+ \frac{4}{\mu} (\frac{L^2}{|\mathcal{A}|} \sum_{j=(n_t-1)q}^{t} \sum_{i=(n_t-1)q}^{t} \|\mathbf{x}_{i+1} - \mathbf{x}_i\|^2)$$

$$+ \frac{\mu}{4c_\gamma} \sum_{t=(n_t-1)q}^{t} \|\alpha^{(t-i)}\mathbf{d}_i\|^2 + + \frac{\mu}{4} \sum_{i=(n_t-1)q}^{t} \frac{\epsilon}{c_\epsilon}$$

$$\overset{(c)}{\le} \frac{1}{2\eta} \left( (1 - \frac{3\mu\eta}{4}) \sum_{i=(n_t-1)q}^{t} \|\mathbf{x}_i - \mathbf{x}_*\|^2 - \sum_{i=(n_t-1)q}^{t} \|\mathbf{x}_{i+1} - \mathbf{x}_*\|^2 \right) + \frac{\mu}{4} \sum_{i=(n_t-1)q}^{t} \frac{\epsilon}{c_\epsilon}$$

$$- (\frac{1}{2}\eta - \frac{\mu}{8}\eta^2 - \frac{1}{2}L\eta^2 - \frac{4}{\mu} \frac{L^2 q\eta^2}{|\mathcal{A}|}) \sum_{i=(n_t-1)q}^{t} \| \sum_{t=0}^{T} \alpha^{(t-i)}\mathbf{d}_i\|^2)$$

$$+ \frac{\mu}{4c_\gamma} \sum_{t=(n_t-1)q}^{t} \|\alpha^{(t-i)}\mathbf{d}_i\|^2 + \frac{\mu}{4} \sum_{i=(n_t-1)q}^{t} \frac{\epsilon}{c_\epsilon}, \tag{89}$$

where $(a)$ follows from Eqs. (87), $(b)$ extends $j$ to $t$. $(c)$ follows from $t \le n_t q - 1$.

We continue the proof by further driving

$$\sum_{t=0}^{T} \sum_{s\in[S]} \lambda_t^s \left[ f_s(\mathbf{x}_{i+1}) - f_s(\mathbf{x}_*) \right]$$

$$= \sum_{i=0}^{q} \sum_{s\in[S]} \lambda_t^s \left[ f_s(\mathbf{x}_{i+1}) - f_s(\mathbf{x}_*) \right] + \sum_{i=q}^{2q} \sum_{s\in[S]} \lambda_t^s \left[ f_s(\mathbf{x}_{i+1}) - f_s(\mathbf{x}_*) \right] +$$

$$\cdot + \sum_{i=(n_T-1)q}^{T} \sum_{s\in[S]} \lambda_t^s \left[ f_s(\mathbf{x}_{i+1}) - f_s(\mathbf{x}_*) \right]$$

$$\leq \frac{1}{2\eta}\left((1-\frac{3\mu\eta}{4})\sum_{i=0}^{T}\|\mathbf{x}_i-\mathbf{x}_*\|^2 - \sum_{t=0}^{T}\|\mathbf{x}_{i+1}-\mathbf{x}_*\|^2\right)$$

$$-(\frac{1}{2}\eta-\frac{\mu}{8}\eta^2-\frac{1}{2}L\eta^2-\frac{4}{\mu}\frac{L^2 q\eta^2}{|\mathcal{A}|})\sum_{t=0}^{T}\|\sum_{t=0}^{T}\alpha^{(t-i)}\mathbf{d}_i\|^2)$$

$$+\frac{\mu}{4c_\gamma}\sum_{t=0}^{T}\|\alpha^{(t-i)}\mathbf{d}_i\|^2 + \frac{\mu}{4}T\frac{\epsilon}{c_\epsilon}. \tag{90}$$

Next, we have

$$\sum_{t=0}^{T}\sum_{s\in[S]}\lambda_t^s\left[f_s(\mathbf{x}_i)-f_s(\mathbf{x}_*)\right]$$

$$=\sum_{t=0}^{T}\sum_{s\in[S]}\lambda_t^s\left[f_s(\mathbf{x}_{i+1})-f_s(\mathbf{x}_*)-f_s(\mathbf{x}_{i+1})+f_s(\mathbf{x}_i)\right]$$

$$\leq\sum_{t=0}^{T}\sum_{s\in[S]}\lambda_t^s\left[f_s(\mathbf{x}_{i+1})-f_s(\mathbf{x}_*)\right]+\sum_{t=0}^{T}\sum_{s\in[S]}\lambda_t^s|f_s(\mathbf{x}_{i+1})-f_s(\mathbf{x}_i)|$$

$$\overset{(a)}{\leq}\frac{1}{2\eta}\left((1-\frac{3\mu\eta}{4})\sum_{i=0}^{T}\|\mathbf{x}_i-\mathbf{x}_*\|^2 - \sum_{t=0}^{T}\|\mathbf{x}_{i+1}-\mathbf{x}_*\|^2\right)$$

$$-(\frac{1}{2}\eta-\frac{\mu}{8}\eta^2-\frac{1}{2}L\eta^2-\frac{4}{\mu}\frac{L^2 q\eta^2}{|\mathcal{A}|}-[\frac{\eta}{4}-\frac{\eta^3 q}{2}\frac{L^2}{|\mathcal{A}|}]-\frac{\mu}{4c_\gamma})\sum_{t=0}^{T}\|\alpha^{(t-i)}\mathbf{d}_i\|^2$$

$$+\frac{\mu}{4}T\frac{\epsilon}{c_\epsilon}, \tag{91}$$

where $(a)$ follows from Eqs. (90). Let $|\mathcal{A}|=q=\lceil\sqrt{n}\rceil$ and $\eta\leq\min\{\frac{1}{2\mu},\frac{1}{8L},\frac{\mu}{64L^2}\}, c_\gamma\geq\frac{8\mu}{\eta}, c_\eta\geq \mu 2$, we have $(\frac{1}{2}\eta-\frac{\mu}{8}\eta^2-\frac{1}{2}L\eta^2-\frac{4}{\mu}\frac{L^2 q\eta^2}{|\mathcal{A}|}-[\frac{\eta}{4}-\frac{\eta^3 q}{2}\frac{L^2}{|\mathcal{A}|}]-\frac{\mu}{4c_\gamma})>\frac{\eta}{32}>0$

Thus, we have

$$\sum_{t=0}^{T}\sum_{s\in[S]}\lambda_t^s\left[f_s(\mathbf{x}_i)-f_s(\mathbf{x}_*)\right]$$

$$\leq\frac{1}{2\eta}\left((1-\frac{3\mu\eta}{4})\sum_{i=0}^{T}\|\mathbf{x}_i-\mathbf{x}_*\|^2-\sum_{t=0}^{T}\|\mathbf{x}_{i+1}-\mathbf{x}_*\|^2\right)+\frac{\epsilon}{2}. \tag{92}$$

Then, we have

$$\mathbb{E}[\sum_{s\in[S]}\lambda_t^s\left[f_s(\mathbf{x}_t)-f_s(\mathbf{x}_*)\right]]$$

$$\leq\frac{1}{2\eta}\left((1-\frac{3\mu\eta}{4})\mathbb{E}\|\mathbf{x}_t-\mathbf{x}_*\|^2-\mathbb{E}\|\mathbf{x}_{t+1}-\mathbf{x}_*\|^2\right)+\frac{\epsilon}{2}. \tag{93}$$

Averaging using weight $w_t=(1-\frac{3\mu\eta}{4})^{1-t}$ and using such weight to pick output $\mathbf{x}$. By using Lemma 1 in Karimireddy et al. (2020) with $\eta\geq\frac{1}{uR}$, we have

$$\mathbb{E}[\sum_{s\in[S]}\lambda_t^s\left[f_s(\mathbf{x}_t)-f_s(\mathbf{x}_*)\right]]\leq\|\mathbf{x}_0-\mathbf{x}_*\|^2\mu\exp(-\frac{3\eta\mu T}{4}) \tag{94}$$

$$=\mathcal{O}(\mu\exp(-\mu T)). \tag{95}$$

Then we have the convergence rate $\mathbb{E}[\sum_{s\in[S]} \lambda_t^s [f_s(\mathbf{x}_t) - f_s(\mathbf{x}_*)]] = \mathcal{O}(\mu \exp(-\mu T))$.

The total sample complexity can be calculated as: $\lceil \frac{T}{q} \rceil n + T \cdot |\mathcal{A}| \leq \frac{T+q}{q} n + T\sqrt{n} = T\sqrt{n} + n + T\sqrt{n} = O(n + \sqrt{n}\ln(\mu/\epsilon))$. Thus, the overall sample complexity is $\mathcal{O}(n + \sqrt{n}\ln(\mu/\epsilon))$. This completes the proof.

$\square$

## E    ADDITIONAL EXPERIMENT ON STRONGLY-CONVEX OPTIMIZATION PROBLEM

In this section, we conducted experiments to assess the performance of our algorithms on a strongly-convex optimization problem, where $\mathbf{F}(\mathbf{x}) = [f_1(\mathbf{x}) = \mathbf{x}^2, f_2(\mathbf{x}) = e^{-\mathbf{x}}]$. For this experiment, we selected hyperparameters $\eta = 0.005$ and $\alpha = 0.3$, while introducing stochasticity into the gradient by adding Gaussian noise with a range of (-1, 1). As shown in Fig. 4, it is evident that all of the algorithms successfully achieved convergence. Notably, the momentum-based algorithms, namely MOCO, STIMULUS-M, and STIMULUS-M$^+$, exhibited faster convergence compared to MGD, MSGD, STIMULUS, and STIMULUS$^+$ . We would also like to note that there isn't a significant difference between the stochastic algorithms (SMGD, MGD) and other algorithms. This is not necessarily because the stochastic algorithms are inferior, but perhaps because the strongly-convex function in question is too simplistic.

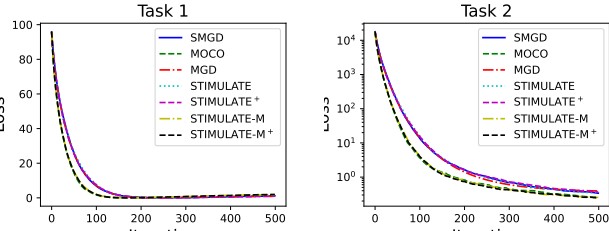

Figure 4: Convergence comparison on strongly-convex optimization problem.

Table 3: Results of normalized loss with the river flow dataset and learning tasks.

| | # of samples | Tasks | | | | | | | |
| --- | --- | --- | --- | --- | --- | --- | --- | --- | --- |
| | | 0 | 1 | 2 | 3 | 4 | 5 | 6 | 7 |
| SMGD | 8000 | 0.985 | 0.558 | 0.521 | 0.384 | 1 | 0.862 | 0.667 | 0.550 |
| MOCO | 8000 | 0.985 | 0.753 | 1 | 0.399 | 0.632 | 1 | 0.595 | 0.926 |
| MGDA | 128000 | 0.989 | 0.396 | 0.532 | 0.174 | 0.589 | 0.945 | 0.417 | 0.669 |
| STIMULUS | 27200 | .985 | 0.546 | 0.675 | 1 | 0.077 | 0.898 | 0.417 | 0.281 |
| STIMULUS$^+$ | 20947 | 0.996 | 1 | 0.528 | 0.178 | 0.990 | 0.395 | 0.427 | 1 |
| STIMULUS-M | 27200 | 0.996 | 0.864 | 0.530 | 0.475 | 0.036 | 0.271 | 1 | 0.264 |
| STIMULUS-M$^+$ | 21085 | 1 | 0.596 | 0.627 | 0.1781 | 0.0376 | 0.482 | 0.430 | 0.055 |