# OpenReview forum: "STIMULUS: Achieving Fast Convergence and Low Sample Complexity in Stochastic Multi-Objective Learning"
_ICLR.cc/2024/Conference — Submitted to ICLR 2024_

### Official Review · Reviewer_A8De · 2023-10-31

**Soundness:** 2 fair
**Presentation:** 3 good
**Contribution:** 2 fair
**Rating:** 3
**Confidence:** 4

**Summary:**

The focus of the paper is designing multi-objective optimization (MOO) algorithms with faster convergence rates compared to existing SOTA methods (and matching deterministic MOO counterpart) , for non-convex and strongly convex settings. The paper leverage variance reduction techniques to achieve the aforementioned faster convergence rates, which were not reported previously in MOO literature. The authors also provide empirical results, comparing the proposed method with prior MOO baselines, and show improved empirical performance as well.

**Strengths:**

* The proposed idea of incorporating variance reduction methods to improve convergence rate in MOO setting seems promising.
* The authors provide some theory (which is unclear, as described in next section) and experiments to validate the proposed method.

**Weaknesses:**

* The definition of Pareto optimality and Pareto stationarity does not seem to align with the metrics used in the convergence results. For example, while the authors claim the convergence to a Pareto stationary point by STIMULUS due to the result obtained in Theorem 1, it is unclear why the merit function used in this result can measure the Pareto stationarity of iterates.

* Due to the problem mentioned above, it is unclear whether the comparison for theoretical results provided in Table 1 is a fair one.

* In proof of Lemma 1, the authors use Lemma 1 of Feng et al. (2018), yet it is hard to see how result in Feng et al. (2018) can be used here, since the problem setting in Feng et al. (2018) is single objective optimization.

* The choice of stepsize in Theorems is unclear. For example, how does one go from equation (13) to (14) (in proof provided in appendix) by the choice of step size $\eta \leq 1/2$ ?

Minor comments:

* $|\mathcal{A}|$ in equation (2) is not defined before using.
* Using index $s$ in equation (3) seems not necessary.

**Questions:**

* Can the authors explain the relationship between the merit functions used in Theorems 1-6, and the definitions of Pareto stationarity/optimality?

* Can the authors elaborate on why the inequality (9) (in proof of Lemma 1) hold, and how it relate to Lemma 1 in Fang et al. (2018) ?

---

> ### Author Response · Authors · 2023-11-22
> **Response to Reviewer A8De's Comments [Part 1]**
>
> > **Your Comment 1:**  The definition of Pareto optimality and Pareto stationarity does not seem to align with the metrics used in the convergence results. For example, while the authors claim the convergence to a Pareto stationary point by STIMULUS due to the result obtained in Theorem 1, it is unclear why the merit function used in this result can measure the Pareto stationarity of iterates.
>
> **Our response:** Thanks for your comments. The reviewer is correct that the definitions of Pareto optimality and Pareto stationarity in Defintion 1 and Defintion 2, respectively, are different from the convergence metric we use in Theorem 1, which follows from Definition 3. Here, we want to clarify their connections as follows:
>
>
>
>  The definitions of Pareto optimality and Pareto stationarity in Definition 1 and Definition 2, respectively, are their original and most general definitions. However, these definitions are based on the notion of "existence," which is not convenient and amenable for stopping criteria in algorithm design. Thus, we need equivalent definitions of Pareto optimality and Pareto stationarity, particularly when the problems are endowed with additional structural properties (differentiable objectives in this paper), which are more amenable for checking Pareto optimality and Pareto stationarity. Such equivalent equivalent conditions are stated in Definition 3, which is used as the convergence metric in Theorem 1.
>
>
>
>
>
>
>
> As an analogy, Definition 3 is similar to the conventional definition of optimality in general optimization, where the most basic optimality concept is not useful for algorithm design and analysis in practice. Thus, when the problem has special structural properties, e.g., convexity and constraint qualifications, the famous KKT condition, which is both necessary and sufficient for optimality in this case, will be used to check whether we have achieved an optimal solution.
>
>
>
>  -------------------------
>
> > **Your Comment 2:** Due to the problem mentioned above, it is unclear whether the comparison for theoretical results provided in Table 1 is a fair one.
>
>
> **Our Response:** Thank you for your comment regarding the fairness of our theoretical comparisons presented in Table 1. We understand your concern and would like to clarify the basis on which these comparisons are made, emphasizing the consistency of our metrics with those used in other referenced works.
>
> **1. Consistency with MGD and SMGD Metrics:** In our comparison, we use the same the metrics as in MGD (Fliege et al., 2019) Theorem 3.1 and Theorem 4.1, and SMGD (Yang et al., 2022) Theorem 1, thus ensuring a fair comparison in terms of theoretical analysis.
>
> **2. Alignment with MoCo and CR-MOGM Metrics:** Similarly, in MoCo (Fernando et al., 2023) Theorems 1-3 and CR-MOGM (Zhou et al., 2022b) Theorems 2 and 3, the metric $||\sum _ {s \in [S]} \lambda _ t^* \nabla F\left(\mathbf{x} _ t\right) ||^2$ is employed. In contrast, our metric is slightly different and we use
> $||\sum _ {s \in [S]}\lambda _ {t}^{s} \nabla F\left(\mathbf{x} _ t\right)||^2$, which is not only compatible with (Fernando et al., 2023) but also avoids the limitation of making some subtle technical assumptions as in (Fernando et al., 2023). Due to this reason, the proposed metric in Definition 3 is also a novelty of this paper. Specifically, our metric satisfies $||\sum _ {s \in [S]} \lambda _ {t}^* \nabla F\left(\mathbf{x} _ t\right)||^2 \leq ||\mathbf{d} _ t||^2 = ||\sum _ {s \in [S]} \lambda _ {t}^{s} \nabla F\left(\mathbf{x} _ t\right)||^2$, meaning that our metric is an even tighter convergence metric and implicitly showing the convergence of $\{ \lambda_t \} \rightarrow \lambda^*$. Thus, a convergence rate measured by our metric will also hold when measured by the metric in (Fernando et al., 2023), thus further substantiating the fairness of our comparison.

---

> ### Author Response · Authors · 2023-11-22
> **Response to Reviewer A8De's Comments [Part 2]**
>
> > **Your Comment 3:** In proof of Lemma 1, the authors use Lemma 1 of Feng et al. (2018), yet it is hard to see how result in Feng et al. (2018) can be used here, since the problem setting in Feng et al. (2018) is single objective optimization.
>
> **Our Response:** Thank you for your observation regarding the application of Lemma 1 from Feng et al. (2018) in our proof of Lemma 1. We appreciate the opportunity to clarify this aspect of our methodology. Although the problem setting in [Feng et al. (2018)] is focused on single-objective optimization, the underlying principles of their Lemma 1 can be appropriately applied to our multi-objective optimization (MOO) context. This is because the term $\mathbb{E}_t[\|\nabla f_s(\mathbf{x}_t) - \mathbf{u}_t^s\|]$ in Lemma 1 only involes **one** objective function $f_s(\mathbf{x}$) can be treated independently as a single-objective problem.
>
>
> -------------------------
>
> > **Your Comment 4:** The choice of stepsize in Theorems is unclear. For example, how does one go from equation (13) to (14) (in proof provided in appendix) by the choice of step size?
>
> **Our Response:** In Eq.(13), we have $f_s(\mathbf{x} _ {t+1}) \leq f _ s(\mathbf{x} _ t) + \frac{\eta}{2}  || \nabla f _ s(\mathbf{x} _ t) - \mathbf{u} _ t^s ||^2 - \eta \left( \frac{1}{2}- \frac{1}{2}L \eta \right) || \mathbf{d} _ t ||^2$.
> As discussed in our paper, by setting $\left( \frac{1}{2} - \frac{1}{2} \eta \right) \geq \frac{1}{4}$, that is, $\eta \leq  \frac{1}{2}$. This ensures that the term associated with $|| \mathbf{d} _ t ||^2$in the inequality is *negative*, which implies a descent in the objective value of $f_s(\cdot)$. Thus, we have $f _ s(\mathbf{x} _ {t+1}) \leq  f _ s(\mathbf{x} _ t) + \frac{\eta}{2}  || \nabla f _ s(\mathbf{x} _ t) - \mathbf{u} _ t^s ||^2 -\frac{\eta}{4} || \mathbf{d} _ t ||^2$ as shown in our Eq.(14). We we will include more detailed explanations and justifications for the choice of step size in the final version of our paper.
>
> -------------------------
>
> > **Your Comment 5:** Minor comments: 1. $\mathcal{A}$ in equation (2) is not defined before using.2.Using index $s$ in equation (3) seems not necessary.
>
> **Our Response:**  Thanks for catching these issues. We have fixed them accordingly in the revision of this paper.
>
>
>
> -------------------------
>
> > **Your Comment 6:** Can the authors explain the relationship between the merit functions used in Theorems 1-6, and the definitions of Pareto stationarity/optimality?
>
>
> **Our Response:** Thanks for your questions. By "merit function," we assume you meant the convergence metrics used in Theorems 1-6. In the following, we will explain how these merit functions relate to the concepts of Pareto stationarity and Pareto optimality in multi-objective optimization (MOO). Recall that we have used the following merit functions in different problem settings:
>
> 1. **The Merit Functions for Nonconvex Objectives:** In this paper, the merit function $\|\sum _ {s \in [S]}\lambda _ {t}^{s} \nabla F\left(\mathbf{x} _ t\right)\|^2$ is used for non-convex objective functions. This merit function meansurs the "Pareto stationarity" defined in Definition 2.
> <!--This function measures the squared norm of the descent direction over a series of iterations. In non-convex MOO problems, minimizing this metric indicates that the algorithm is making progress towards a stationary point.-->

---

> ### Author Response · Authors · 2023-11-22
> **Response to Reviewer A8De's Comments [Part 3]**
>
> 2. **The Merit Functions for Strongly-Convex Objectives:** In this paper, the merit function $\sum _ {s \in [S]} \lambda _ t^{s} [ f _ s(\mathbf{x} _ t) - f _ s(\mathbf{x} _ *) ]$ is used for strongly convex objective functions. This merit functions measures the "pareto optimality" defined in Definition 1.
> <!--This metric reflects the differences between the current solution and an optimal solution across all objectives. It's particularly relevant in strongly-convex settings where the objective functions exhibit certain curvature properties.-->
>
> Also, recall from Defintions 1 and 2, Pareto optimality and Pareto stationarity are defined as follows:
>
> 3. **Pareto Optimality:** A solution $\mathbf{x}$ is Pareto-optimal if there is no other solution that is better in all objectives. Weak Pareto optimality relaxes this by not requiring improvement in all objectives simultaneously.
>
> 4. **Pareto Stationarity:** A solution $\mathbf{x}$ is Pareto-stationary if no common descent direction $\mathbf{d}$ exists such that $\nabla f_s(\mathbf{x})^{\top} \mathbf{d} <0$, $\forall s$.
>
> It has been shown in [R1] that Pareto stationarity implies weak Pareto optimality if all objectives are convex. Further, Pareto stationarity implies Pareto optimality if all objectives are strongly convex. The relationships between the merit functions and the basic definitions of Pareto optimaltiy and Pareto stationarity are as follows:
>
> - **Relationship between $\|\sum _ {s \in [S]}\lambda _ {t}^{s} \nabla F\left(\mathbf{x} _ t\right)\|^2$ and Pareto stationarity:** From the quantity $\|\sum _ {s \in [S]}\lambda _ {t}^{s} \nabla F\left(\mathbf{x} _ t\right)\|^2$, it is clear that if $\|\sum _ {s \in [S]}\lambda _ {t}^{s} \nabla F\left(\mathbf{x} _ t\right)\|^2 \leq \epsilon$ at $\mathbf{x}$ for some small $\epsilon >0$, each objective function's gradient norm $\|\nabla f_s(\mathbf{x})\|$ is also small, which implies that $\mathbf{x}$ is a (near) Pareto stationary point. Thus, the merit function $\|\sum _ {s \in [S]}\lambda _ {t}^{s} \nabla F\left(\mathbf{x} _ t\right)\|^2$ can be used as a convergence metric.
>
> - **Relationship between $\sum _ {s \in [S]} \lambda _ t^{s} [ f _ s(\mathbf{x} _ t) - f _ s(\mathbf{x} _ *) ]$ and Pareto optimality under strong convexity:** To see their connection, note that from strong convexity, we have $f _ s(\mathbf{x} _ t) \geq f _ s(\mathbf{x} _ *) + \nabla f _ s^{\top}(\mathbf{x} _ *)(\mathbf{x} _ t-\mathbf{x} _ *) +\frac{\mu}{2}||\mathbf{x} _ t-\mathbf{x} _ *||^2$ for some $\mu>0$. Thus, for any $\lambda _ t^s>0$, $\forall s \in [S]$ with $\sum _ {s=1}^{S} \lambda _ t^s =1$, we have
>
>     $\sum _ {s\in[S]} \lambda _ t^s[f _ s(\mathbf{x} _ t) - f _ s(\mathbf{x} _ *)]$
>     $\geq \sum _ {s \in [S]} \lambda _ t^s \nabla f _ s^{\top}(\mathbf{x} _ *)(\mathbf{x} _ t-\mathbf{x} _ *) + \frac{\mu}{2}||\mathbf{x} _ t-\mathbf{x} _ *||^2$
>     $= \sum _ {s \in [S]} \lambda _ t^s \nabla f _ s^{\top} \mathbf{d} + \frac{\mu}{2}||\mathbf{x} _ t-\mathbf{x} _ *||^2$,
>
>     where we define $\mathbf{d} \triangleq \mathbf{x} _ t-\mathbf{x} _ *$ in the last equality for convenience. Since $\mathbf{x} _ *$ is Pareto-stationary and all objective functions are strongly convex, it follows that $\mathbf{x} _ *$ is also Pareto-optimal. Thus, there must exist at least one $\tilde{s} \in [S]$ such that $\nabla f _ {\tilde{s}}^{\top}(\mathbf{x} _ *) \mathbf{d} > 0$. Now, we show there always *exist* $\lambda _ t^s$, $\forall s\in [S]$ to make the merit function **non-negative**: For $\tilde{s}$, we choose a $\lambda _ t^{\tilde{s}}$-value that is close to 1. For all other $s \ne \tilde{s}$, we choose a small $\lambda _ t^s$-value that is close to 0. Then, by pushing the $\lambda _ t^{\tilde{s}}$-value toward 1 and other $\lambda _ t^s$-values towards 0 (whiling maintaining $\sum _ {s=1}^{S} \lambda _ t^s =1$), we can always make $\sum _ {s\in[S]} \lambda _ t^s[f _ s(\mathbf{x} _ t) - f _ s(\mathbf{x} _ *)]$ non-negative.
>
> We hope the explanations above help the reviewer see the relationship between the merit functions and the Pareto optimality/stationarity concepts.
>
>
> [R1] Hiroaki Mukai. Algorithms for multicriterion optimization, IEEE Transactions on Automatic Control, 25(2): 177-186, 1980.

---

> > ### Author Response · Authors · 2023-11-23
> > **Response to Reviewer A8De's Comments [Part 4]**
> >
> > > **Your Comment 7:** Can the authors elaborate on why the inequality (9) (in proof of Lemma 1) hold, and how it relate to Lemma 1 in Fang et al. (2018) ?
> >
> > **Our Response:** Thank you for your question regarding the derivation of inequality (9) in the proof of Lemma 1 and its relation to Lemma 1 in Fang et al. (2018). We would like to provide further details:
> >
> >
> > Similar to the result shown on Page 12 in Appendix,  Eqs. (A.1) and (A.2) in [Fang et al. (2018)], we have:
> >
> > $\mathbb{E} _ t \| \nabla f _ s(\mathbf{x} _ t) - \mathbf{u} _ t^s \|^2$
> >
> > $= \mathbb{E} _ t \| \nabla f _ s(\mathbf{x} _ {t-1}) - \mathbf{u} _ {t-1}^s \|^2 + \mathbb{E} _ t \| \frac{1}{|\mathcal{A}|} \sum _ {j\in \mathcal{A}}\left( \nabla f _ {sj} (\mathbf{x} _ {t};\xi _ {sj}  ) - \nabla f _ {sj} (\mathbf{x} _ {t-1};\xi _ {sj} )  +\nabla f _ s(\mathbf{x} _ {t-1})-\nabla f _ s(\mathbf{x} _ {t}) \right) \|$
> >
> > $\leq\mathbb{E} _ t \| \nabla f _ s(\mathbf{x} _ {\left(n _ t-1\right) q}) - \mathbf{u} _ {\left(n _ t-1\right) q}^s \|^2 +  L^2 \sum _ {i=\left(n _ t-1\right) q}^t\frac{1}{|\mathcal{A}|} \mathbb{E}\|\mathbf{x} _ {i+1}-\mathbf{x} _ {i}\|^2$.
> >
> > The first equality stems from Proposition 1 in [Fang et al. (2018)], where the expectation of the gradient difference is expanded by adding and substracting identical terms. The second inequality leverages Eq. (2.3) from [Fang et al. (2018)] and bounding based on the Lipschitz continuity of the gradients.
> >
> > We can draw a similar conclusion to that in Lemma 1 of [Fang et al. (2018)] because the term $\mathbb{E}_t[\|\nabla f_s(\mathbf{x}_t) - \mathbf{u}_t^s\|]$ in Lemma 1 only involes **one** objective function $f_s(\mathbf{x}$) can be treated independently as a single-objective problem.

---

### Official Review · Reviewer_zAtp · 2023-11-04

**Soundness:** 2 fair
**Presentation:** 3 good
**Contribution:** 2 fair
**Rating:** 5
**Confidence:** 3

**Summary:**

This paper proposes to use variance reduction techniques to improve the sample complexity of stochastic multi-objective learning in finite-sum problems. It achieves the state-of-the-art sample complexity, matching the one with full-batch gradient descent.
Experiments on some benchmark datasets demonstrate the effectiveness of the proposed method.

**Strengths:**

1. This paper studies MOO in the finite-sum problem, which has not been extensively considered in MOO literature before as far as I know.

2. This paper proposes a variance-reduced algorithm that improves the state-of-the-art sample complexity of existing algorithms for multi-objective finite-sum problems.

**Weaknesses:**

1. The comparison with existing algorithms in Table 1 may not be fair because they are not focused on the same settings. The setting analyzed in this paper is the finite-sum setting which is more restrictive.

2. The benefit of the proposed method over linear scalarization in MOO is unclear. This is because linear scalarization can also achieve convergence to Pareto stationary points. By applying variance reduction techniques such as SVRG to linear scalarization, it can achieve a similar convergence rate to Pareto stationary points as this paper.
Therefore, only providing convergence to Pareto stationary points is not enough to show the benefit of the proposed method over the simplest linear scalarization.
More discussion should be provided.



3. Quantitative results are too limited to understand the practical performance of the proposed method.
Also, in addition to performance of each task, a widely used measure is $\Delta m $\% (e.g. in MOCO paper) to show the overall performance on all tasks.


### Minor

1. Some notations or definitions are not clear. See **Questions-2**.

2. In Section 2 - 2) overview of MOO algorithm, it is inaccurate to say that "recent work such as (Fernando et al., 2022) uses bi-level formulation to mitigate bias". In fact, (Fernando et al., 2022) uses momentum-based methods to mitigate bias, and apply to bi-level optimization problems.


3. Typos

- Below Definition 2: "non-convex MOO probolems" -> "non-convex MOO problems"

- Below Theorem 1: "sample compleixty" -> "sample complexity"

**Questions:**

1. What is the benefit of the proposed algorithm compared to applying variance-reduced algorithms such as SVRG to linear scalarization in MOO? In other words, applying such algorithms can also achieve similar sample complexity or convergence rate to Pareto stationary points. Therefore, the benefit of using the proposed stochastic variant of MGD is unclear.

2. Some notations are not defined clearly. See below.

- In Definition 3, Theorem 2 and 4, what is $i$ in $\lambda_i^s$? Shouldn't it be $\lambda_t^s$?

- What is $\xi$ in Eq.(6)? In Eq.(6), are you missing a sum of all samples $\xi \in \mathcal{N}_ s$?

- In Definition 4, what is "incremental first-order oracle (IFO)"? I know it is a widely used concept in finite-sum problems, but it is better to provide a formal definition or at least some references for completeness.
In addition, it could benefit to introduce finite-sum problems and IFO earlier to provide some context for readers.

- In Algorithm 1, line 5, it says "compute $\mathbf{u}_ t^s$ as in Eq.(4)", but Eq.(4) computes $\lambda_ t^s$, is this a typo?



3. Why only non-convex and strongly-convex cases are analyzed? What is the rate for convex cases? Are there any additional challenges to analyzing convex cases? It would be better to provide some discussion on this aspect.


4. Below Table 1, it mentions $\mathbf{x}^*$ is the Pareto-optimal point. However, there can be multiple Pareto-optimal points with different function values. This will result in the term $||\mathbf{x}_ 0 - \mathbf{x}_ *||$ not well defined in Theorem 2. Could you elaborate more on this?

5. The measure $\sum_{s\in [S]} \lambda_t^s [f_s(x_t) - f_s(x_*)]$ has some issues because it can be negative. See more discussions in (Liu & Vincente 2021). You need to make additional assumptions to make this a valid convergence metric.

---

> ### Author Response · Authors · 2023-11-22
> **Response to Reviewer zAtp's Comments [Part 1]**
>
> > **Your Comment 1:** The comparison with existing algorithms in Table 1 may not be fair because they are not focused on the same settings. The setting analyzed in this paper is the finite-sum setting which is more restrictive.
>
>
> **Our Response:** Thank you for your comments. However, it is inaccurate to state that "the setting analyzied in this paper is the finite-sum setting.". Thus, the comparison in Table 1 remains fair and justifiable for the following reasons:
>
> **1. The comparisons between STIMULUS/STIMULUS-M and MGD are fair:** Both STIMULUS/STIMULUS-M involves computation of full gradients, which is typically feasible in the fnite-sum setting. To see this, note that the sample complexity results in MGD and STIMULUS/STIMULUS-M in Table 1 all depend on $n$, which is the size of the dataset.
>
> **2. The comparisons between STIMULUS$^+$/STIMULUS-M$^+$ and SMGD/MOCO/CR-MOGM are fair:** All these methods, including our STIMULUS$^+$/STIMULUS-M$^+$ do **not** require full gradient evaluation. Thus, our proposed STIMULUS$^+$/STIMULUS-M$^+$ methods are also applicable to the same expectation minimization MOO problems (batch size is chosen as $|\mathcal{N} _ s| = \min \\{ c _ \gamma \sigma^2\gamma _ {t}^{-1}, c _ \epsilon \sigma^2 \epsilon^{-1} \\}$. Specifically, in STIMULUS$^+$/STIMULUS-M$^+$, we propose a **adaptive batch** approach, which allows STIMULUS$^+$/STIMULUS-M$^+$ to work with expectation minimization MOO problems as those studied by SMGD/MOCO/CR-MOGM.
>
>
>
> ------------------
>
>
> > **Your Comment 2:** The benefit of the proposed method over linear scalarization in MOO is unclear. This is because linear scalarization can also achieve convergence to Pareto stationary points. Therefore, only providing convergence to Pareto stationary points is not enough to show the benefit of the proposed method over the simplest linear scalarization. More discussion should be provided.
>
>
> **Our Response:**
> It is worth pointing out that linear scalarization methods are limited to identifying the convex hull of the Pareto front ([R5,R6]), whereas (stochastic) multi-gradient methods, including our proposed algorithms, have the capability to uncover the Pareto front. Essentially, this represents a distinct advantage for all multi-gradient algorithms over linear scalarization methods. This paper contributes by demonstrating that variance reduction can significantly enhance the complexity of stochastic multi-gradient methods by improved convergence.
>
> Thank you for your insightful comment regarding the comparison of our proposed method with linear scalarization in MOO. We agree with the reviewer that linear scalarization is a commonly used and relatively straightforward approach in MOO. However, working with vector-valued objectives in MOO offers unique benefits that do not exist in linear scalarization:
>
>
> Specifically, our method is built upon the multi-gradient descent approach (MGDA), which dynamically calculates the weights for each task based on the gradient information in each iteration. Compared to the linear scalarization method that uses fixed or pre-defined weights for each objective, the dynamic weighting approach adapts much better to the landscapes of different MOO problems, which enables a much more flexible exploration on the Pareto front.
>
> [R5] S. Boyd and L. Vandenberghe. Convex optimization. Cambridge University Press, 2004.
>
> [R6] M. Ehrgott. Multicriteria optimization, volume 491. Springer Science & Business Media, 2005).

---

> > ### Author Response · Authors · 2023-11-22
> > **Response to Reviewer zAtp's Comments [Part 2]**
> >
> > > **Your Comment 3:** Quantitative results are too limited to understand the practical performance of the proposed method. Also, in addition to performance of each task, a widely used measure is $\Delta m\%$ (e.g. in MOCO paper) to show the overall performance on all tasks.
> >
> >
> > **Our Response:** Thank you for your comments regarding the quantitative results and the suggestions on using of measures like $\Delta m\%$ to assess the overall performance on all tasks. We would like to further clarify why we conduct experiments in the way we did. Note that the main theoretical results and findings in this paper is the demonstration of enhanced sample efficiency in our proposed VR-assisted MGDA-based algorithms for MOO problems. We emphasize the convergence rate and sample complexity as these are key indicators of the theoretical advancements we have made. Thus, our experimental setup and results are designed to validate these theoretical aspects. By showcasing that our algorithms not only converge but do so with greater convergence speed and sample efficiency (cf. Fig. 2, Fig. 3, Fig. 4, and Table 3), we provide substantial evidence of their practical utility and effectiveness.
> >
> > Regarding the suggestion on comparing with other algorithms using the $\Delta m \%$ metric, we agree that this metric is useful to show the overall performance on all tasks, particularly useful in scenarios where individual task performance and trade-offs among tasks are crucial. While $\Delta m\%$ is a valuable measure, it is challenging for us to conduct comprehensive experiments using this metric in this rebuttal period due to time limitation. As mentioned earlier, our primary aim is to highlight the low sample complexity and fast convergence properties of our algorithms. Thus, we hope that our existing experiments still provide sufficient evidence to showcase the effectiveness of our proposed algorithms. We will defnitely conduct experiments using the $\Delta m\%$ metric in revisions of this work, and we appreciate the reviewer for this great suggestion.
> >
> > --------------------
> >
> >
> > > **Your Comment 4:** In Section 2-2) overview of MOO algorithm, it is inaccurate to say that "recent work such as (Fernando et al., 2022) uses bi-level formulation to mitigate bias". In fact, (Fernando et al., 2022) uses momentum-based methods to mitigate bias, and apply to bi-level optimization problems.
> >
> > **Our Response:** Thank you for pointing out the inaccuracy in our description of the work [Fernando et al., 2022] in the overview of MOO algorithms. You are correct in stating that [Fernando et al., 2022] primarily employed momentum-based methods to mitigate bias in multi-objective optimization (MOO). Their approach indeed revolves around the use of these methods, which is a significant aspect of their contribution. The application of these methods to bi-level optimization problems is an additional facet of their work, rather than the core mechanism for bias mitigation as our original statement may have implied.
> >
> > We acknowledge this oversight and will revise the section accordingly to accurately reflect the contributions of [Fernando et al., 2022]. Our revised text will clarify that their work focuses on using momentum-based methods for bias mitigation in MOO, and these methods are further applied within the context of bi-level optimization problems.
> >
> > -------------------
> >
> > > **Your Comment 5:** Below Definition 2: "non-convex MOO probolems" -> "non-convex MOO problems". Below Theorem 1: "sample compleixty" -> "sample complexity"
> >
> > **Our Response:**  Thanks for catching these typos, which we have fixed accordingly. In addition to these typos, we have conducted a thorough review of the entire paper to ensure the quality of writing.
> >
> > -------------------------

---

> > > ### Author Response · Authors · 2023-11-22
> > > **Response to Reviewer zAtp's Comments [Part 3]**
> > >
> > > > **Your Comment 6:** By applying variance reduction techniques such as SVRG to linear scalarization, it can achieve a similar convergence rate to Pareto stationary points as this paper. What is the benefit of the proposed algorithm compared to applying variance-reduced algorithms such as SVRG to linear scalarization in MOO? In other words, applying such algorithms can also achieve similar sample complexity or convergence rate to Pareto stationary points. Therefore, the benefit of using the proposed stochastic variant of MGD is unclear.
> > >
> > >
> > > **Our Response:**
> > >
> > > Thank you for your question. We recognize the effectiveness of linear scalarization as a straightforward and commonly used approach in MOO. However, our method, which employs a multi-gradient descent approach(MGDA), offers unique benefits that do not exist in linear scalarization:
> > >
> > > 1.  Benefits of MGDA: Unlike linear scalarization, which typically uses fixed or pre-defined weights for each objective, MGDA dynamically calculates the weights for each task based on the gradient information at each iteration. This dynamic weighting offers a much more flexible exploration on the Pareto front.
> > >
> > > 2. Benefits Over VR-techniques on solving MGDA-based MOO:
> > > Variance reduction techniques can significantly improve the convergence rates and sample-complexities in MOO optimization problem. Faster convergence and smaller sample-complexities are particularly beneficial in large-scale problems or when dealing with complex objective functions.
> > >
> > > In conclusion, balancing the trade-offs between different objectives is a key challenge in MOO. VR techniques in MGDA-based methods help in achieving a more balanced and efficient exploration of the trade-offs between multiple objectives, leading to more satisfactory and well-rounded Pareto optimal solutions.
> > >
> > >
> > > ------------------
> > >
> > > > **Your Comment 7:**  Some notations are not defined clearly. See below.
> > > > - In Definition 3, Theorem 2 and 4, what is $i$ in $\lambda _ i^s$? Shouldn't it be $\lambda _ t^s$?
> > > > - What is $\xi$ in Eq.(6)? In Eq.(6), are you missing a sum of all samples?
> > > > - In Definition 4, what is "incremental first-order oracle (IFO)"? I know it is a widely used concept in finite-sum problems, but it is better to provide a formal definition or at least some references for completeness. In addition, it could benefit to introduce finite-sum problems and IFO earlier to provide some context for readers.
> > > > - In Algorithm 1, line 5, it says "compute $\mathbf{{u} _ t^s$ as in Eq.(4)", but Eq.(4) computes $\lambda _ t^s$, is this a typo?
> > >
> > >
> > >
> > > **Our response:** Thanks for your comments.
> > >
> > >
> > > 1. Thanks for catching. In Theorem 2 and 4, our convergence metric is $\mathbb{E}||\sum _ {s\in [S]}\lambda _ t^s\nabla f _ s(\mathbf{x} _ t)||^2$ for non-convex MOO problems.
> > >
> > > 2. $\xi _ {sj}$ denote the $j$-th sample for task $s$. Eqs.(6) is $\mathbf{u} _ t^s = \frac{1}{|\mathcal{N} _ s|} \sum _ {j\in \mathcal{N} _ s}\nabla f _ s(\mathbf{x} _ {t};\xi _ {sj} ), \quad \forall s \in [S].$ We are grateful for your constructive feedback and assure you that similar issues will be addressed in the final version of our paper.
> > >
> > >
> > > 3. **Basic Concept of Incremental First-Order Oracle (IFO):** An incremental first-order oracle (IFO) in the context of optimization, particularly finite-sum problems, is a computational model that, given a point $\mathbf{x}$ and a sample index $j$, returns the gradient of the function with respect to $f(\mathbf{x};\xi _ j)$ for that particular sample. In the case of MOO, the IFO evaluates the $\nabla f _ {sj}(\mathbf{x}; \xi _ {sj})$ for task $s$.
> > > **Role in Optimization:** The IFO is a crucial concept in finite-sum optimization problems, where the objective function is expressed as the sum of several component functions. Each call to the IFO provides gradient information for a specific component, aiding in the iterative optimization process.
> > > A comprehensive explanation of IFOs can be found in the work of [R4, R5]. This reference provides a formal definition and explores the application of IFOs in various optimization scenarios.
> > > We appreciate your constructive feedback and will incorporate a formal definition and reference for the incremental first-order oracle (IFO), as well as provide necessary background on finite-sum problems early in the paper.
> > > [R4] Gao, Hongchang, and Heng Huang. "Can stochastic zeroth-order Frank-Wolfe method converge faster for non-convex problems?." International conference on machine learning. PMLR, 2020.
> > > [R5] Reddi, Sashank J., et al. "Fast incremental method for smooth nonconvex optimization." 2016 IEEE 55th conference on decision and control (CDC). IEEE, 2016
> > >
> > > 4. Thanks for cathching this typo. In line 5, ALgorithm 1. We compute  the full gradient as follow: $\mathbf{u} _ t^s=\frac{1}{n}\sum _ {j=1}^n \nabla f _ {sj} (\mathbf{x} _ {t};\xi _ {sj}  ),\forall s \in [S]$. In light of this, we will conduct a careful review of our manuscript to ensure such typographical errors are corrected.

---

> > > > ### Author Response · Authors · 2023-11-22
> > > > **Response to Reviewer zAtp's Comments [Part 4]**
> > > >
> > > > > **Your Comment 10:** Why only non-convex and strongly-convex cases are analyzed? What is the rate for convex cases? Are there any additional challenges to analyzing convex cases? It would be better to provide some discussion on this aspect.
> > > >
> > > > **Our Response:** Thanks for your comments and questions. We would like to further clarify our choice of (non-)convexity settings as follows:
> > > >
> > > > 1. **Why non-convex and strongly-convex cases:** We note that the **non-convex** case is the *most relevant* setting in machine learning, particularly in problems with neural network models that are highly non-convex and unstructured. Non-convex optimization is often considered the most challenging scenario in optimization, which is NP-hard in general. On the other hand, the **strongly-convex** setting, though special and far more analytically tractable, represents a theoretically and practically significant case. Strong convexity provides additional geometic structure that guarantees faster convergence rates. This makes it an important case to study and understand in depth. Also, in practice, the strongly convex setting corresponds to many common ML applications (e.g., linear models with $\ell _ 2$ regularization).
> > > >
> > > > 2. **What about the convex case:** The general convex case does not possess the challenging and complex landscapes with multiple local minima, and it also lacks the strong curvature of the strongly convex setting that leads to faster convergence. For the general convex settings, the convergence rates typically lie between those of non-convex and strongly-convex cases. While the general convex setting is simpler compared to the non-convex setting, they still present unique challenges in MOO. For instance, the interaction between multiple convex objectives can still create complex Pareto fronts, making the analysis non-trivial. The convex case indeed merits further investigation for future research, which deserves an independent paper dedicated to this topic. We thank the reviewer for suggesting this direction.
> > > >
> > > > -------------------------
> > > >
> > > > > **Your Comment 11:** Below Table 1, it mentions $\mathbf{x} _ *$ is the Pareto-optimal point. However, there can be multiple Pareto-optimal points with different function values. This will result in the term $||\mathbf{x} _ t - \mathbf{x} _ *||$ not well defined in Theorem 2. Could you elaborate more on this?
> > > >
> > > > **Our Response:** In MOO, there are typically **multiple** Pareto-optimal points, each representing a different trade-off among the objectives. A Pareto-optimal point is one where no objective can be improved without worsening at least one other objective.
> > > > Different Function Values: Each Pareto-optimal point can have distinct function values for the different objectives. This diversity is a key feature of the Pareto front in MOO.
> > > >
> > > > In our paper, $\mathbf{x} _ *$ represnts a **Pareto-stationary point, not a common minimizer for all $f _ s(\cdot)$**. In the strongly convex setting, a Pareto-stationary solution further implies Pareto-optimal solution [R4]. More specifically, a solution $\mathbf{x} _ *$ is considered Pareto-optimal if there is no other feasible solution that would **improve one objective without causing at least one other objective to worsen**. Essentially, a Pareto-optimal solution represents a point of equilibrium where no objective can be improved without compromising others.
> > > >
> > > >
> > > > Regarding Theorem 2, the term $||\mathbf{x} _ t - \mathbf{x}^*||^2$  measured the difference between the current point $\mathbf{x} _ t$ in the algorithm's iteration and a Pareto-stationary (and hence Pareto-optimal) point $\mathbf{x}^*$.
> > > >
> > > > While it's true that there could be multiple such Pareto-optimal points, each theorem or result in the context of MOO typically refers to convergence or behavior relative to one such point, chosen based on the pareto optimal point corresponding to the limit point $\lambda _ *$ of the sequence {$\lambda _ t$}.

---

> > > > > ### Author Response · Authors · 2023-11-22
> > > > > **Response to Reviewer zAtp's Comments [Part 5]**
> > > > >
> > > > > > **Your Comment 12:** The measure $\mathbb{E}\left[\sum _ {s \in [S]} \lambda _ t^s \left[ f _ s(\mathbf{x} _ t) - f _ s(\mathbf{x} _ *) \right] \right]$ can be negative. See more discussions in (Liu & Vincente 2021). You need to make additional assumptions to make this a valid convergence metric.
> > > > >
> > > > > **Our Response:** Definition 3 in our initial submission was indeed not stated clearly. We also note that our metric is slightly different from that in (Liu & Vincente 2021), where we replace $\lambda _ *$ by $\lambda _ k$. In the following, we first restate Definition 3 more clearly (we have also updated Definition 3 acccordingly in the revision):
> > > > >
> > > > > **Definition 3** ($\epsilon$-Pareto stationarity). In MOO, a point $\mathbf{x} _ t$ is $\epsilon$-Pareto-stationary if for any $\epsilon>0$, there exists a set $\{ \lambda _ t^s>0, \forall s \in [S]: \sum _ {s=1}^{S} \lambda _ t^s =1\}$, such that the following conditions hold: 1) $\mathbb{E}||\sum _ {s\in [S]}\lambda _ t^s\nabla f _ s(\mathbf{x} _ t)||^2 \leq \epsilon$ for non-convex MOO problems; or 2) $\mathbb{E}[\sum _ {s \in [S]} \lambda _ t^s [ f _ s(\mathbf{x} _ t) - f _ s(\mathbf{x} _ *) ] ] \in [0,\epsilon]$ for strongly-convex MOO problems.
> > > > >
> > > > > Next, regarding your question on the potential non-positivity of the expression $\mathbb{E}[\sum _ {s \in [S]} \lambda _ t^s \left[ f _ s(\mathbf{x} _ t) - f _ s(\mathbf{x} _ *) \right] ]<\epsilon$, in what follows, we will show that it is always possible to make $\mathbb{E}[\sum _ {s \in [S]} \lambda _ t^s [ f _ s(\mathbf{x} _ t) - f _ s(\mathbf{x} _ *) ] ]$ non-negative, hence Definition 3 is meaningful convergence metric (i.e., Definition 3 is well-defined). To see this, from strong convexity, we have $f _ s(\mathbf{x} _ t) \geq f _ s(\mathbf{x} _ *) + \nabla f _ s^{\top}(\mathbf{x} _ *)(\mathbf{x} _ t-\mathbf{x} _ *) +\frac{\mu}{2}||\mathbf{x} _ t-\mathbf{x} _ *||^2$ for some $\mu>0$. Thus, for any $\lambda _ t^s>0$, $\forall s \in [S]$ with $\sum _ {s=1}^{S} \lambda _ t^s =1$, we have
> > > > >
> > > > > $\sum _ {s\in[S]} \lambda _ t^s[f _ s(\mathbf{x} _ t) - f _ s(\mathbf{x} _ *)]$
> > > > >
> > > > > $\geq \sum _ {s \in [S]} \lambda _ t^s \nabla f _ s^{\top}(\mathbf{x} _ *)(\mathbf{x} _ t-\mathbf{x} _ *) + \frac{\mu}{2}||\mathbf{x} _ t-\mathbf{x} _ *||^2$
> > > > >
> > > > > $= \sum _ {s \in [S]} \lambda _ t^s \nabla f _ s^{\top} \mathbf{d} + \frac{\mu}{2}||\mathbf{x} _ t-\mathbf{x} _ *||^2$,
> > > > >
> > > > > where we define $\mathbf{d} \triangleq \mathbf{x} _ t-\mathbf{x} _ *$ in the last equality for convenience. Since $\mathbf{x} _ *$ is Pareto-stationary and all objective functions are strongly convex, it follows that $\mathbf{x} _ *$ is also Pareto-optimal. Thus, there must exist at least one $\tilde{s} \in [S]$ such that $\nabla f _ {\tilde{s}}^{\top}(\mathbf{x} _ *) \mathbf{d} > 0$.
> > > > >
> > > > > Now, consider the following strategy for choosing $\lambda _ t^s$, $\forall s\in [S]$: For $\tilde{s}$, we choose a $\lambda _ t^{\tilde{s}}$-value that is close to 1. For all other $s \ne \tilde{s}$, we choose a small $\lambda _ t^s$-value that is close to 0. Then, by pushing the $\lambda _ t^{\tilde{s}}$-value toward 1 and other $\lambda _ t^s$-values towards 0 (whiling maintaining $\sum _ {s=1}^{S} \lambda _ t^s =1$), we can always make $\sum _ {s\in[S]} \lambda _ t^s[f _ s(\mathbf{x} _ t) - f _ s(\mathbf{x} _ *)]$ non-negative.
> > > > >
> > > > > Therefore, as long as an algorithm can find a $\lambda _ t^s$-convex combination such that $\sum _ {s\in[S]} \lambda _ t^s[f _ s(\mathbf{x} _ t) - f _ s(\mathbf{x} _ *)] \in [0, \epsilon]$, then it is a near Pareto-stationary point under the strongly convex setting.
> > > > >
> > > > > Meanwhile, we also want to point out that, due to research on MOO is still in its infancy, there is no standard consensus definition of Pareto-stationarity that is universally adopted in the MOO literature. The condition in Definition 3 is proposed by us, which is also part of the novelty of this paper. Also, it is worth noting that several existing papers, including [R1, R2, R3], employed similar metrics, rendering our results directly comparable to theirs. Additionally, we would also like to note that by using an additional assumption similar to Assumption 5.6 in [R3], it is possible to adopt a strong Pareto-stationarity condition in the strongly convex case, where $\lambda _ t^s$ is replaced by an optimal convex combination at $\mathbf{x} _ *$ for all $t$.
> > > > >
> > > > > [R1] Fliege, Jörg, A. Ismael F. Vaz, and Luís Nunes Vicente. "Complexity of gradient descent for multiobjective optimization." Optimization Methods and Software 34.5 (2019): 949-959
> > > > >
> > > > > [R2] Yang, H., Liu, Z., Liu, J., Dong, C., & Momma, M. (2023). Federated Multi-Objective Learning. arXiv preprint arXiv:2310.09866.
> > > > >
> > > > > [R3] Suyun Liu and Luis Nunes Vicente. The stochastic multi-gradient algorithm for multi-objective optimization and its application to supervised machine learning. Annals of Operations Research,pp. 1–30, 2021
> > > > >
> > > > > [R4] Hiroaki Mukai. Algorithms for multicriterion optimization, IEEE Transactions on Automatic Control, 25(2): 177-186, 1980.

---

> > ### Comment · Reviewer_zAtp · 2023-11-22
> > **Unjustified claims in the response**
> >
> > Thanks for the response.
> >
> > You mentioned that linear scalarization methods are limited to identifying the convex hull of the Pareto front ([R5,R6]) which I am fully aware of. However, there is no proof or reference that **"(stochastic) multi-gradient methods, including our proposed algorithms, have the capability to uncover the Pareto front."**
> >
> > Could you provide proof or reference for this claim?

---

> > > ### Author Response · Authors · 2023-11-23
> > > **Response to Reviewer zAtp's Second Round Comments**
> > >
> > > > **Your Comment:** You mentioned that linear scalarization methods are limited to identifying the convex hull of the Pareto front ([R5,R6]) which I am fully aware of. However, there is no proof or reference that "(stochastic) multi-gradient methods, including our proposed algorithms, have the capability to uncover the Pareto front."
> > > Could you provide proof or reference for this claim?
> > >
> > > **Our Response:** Thanks for your prompt response. For multi-gradient methods, the goal is to find an $\mathbf{x}^*$ such that $d^* = \sum _ {s \in[S]} \lambda_s \nabla f_s(x^*) = 0, \sum _ {s \in[S]} \lambda_s = 1, \lambda_s \geq 0$, as shown by the metrics $\| \sum _ {s \in[S]} \lambda_s \nabla f_s(x^*) \|^2 \rightarrow 0$ (i.e., we can not find a common descent direction $\mathbf{d}$ at $\mathbf{x}^*$ such that $\forall s \in [S], \mathbf{d}^{\top} \nabla f_s(x^*) < 0$). This means the solution generated by the multi-gradient methods is guaranteed to find a Pareto stationary point that lies in the Pareto front. Also, due to the dynamic weighting in multi-gradient-based methods, it is possible to achieve different Pareto-stationary points by using different hyper-parameters (e.g., learning rate, starting point, batch size if in stochastic multi-gradient-based methods, etc.). On the other hand, if $x$ is a Pareto stationary point in the Pareto font, then $\sum _ {s \in[S]} \lambda_s \nabla f_s(x) = 0$ (Lemma 2.1 in R[1]). Therefore, the vanilla MGDA method is guaranteed to converge to a Pareto stationary point. Moreover, several enhanced MGDA-based algorithms ([R2, R3]) have been shown to be capable of exploring the entire Pareto front.
> > >
> > > [R1] Fliege, Jörg, A. Ismael F. Vaz, and Luís Nunes Vicente. "Complexity of gradient descent for multiobjective optimization." Optimization Methods and Software 34.5 (2019): 949-959.)
> > > [R2] Michinari Momma, Chaosheng Dong, and Jia Liu. "A multi-objective/multi-task learning framework induced by pareto stationarity." International Conference on Machine Learning. PMLR, 2022.
> > > [R3] D. Mahapatra and Vaibhav Rajan. "Multi-task learning with user preferences: Gradient descent with controlled ascent in pareto optimization." International Conference on Machine Learning. PMLR, 2020.
> > >
> > > **Our Response:** Thanks for your prompt response. It seems the word "uncover" might be confusing in our response in the previous round. We want to clarify that "uncover" here means the possibility of obtaining a variety of solutions on the Pareto front (i.e., "exploring" the Pareto front). In contrast, linear scalarization can at best achieve only one Pareto stationary due to the use of fixed weights.
> > >
> > > More specifically, for multi-gradient methods, the goal is to find $x^*$ such that $d^* = \sum _ {s \in[S]} \lambda_s \nabla f_s(x^*) = 0, \sum _ {s \in[S]} \lambda_s = 1, \lambda_s \geq 0$, as shown by the metrics $\| \sum _ {s \in[S]} \lambda_s \nabla f_s(x^*) \|^2 \rightarrow 0$. This means the solution generated by the multi-gradient methods is guaranteed to find a point that lies in the Pareto front. Also, due to the dynamic weighting in multi-gradient-based methods, it is possible to achieve different Pareto-stationary points by using different hyper-parameters (e.g., learning rate, starting point, batch size if in stochastic multi-gradient-based methods, etc.).

---

### Official Review · Reviewer_jjKe · 2023-11-05

**Soundness:** 4 excellent
**Presentation:** 3 good
**Contribution:** 3 good
**Rating:** 8
**Confidence:** 3

**Summary:**

This paper gives a systematic study on variance-reduction-aided gradient-based algorithms for multi-objective optimization. A new variance reduction multi-gradient estimator is proposed by combining periodic full multi-gradients and recursive correction with batch gradients, followed by a momentum-based variant. The adaptive-batching technique is further introduced to eschew the need of computing full gradients. Theoretical analysis on convergence rate and sample complexity are provided for all the proposed algorithms, showing superiority over previous stochastic multi-gradient algorithms. Experiments on three datasets verify the theoretical claims in this work.

**Strengths:**

1. This paper conducts a systematic study on the VR-aided multi-gradient method. Various versions of VR-based algorithms are proposed and supported by theoretical analysis, which may inspire future research in this field.

2. This paper is technical sound. The convergence analysis is comprehensive and non-trivial.

3. This paper is well-written in general and easy to follow.

**Weaknesses:**

1. The presentation of adaptive-batching versions is a bit ambiguous. I am not sure whether the adaptive batch is applied to the $q$-periodic full gradient or to each step. Adding more background knowledge on adaptive batch technique or a diagram for STIMULUS$^+$ would be helpful. In addition, it is unclear how to decide the batch size in experiments.

2. Besides SMGD and MOCO, CR-MOGM (Zhou et al., 2022b) should also be considered in experiments as a SOTA method.

**Questions:**

My main concerns are given in the weaknesses part.

---

> ### Author Response · Authors · 2023-11-22
> **Response to Reviewer jjKe's Comments**
>
> > **Your Comment 1:** The presentation of adaptive-batching versions is a bit ambiguous. I am not sure whether the adaptive batch is applied to the periodic full gradient or to each step. Adding more background knowledge on adaptive batch technique or a diagram for STIMULUS would be helpful. In addition, it is unclear how to decide the batch size in experiments.
>
> **Our Response:** Thank you for your comments regarding the presentation of the adaptive-batching versions in our algorithms, STIMULUS+/STIMULUS-M+. We would like to clarify our adaptive batch size approach:
>
> 1. **To periodic full gradient or to each step:** The adaptive batch size is applied to replace the *periodic full gradient*. Specifically, we modify the gradient estimators in Line 5 Algorithm 1 in the t-th iteration that satisfies $\mathrm{mod}(t, q) = 0$ as follows (i.e., every $q$ steps): $\mathbf{u} _ t^s = \frac{1}{|\mathcal{N} _ s|} \sum _ {j\in \mathcal{N} _ s}\nabla f _ s(\mathbf{x} _ {t};\xi _ {sj} )$, $\forall s \in [S],$ where $\mathcal{N} _ s$ represents an $\epsilon$-adaptive batch sampled from the dataset uniformly at random. In the revision, we will provide more background knowledge on the adaptive batch technique as you suggested.
>
>  2. **How to decide the batch size:** We choose the batch size adaptive to $\epsilon$ as: $|\mathcal{N} _s| = \min \\{ c _ \gamma \sigma^2\gamma _{t}^{-1}, c _ \epsilon \sigma^2 \epsilon^{-1}, n \\}$, where we use $c _ \gamma \geq 8$, $c _ {\epsilon}\geq \eta$ for the non-convex case and use $c _ {\gamma}\geq \frac{8\mu}{\eta}, c _ {\epsilon}\geq \frac{\mu}{2}$ for the strongly-convex case. In our experiments, as shown in our paper, we choose constant $c _ {\gamma}=c _ {\epsilon}=c = 32$ and solution accuracy $\epsilon = 10^{−3}$.
>
>
> ----------------
>
> > **Your Comment 2:** Besides SMGD and MOCO, CR-MOGM (Zhou et al., 2022b) should also be considered in experiments as a SOTA method.
>
>
> **Our Response:** Thank you for your suggestion to include CR-MOGM (Zhou et al., 2022b) in our experimental comparisons. We note that that CR-MOGM can be viewed as a momentum version of the SGD method tailored for solving MOO problems. Also, both MOCO and CR-MOGM utilize a similar approach in employing momentum-based SGD for MOO.
>
>
>
> However, there are key distinctions between these two methods. More specifically, CR-MOGM can be viewed as a special case of the more general MOCO framework. By contrast, MOCO not only tackles the general MOO problem formulation, but also considers MOO problems with special structures, such as regularization and bilevel structures. Given the facts that i) MOCO is more recent and state-of-the-art and ii) The similarities in foundational technique between MOCO and CR-MOGM provide a reasonable basis to anticipate comparable experimental outcomes. In future work, we aim to conduct additional experiments specifically focusing on CR-MOGM to further explore and validate this hypothesis.

---

### Official Review · Reviewer_ccVZ · 2023-11-05

**Soundness:** 3 good
**Presentation:** 3 good
**Contribution:** 2 fair
**Rating:** 5
**Confidence:** 4

**Summary:**

The paper considers multi-objective learning problems based on gradient methods. The paper introduce a novel stochastic gradient methods with variance-reduction to minimize multi-objective learning problems. The algorithm is a variant of the spider algorithm in Fang et al 2018 from single-objective learning to multi-objective learning. The algorithm first builds a common descent direction based on stochastic gradients, using the recursive gradient estimates to reduce variance. The paper further improves the efficiency by introducing the momentum scheme and the adaptive batching. Theoretical convergence and sample complexity are present for both nonconvex and strongly convex problems, under a smoothness assumption on loss functions. Experimental results are also presented to verify the efficiency of the proposed algorithm.

**Strengths:**

The paper introduce several stochastic algorithms for multi-objective optimization problems, which are more challenging than the single-objective problems. The paper The algorithms have better convergence rates and sample complexity than the existing results. The paper is clearly written and the main results are clearly presented.

**Weaknesses:**

As far as I see, the theoretical analysis seems to be problematic. For example, Theorem 1 gives convergence rates on $\frac{1}{T}\sum_{t=0}^{T-1}\|d_t\|^2$. However, the terms $d_t$ are just common descent directions built based on stochastic gradients (which is similar to the stochastic gradient in SGD). According to Definition 3 and the paragraph above, the quantity to our interest is $d=\lambda^\top\nabla F(\mathbf{x})$. note that $F(\mathbf{x})$ are the true objective functions, instead of the stochastic functions randomly sampled in the optimization process. Therefore, Theorem 1 does not give convergence rates on the $\epsilon$-stationarity, and the convergence in terms of $\|d_t\|^2$ does not show the real behavior of the algorithm. Furthermore, as far as I see from the proof of Theorem 1, one can get convergence rates of $\|d_t\|^2$ if only $q=|\mathcal{A}|$, even if $q$ is very small. In this case, one can choose very $q$ to derive the same convergence rates for $\|d_t\|^2$, but with much less sample complexity.

Definition 3 implicitly assumes that all $f_s$ should have the same minimizer $x_*$, which is a very strong assumption. In multi-objective optimization, it is very unlikely that we have the same minimizer for all tasks. Then, the convergence rates for strongly convex problems are restrictive.

**Questions:**

Can we derive convergence rates in terms of $d_t=\lambda^\top \nabla F(\mathbf{x}_t)$? Indeed, the convergence of $\lambda^\top \nabla F(\mathbf{x}_t)$ reflects the convergence behavior of the algorithm.

Can we relax the assumption in Definition 3 by letting the $t$-th task have a minimizer $\mathbf{x}_*^t$, i.e., each task has its own minimizer?

In Corollary 2, if $\epsilon>\mu$, then $\log (\mu/\epsilon)<0$. In this case, it seems that the result would no longer hold?

Minor issues:

- Eq (2): there is a missing summation over $\mathcal{A}$
- Eq (4): there is a missing constraint on the nonnegativity of $\lambda$
- Line 4 of Algorithm 1: Eq (4) does not give formula to compute $u_t^s$

---

> ### Author Response · Authors · 2023-11-22
> **Response to Reviewer ccVZ's Comments [Part 1]**
>
> > **Your Comment 1:** As far as I see, the theoretical analysis seems to be problematic. For example, Theorem 1 gives convergence rates on  $\frac{1}{T} \sum _ {t=0}^{T-1}\left||d _ t\right||^2$. However, the terms are just common descent directions built based on stochastic gradients (which is similar to the stochastic gradient in SGD). According to Definition 3 and the paragraph above, the quantity to our interest is $d=\lambda^{\top} \nabla F(\mathbf{x})$. note that $F(\mathbf{x})$ are the true objective functions, instead of the stochastic functions randomly sampled in the optimization process. Therefore, Theorem 1 does not give convergence rates on the $\epsilon$-stationarity, and the convergence in terms of does not show the real behavior of the algorithm. Furthermore, as far as I see from the proof of Theorem 1, one can get convergence rates of if only, even if is very small. In this case, one can choose very to derive the same convergence rates for , but with much less sample complexity. Can we derive convergence rates in terms of $d _ t=\lambda^{\top} \nabla F\left(\mathbf{x} _ t\right)$ ? Indeed, the convergence of $\lambda^{\top} \nabla F\left(\mathbf{x} _ t\right)$ reflects the convergence behavior of the algorithm.
>
>
> **Our Response:** Thank you for your detailed observations regarding the theoretical analysis in our paper, particularly concerning the convergence rates and the interpretation of $||\mathbf{d} _ t||^2$in Theorem 1. Your insights have led us to reconsider and refine our approach:
>
> In our initial submission, we derived the convergence rate for $||\mathbf{d} _ t||^2=||\sum _ {s \in [S]}\lambda _ {t}^{s} \mathbf{u} _ {t}^s||^2$, where $\mathbf{u} _ {t}^s$ represents the gradient-estimation-based moving direction with VR-adjustment, which can be seen in Line 4-11 in Algorithm 1.
>
>
> We understand your concern that this might not fully capture the convergence behavior in terms of the true gradient. Based on your feedback, we have revised our approach to derive the convergence rate for $||\sum _ {s \in [S]}\lambda _ {t}^{s} \nabla F\left(\mathbf{x} _ t\right)||^2$, which more accurately reflects the convergence behavior in terms of the true objectives. We have updated our proofs to reflect this new metric, ensuring that our theoretical results align more closely with the actual behavior of the algorithm in converging to $\epsilon$-stationarity.
>
>
>
> ----------------
>
> > **Your Comment 2:**
> Definition 3 implicitly assumes that all $f _ s$ should have the same minimizer, which is a very strong assumption. In multi-objective optimization, it is very unlikely that we have the same minimizer for all tasks. Then, the convergence rates for strongly convex problems are restrictive.
>
> **Our Response:** Thanks for your comments. It appears that there is some misunderstanding on the notation $\mathbf{x} _ *$, which represnts a **Pareto-stationary point, not a common minimizer for all $f _ s(\cdot)$**. In the strongly convex setting, a Pareto-stationary solution further implies Pareto-optimal solution [R4]. More specifically, a solution $\mathbf{x} _ *$ is considered Pareto-optimal if there is no other feasible solution that would **improve one objective without causing at least one other objective to worsen**. Essentially, a Pareto-optimal solution represents a point of equilibrium where no objective can be improved without compromising others.
>
> Also, we want to clarify Definition 3 in our initial submission by restating it as follows (with further elaborations):
>
> **Definition 3** ($\epsilon$-Pareto stationarity). In MOO, a point $\mathbf{x} _ t$ is $\epsilon$-Pareto-stationary if for any $\epsilon>0$, there exists a set $\{ \lambda _ t^s>0, \forall s \in [S]: \sum _ {s=1}^{S} \lambda _ t^s =1\}$, such that the following conditions hold: 1) $\mathbb{E}||\sum _ {s\in [S]}\lambda _ t^s\nabla f _ s(\mathbf{x} _ t)||^2 \leq \epsilon$ for non-convex MOO problems; or 2) $\mathbb{E}[\sum _ {s \in [S]} \lambda _ t^s [ f _ s(\mathbf{x} _ t) - f _ s(\mathbf{x} _ *) ] \in [0,\epsilon]$ for strongly-convex MOO problems.

---

> > ### Author Response · Authors · 2023-11-22
> > **Response to Reviewer ccVZ's Comments [Part 2]**
> >
> > Next, we justify why the expression $\mathbb{E}[\sum _ {s \in [S]} \lambda _ t^s [ f _ s(\mathbf{x} _ t) - f _ s(\mathbf{x} _ *) ] ] \in [0,\epsilon]$ in Definition 3 does **not imply** $\mathbf{x} _ *$ being a common minimizer. First, from strong convexity, we have $f _ s(\mathbf{x} _ t) \geq f _ s(\mathbf{x} _ *) + \nabla f _ s^{\top}(\mathbf{x} _ *)(\mathbf{x} _ t-\mathbf{x} _ *) +\frac{\mu}{2}||\mathbf{x} _ t-\mathbf{x} _ *||^2$ for some $\mu>0$. Thus, for any $\lambda _ t^s>0$, $\forall s \in [S]$ with $\sum _ {s=1}^{S} \lambda _ t^s =1$, we have
> >
> > $\sum _ {s\in[S]} \lambda _ t^s[f _ s(\mathbf{x} _ t) - f _ s(\mathbf{x} _ *)]$
> >
> > $\geq \sum _ {s \in [S]} \lambda _ t^s \nabla f _ s^{\top}(\mathbf{x} _ *)(\mathbf{x} _ t-\mathbf{x} _ *) + \frac{\mu}{2}||\mathbf{x} _ t-\mathbf{x} _ *||^2$
> >
> > $= \sum _ {s \in [S]} \lambda _ t^s \nabla f _ s^{\top} \mathbf{d} + \frac{\mu}{2}||\mathbf{x} _ t-\mathbf{x} _ *||^2$,
> >
> > where we define $\mathbf{d} \triangleq \mathbf{x} _ t-\mathbf{x} _ *$ in the last equality for convenience. Since $\mathbf{x} _ *$ is **Pareto-stationary** and all objective functions are strongly convex, it follows that $\mathbf{x} _ *$ is also **Pareto-optimal** [R4]. Thus, there must exist at least one $\tilde{s} \in [S]$ such that $\nabla f _ {\tilde{s}}^{\top}(\mathbf{x} _ *) \mathbf{d} > 0$ (i.e., there must exists a least one objective function whose objective value worsens if moving along $\mathbf{d}$).
> >
> > To show the *existence* of $\{ \lambda _ t^s>0, \forall s \in [S]: \sum _ {s=1}^{S} \lambda _ t^s =1\}$ for the inequality $\mathbb{E}[\sum _ {s \in [S]} \lambda _ t^s [ f _ s(\mathbf{x} _ t) - f _ s(\mathbf{x} _ *) ] ]< [0,\epsilon]$ to hold, consider the following strategy for choosing $\lambda _ t^s$, $\forall s\in [S]$: For $\tilde{s}$, we choose a $\lambda _ t^{\tilde{s}}$-value that is close to 1. For all other $s \ne \tilde{s}$, we choose a small $\lambda _ t^s$-value that is close to 0. Then, by pushing the $\lambda _ t^{\tilde{s}}$-value toward 1 and other $\lambda _ t^s$-values towards 0 (whiling maintaining $\sum _ {s=1}^{S} \lambda _ t^s =1$), we can always make $\sum _ {s\in[S]} \lambda _ t^s[f _ s(\mathbf{x} _ t) - f _ s(\mathbf{x} _ *)]$ non-negative. Therefore, as long as an algorithm can find a $\lambda _ t^s$-convex combination such that $\sum _ {s\in[S]} \lambda _ t^s[f _ s(\mathbf{x} _ t) - f _ s(\mathbf{x} _ *)] \in [0, \epsilon]$, then it is a near Pareto-stationary point under the strongly convex setting.
> >
> >
> >
> > We would like to highlight that this metric $\mathbb{E}[\sum _ {s \in [S]} \lambda _ t^s [ f _ s(\mathbf{x} _ t) - f _ s(\mathbf{x} _ *) ] ]< [0,\epsilon]$ has been adopted in the MOO literature for strongly-convex MOO problems, as evidenced in references [R1, R2, R3].
> >
> > [R1]S. Liu and L. N. Vicente, “The stochastic multi-gradient algorithm for multi-objective optimization and its application to supervised machine learning,” Annals of Operations Research, pp.1–30, 2021
> >
> > [R2]J. Fliege, A. I. F. Vaz, and L. N. Vicente, “Complexity of gradient descent for multiobjective
> > optimization,” Optimization Methods and Software, vol. 34, no. 5, pp. 949–959, 2019
> >
> > [R3]Yang, Haibo, et al. "Federated Multi-Objective Learning." arXiv preprint arXiv:2310.09866 (2023).
> >
> > [R4] Hiroaki Mukai. Algorithms for multicriterion optimization, IEEE Transactions on Automatic Control, 25(2): 177-186, 1980.
> >
> >
> > ------------------------------
> >
> > > **Your Comment 3:** Can we relax the assumption in Definition 3 by letting the $t$-th task have a minimizer $\mathbf{x} _ *^t$, i.e., each task has its own minimizer?
> >
> > **Our Response:** Thanks for your question. Again, it appears that this question is due to a misunderstanding of MOO. Yes, we **do allow** each task in MOO to have their own minimizer. However, the notion of minimizer is **irrelevant** in MOO. Exactly because each task could have its own minimizer, it is in general impossible to find a common solution that can minimize all objective functions simultaneously. Moreover, in MOO, the tasks (objectives) could even be **conflicting** (i.e., improving one objective could worsen some others). Thus, achiving a minimizer for one objective function could signifiantly worsen another objective function. As a result, a more suitable notion of "optimality" in MOO is the **Pareto-optimality**.
> >
> > Specifically, the goal of MOO is to find a solution in the "Pareto front" (i.e., the set of all Pareto-optimal solutions). A solution $\mathbf{x} _ *$ is considered Pareto-optimal if it cannot improve any objective without worsening at least one other objective. Also, it is important to note that a Pareto-optimal solution may not be an minimizer of any objective function. Again, a Pareto-optimal solution represents a point of **equilibrium** where no objective can be improved without compromising others.

---

> > > ### Author Response · Authors · 2023-11-22
> > > **Response to Reviewer ccVZ's Comments [Part 3]**
> > >
> > > > **Your Comment 4:** In Corollary 2 , if $\epsilon>\mu$, then $\log (\mu / \epsilon)<0$. In this case, it seems that the result would no longer hold?
> > >
> > > **Our Response:** Thanks for your comment. There is no error if $\epsilon > \mu$ in Corollary 2.
> > > Recall that, in Corollary 2, where we show that the overall sample complexity of SITIMULUS/STIMULUS+ for solving strongly-convex MOO is $\mathcal{O}\left(n+ \sqrt{n} \ln ({\mu/\epsilon})\right)$. Depending on the relationships between $\mu$ and $\epsilon$, we have different interpretations as follows:
> > >
> > > **1. The $\epsilon \leq \mu$ case:** In this case, $\log(\mu/\epsilon)\geq 0$. Note that $\epsilon>0$ denotes the Pareto-stationarity gap and $\mu>0$ denotes the strong convexity modulus. If $\epsilon$ is small in the sense that $\epsilon \leq \mu$, then it is natural that converging to within this $\epsilon$-Pareto-stationarity gap takes more training samples, which increases at a rate $\mathcal{O}(\ln(\mu/\epsilon))$ as $\epsilon \rightarrow 0$.
> > >
> > > **2. The $\epsilon > \mu$ case:** In this case, if $\log(\mu/\epsilon)<0$. However,, this does **not** mean the result is invalid. Rather, it implies the following interpretation: As $\epsilon$ increases, the Pareto-stationarity convergence criterion becomes more "relaxed"  increases. In particular, when $\epsilon$ grows larger than $\mu$, indicating that the convergence criterion is so relaxed, then it takes fewer and fewer samples to reach the convergence criterion, the number of samples **shrinks** at a rate $\mathcal{O}(\ln(\mu/\epsilon))$ as $\epsilon$ continues to increase after $\epsilon > \mu$.
> > >
> > >
> > >
> > >
> > > > **Your Comment 5:** Minor issues: 1.Eq (2): there is a missing summation over $\mathcal{A}$. 2. Eq (4): there is a missing constraint on the nonnegativity of $\lambda$. 3. Line 4 of Algorithm 1: Eq (4) does not give formula to compute $u _ t^s$.
> > >
> > > **Our Response:** Thanks for catching these issue. The corrections are as follows:
> > >
> > > 1) Regarding Eq. (2): The correct expression should be: $\mathbf{u} _ t^s = \mathbf{u} _ {t-1}^s + \frac{1}{|\mathcal{A}|} \sum _ {j\in \mathcal{A}}\left( \nabla f _ {sj} (\mathbf{x} _ {t};\xi _ {sj}  ) - \nabla f _ {sj} (\mathbf{x} _ {t-1};\xi _ {sj} ) \right), \text{for all }s \in [S].$
> > > 2) Modification for Eq. (4): The correct formulation of this optimization problem should be: $\min _ {\mathbf{\lambda _ t^s}\geq 0} \Big ||  \sum _ {s \in [S]} \lambda _ {t}^s \mathbf{u} _ {t}^s \Big||^2, \,\,
> > >     \mathrm{s.t.} \,\, \sum _ {s \in [S]} \lambda _ {t}^s = 1.$
> > > 3) Update for Line 5 in Algorithm 1: The computation of $\mathbf{u} _ t^s$ should be written as: $\mathbf{u} _ t^s=\frac{1}{n}\sum _ {j=1}^n \nabla f _ {sj} (\mathbf{x} _ {t};\xi _ {sj}  )$.

---

### Official Review · Reviewer_RiGJ · 2023-11-06

**Soundness:** 2 fair
**Presentation:** 2 fair
**Contribution:** 2 fair
**Rating:** 3
**Confidence:** 2

**Summary:**

This paper proposes  STIMULUS, which can achieve lower sample complexities than existing algorithms.

**Strengths:**

This paper proposes  STIMULUS, which can achieve lower sample complexities than existing algorithms.

**Weaknesses:**

There are many typos in this paper. Some proofs of this paper are unclear.
1. Eq. (23) sums both sides of Eq. (22) weighted with  $\lambda_t^s$  from $s\in S$. But why $\frac{1}{2\delta} \|\nabla f_s(x_t) - u_t^s\|^2$ is not weighted with $\lambda_t^s$?
2. Why does it hold that $\|\nabla f_s(x_t) - u_t^s\|^2 = \sum_{...} \|x_{i+1} - x_i\|^2 + \| \nabla f_s(x_{(n_t−1)q}) − u^s_{(n_t−1)q}\|^2  $ in Eq.(23)
3. In the Definition 3, why should $\mathbb{E}  [\sum_{s} \lambda_i^s (f_s(x_t) - f_s(x_*))]$  be non-positive? This is not pointed out and proved in this paper. If this value is not non-positive, it is less than $\epsilon$ is not meaningful. Furthermore, what is the meaning of $i$ in the notation $\lambda_i^s$.
4. In the Line-7, it should be ``gradient'' other than ``graident''.
5. This paper consider the case that $f_s(x)$ are of the finite sum form. However, detailed description of finite sum form is lacked.

**Questions:**

No

---

> ### Author Response · Authors · 2023-11-22
> **Response to Reviewer RiGJ's Comments [Part 1]**
>
> > **Your Comment 1:**  Eq. (23) sums both sides of Eq. (22) weighted with $\lambda_t^s$ from $s \in S$. But why $\frac{1}{2 \delta}\left|\nabla f_s\left(x_t\right)-u_t^s\right|^2$ is not weighted with $\lambda_t^s$ ?
>
> **Our Response:** Thanks for your question. To see this, note that $\sum_{s \in [S]} \lambda_t^{s}=1$ and $\|\nabla f_s(\mathbf{x}_{\left(n_t-1\right) q}) - \mathbf{u}_{\left(n_t-1\right) q}^s \|^2 =0$.
>
> It then follows that:
>
> $\sum_{s \in [S]} \lambda_t^{s} || \nabla f_s(\mathbf{x}_t)-\mathbf{u}_t^s||^2$
>
> $\leq \frac{L^2}{|\mathcal{A}|} \sum _{s \in [S]} \lambda _t^{s}\sum _{i=\left(n _t-1\right) q}^t ||\mathbf{x} _{i+1}-\mathbf{x} _{i}||^2 + \sum
>  _{s \in [S]} \lambda _t^{s}||\nabla f _s(\mathbf{x} _{\left(n _t-1\right) q}) - \mathbf{u} _{\left(n _t-1\right) q}^s ||^2$
>
> $=\frac{L^2}{|\mathcal{A}|} \sum _{i=\left(n _t-1\right) q}^t ||\mathbf{x} _{i+1}-\mathbf{x} _{i}||^2 + \sum _{s \in [S]} \lambda _t^{s}||\nabla f _s(\mathbf{x} _{\left(n_t-1\right) q}) - \mathbf{u} _{\left(n _t-1\right) q}^s ||^2$
>
> $= \frac{L^2}{|\mathcal{A}|} \sum_{i=\left(n_t-1\right) q}^t ||\mathbf{x} _{i+1}-\mathbf{x} _{i}||^2.$
>
> Thus, we conclude our results outlined in Eq. (23). Further proof details and expanded explanations can be found in the updated version of our paper.
>
>
> ---------------
>
> > **Your Comment 2:**  Why does it hold that $\left|\nabla f_s\left(x_t\right)-u_t^s\right|^2=\sum_{\ldots}\left|x_{i+1}-x_i\right|^2+\left|\nabla f_s\left(x_{\left(n_t-1\right) q}\right)-u_{\left(n_t-1\right) q}^s\right|^2$ in Eq.(23）
>
> **Our Response:** Thanks for your question. There appears to be some misunderstandings. We want to clarify that the equality relationship $\left|\nabla f_s\left(x_t\right)-u_t^s\right|^2=\sum_{\ldots}\left|x_{i+1}-x_i\right|^2+\left|\nabla f_s\left(x_{\left(n_t-1\right) q}\right)-u_{\left(n_t-1\right) q}^s\right|^2$ mentioned by the reviewer is *never* used in Eq. (23).
>
> Rather, in the latest version of our paper as shown in Eq. (25), we have used the following *inequality relationship*: $\left||\nabla f _s\left(\mathbf{x} _t\right)-\mathbf{u} _t^s\right||^2 \leq \frac{L^2}{|\mathcal{A}|} \sum _{i=\left(n _t-1\right) q}^t\left||\mathbf{x} _{i+1}-\mathbf{x} _i\right||^2+||\nabla f_s\left(\mathbf{x} _{\left(n_t-1\right) q}\right)-\mathbf{u} _{\left(n_t-1\right) q}^s||^2$. This inequality is stated and proved in Lemma 1 (cf. Eq. (8)).
>
>
> --------------------

---

> > ### Author Response · Authors · 2023-11-22
> > **Response to Reviewer RiGJ's Comments [Part 2]**
> >
> > > **Your Comment 3:**  In the Definition 3, why should $\mathbb{E} [\sum _s \lambda _i^s\left(f _s\left(x _t\right)-f _s\left(x
> >  _*\right)\right) ]$ be non-positive? This is not pointed out and proved in this paper. If this value is not non-positive, it is less than $\epsilon$ is not meaningful. Furthermore, what is the meaning of $i$ in the notation $\lambda _i^s$.
> >
> > **Our Response:** Thanks for your question. This confusion may be caused by a typo (our apologies). The correct expression should be $\mathbb{E}\left[\sum _{s \in [S]} \lambda _t^s \left[ f_s(\mathbf{x} _t) - f _s(\mathbf{x} _*) \right] \right] < \epsilon$. In this revision, we have carefully reviewed our paper and corrected all such typos. With the updated expression, in Definition 3, we define the $\epsilon$-stationarity for both non-convex and strongly-convex multi-objective optimization (MOO) problems. In our initial submission, the $\lambda_t^s$-values were not stated very clearly, which could be another reason that caused the confusion. In the following, we restate Definition 3 more clearly (we have also updated Definition 3 accordingly in the revision):
> >
> > **Definition 3** ($\epsilon$-Pareto stationarity). In MOO, a point $\mathbf{x} _t$ is $\epsilon$-Pareto-stationary if for any $\epsilon>0$, there exists a set $\{ \lambda _t^s>0, \forall s \in [S]: \sum _{s=1}^{S} \lambda _t^s =1\}$, such that the following conditions hold: 1) $\mathbb{E}\|\sum _{s\in [S]}\lambda _t^s\nabla f _s(\mathbf{x} _t)\|^2 \leq \epsilon$ for non-convex MOO problems; or 2) $\mathbb{E}[\sum _{s \in [S]} \lambda _t^s [ f_s(\mathbf{x} _t) - f_s(\mathbf{x} _*) ] ]\in [0,\epsilon]$ for strongly-convex MOO problems.
> >
> > Next, regarding your question on the potential non-positivity of the expression $\mathbb{E}[\sum _{s \in [S]} \lambda _t^s \left[ f _s(\mathbf{x} _t) - f _s(\mathbf{x} _*) \right] ]<\epsilon$,
> >
> > in what follows, we will formally prove that it is always possible to make $\mathbb{E}[\sum _{s \in [S]} \lambda _t^s [ f _s(\mathbf{x} _t) - f _s(\mathbf{x} _*) ] ]$ non-negative, hence our definition is **always meaningful** (i.e., not ill-defined).
> >
> > **Lemma 1:** *In the strongly convex case, there always exists a set $\{ \lambda _t^s>0, \forall s \in [S]: \sum _{s=1}^{S} \lambda
> >  _t^s =1\}$ such that
> >
> >  $\mathbb{E}[\sum _{s \in [S]} \lambda _t^s [ f _s(\mathbf{x} _t) - f _s(\mathbf{x} _*) ] ] \geq 0$.
> >
> > *Proof.* From strong convexity, we have
> >
> > $f _s (\mathbf{x} _t) \geq f _s(\mathbf{x} _ *)+ \nabla f _s ^{\top}(\mathbf{x} _ *)(\mathbf{x} _t-\mathbf{x} _ *)+\frac{\mu}{2}||\mathbf{x} _t-\mathbf{x} _ *||^2$  for some $\mu>0$.
> >
> > Thus, for any $\lambda _t^s>0$, $\forall s \in [S]$ with $\sum _{s=1}^{S} \lambda _t^s =1$, we have
> >
> > $\sum _{s\in[S]} \lambda _t^s[f _s(\mathbf{x} _t) - f _s(\mathbf{x} _*)]$
> >
> > $\geq \sum _{s \in [S]} \lambda _t^s \nabla f _s^{\top}(\mathbf{x} _ *)(\mathbf{x} _t-\mathbf{x} _ *) + \frac{\mu}{2}\|\mathbf{x} _t-\mathbf{x} _ *\|^2$
> >
> > $= \sum_{s \in [S]} \lambda _t^s \nabla f _s^{\top} \mathbf{d} + \frac{\mu}{2}\|\mathbf{x} _t-\mathbf{x} _ *\|^2$,
> >
> > where we define $\mathbf{d} \triangleq \mathbf{x} _t-\mathbf{x} _ *$ in the last equality for convenience. Since $\mathbf{x} _ *$ is Pareto-stationary and all objective functions are strongly convex, it follows that $\mathbf{x} _ *$ is also Pareto-optimal. Thus, there must exist at least one $\tilde{s} \in [S]$ such that $\nabla f _{\tilde{s}}^{\top}(\mathbf{x} _ *) \mathbf{d} > 0$.
> >
> > Now, consider the following strategy for choosing $\lambda_t^s$, $\forall s\in [S]$: For $\tilde{s}$, we choose a $\lambda_t^{\tilde{s}}$-value that is close to 1. For all other $s \ne \tilde{s}$, we choose a small $\lambda_t^s$-value that is close to 0. Then, by pushing the $\lambda_t^{\tilde{s}}$-value toward 1 and other $\lambda_t^s$-values towards 0 (whiling maintaining $\sum_{s=1}^{S} \lambda_t^s =1$), we can always make $\sum _{s\in[S]} \lambda _t^s[f_s(\mathbf{x} _t) - f _s(\mathbf{x} _ *)]$ non-negative. This completes the proof. $\square$

---

> > > ### Author Response · Authors · 2023-11-22
> > > **Response to Reviewer RiGJ's Comments [Part 3]**
> > >
> > > Therefore, as long as an algorithm can find a $\lambda_t^s$-convex combination such that $\sum_{s\in[S]} \lambda_t^s[f_s(\mathbf{x} _t) - f_s(\mathbf{x} _ *)] \in [0, \epsilon]$, then it is a near Pareto-stationary point under the strongly convex setting.
> > >
> > > Meanwhile, we also want to point out that, due to research on MOO is still in its infancy, there is **no** standard consensus definition of Pareto-stationarity that is universally adopted in the MOO literature. The condition in Definition 3 is proposed by us, which is also part of the novelty of this paper. Also, it is worth noting that several existing papers, including [R1, R2, R3], employed similar metrics, rendering our results directly comparable to theirs. Additionally, we would also like to note that by using an additional assumption similar to Assumption 5.6 in [R3], it is possible to adopt a strong Pareto-stationarity condition in the strongly convex case, where $\lambda_t^s$ is replaced by an optimal convex combination at $\mathbf{x}_*$ for all $t$.
> > >
> > >
> > > [R1] Fliege, Jörg, A. Ismael F. Vaz, and Luís Nunes Vicente. "Complexity of gradient descent for multiobjective optimization." Optimization Methods and Software 34.5 (2019): 949-959
> > >
> > > [R2] Yang, H., Liu, Z., Liu, J., Dong, C., & Momma, M. (2023). Federated Multi-Objective Learning. arXiv preprint arXiv:2310.09866.
> > >
> > > [R3] Suyun Liu and Luis Nunes Vicente. The stochastic multi-gradient algorithm for multi-objective optimization and its application to supervised machine learning. Annals of Operations Research,pp. 1–30, 2021
> > >
> > > [R4] Hiroaki Mukai. Algorithms for multicriterion optimization, IEEE Transactions on Automatic Control, 25(2): 177-186, 1980.
> > >
> > >
> > > -----------------
> > >
> > > > **Your Comment 4:** In the Line-7, it should be gradient " ' other than graident".
> > >
> > > **Our Response:** Thank you for pointing out the typo. We have carefully reviewed our paper and corrected all such typos.
> > >
> > > -----------------
> > >
> > > > **Your Comment 5:** This paper consider the case that $f_s(x)$ are of the finite sum form. However, detailed description of finite sum form is lacked.
> > >
> > > **Our Response:**  Thank you for your feedback. We appreciate your observation, and we will address this issue by providing a more detailed description of the finite sum form of $f_s(x)$ in our paper Section 1. This will help clarify the context and enhance the understanding of our work.

---

### Meta-Review · Area_Chair_YKHj · 2023-12-06

**Metareview:**

The paper studies multi-objective learning problems. The paper introduces a new stochastic gradient methods to optimize the objective function, which extends the spider algorithm in Fang et al 2018 from single-objective learning to multi-objective learning. Theoretical convergence and sample complexity are given for both nonconvex and strongly convex problems.

**Strengths**
- A new optimization algorithm is presented to solve the multi-objective learning problem.
- The assumptions and main results are clearly stated.

**Weaknesses**
- Many technical flaws in the proof were found.
- Typos/errors span over the manuscript.
- The improvement over linear scalarization is unclear.

**Suggestions to authors**
- Authors are strongly suggested to correct the flaws in their proof and fix the typos and presentations.
- Authors are suggest to theoretically compare the proposed method to best known results in the same setting.

**Justification For Why Not Higher Score:**

Many flaws in proof were found.

**Justification For Why Not Lower Score:**

N/A

---

### Decision · Program_Chairs · 2024-01-16

Reject